# A Cdc42-mediated supracellular network drives polarized forces and *Drosophila* egg chamber extension

Anna Popkova[1,2], Orrin J. Stone[3], Lin Chen[1,4], Xiang Qin [1,5], Chang Liu[1,6], Jiaying Liu[1], Karine Belguise[1], Denise J. Montell [7], Klaus M. Hahn [3], Matteo Rauzi[2,8 ✉] & Xiaobo Wang [1,8 ✉]

Actomyosin supracellular networks emerge during development and tissue repair. These cytoskeletal structures are able to generate large scale forces that can extensively remodel epithelia driving tissue buckling, closure and extension. How supracellular networks emerge, are controlled and mechanically work still remain elusive. During *Drosophila* oogenesis, the egg chamber elongates along the anterior-posterior axis. Here we show that a dorsal-ventral polarized supracellular F-actin network, running around the egg chamber on the basal side of follicle cells, emerges from polarized intercellular filopodia that radiate from basal stress fibers and extend penetrating neighboring cell cortexes. Filopodia can be mechanosensitive and function as cell-cell anchoring sites. The small GTPase Cdc42 governs the formation and distribution of intercellular filopodia and stress fibers in follicle cells. Finally, our study shows that a Cdc42-dependent supracellular cytoskeletal network provides a scaffold integrating local oscillatory actomyosin contractions at the tissue scale to drive global polarized forces and tissue elongation.

[1] LBCMCP, Centre de Biologie Intégrative (CBI), Université de Toulouse, CNRS, UPS, 31062 Toulouse, France. [2] Université Côte d'Azur, CNRS, Inserm, iBV, Nice, France. [3] Department of Pharmacology and Lineberger Cancer Center, University of North Carolina at Chapel Hill, Chapel Hill, NC, USA. [4] Department of Anesthesia, Southwest Hospital, Third Military Medical University, 400038 Chongqing, People's Republic of China. [5] Department of Biophysics, School of Life Science and Technology, University of Electronic Science and Technology of China, 610054 Chengdu, Sichuan, People's Republic of China. [6] Department of Anaesthesia, The Central Hospital of Wuhan, Tongji Medical College, Huazhong University of Science and Technology, 430014 Wuhan, Hubei, People's Republic of China. [7] Molecular, Cellular and Developmental Biology Department, University of California, Santa Barbara, CA, USA. [8] These authors jointly supervised this work: Matteo Rauzi, Xiaobo Wang. ✉email: matteo.rauzi@univ-cotedazur.fr; xiaobo.wang@univ-tlse3.fr

Actomyosin networks are functionally organized in cells to generate contractile forces that drive cell shape changes. In epithelia, the architecture of actomysin networks can extend beyond the size of a single cell: this results in supracellular actomyosin structures emerging at the scale of a tissue or of the entire animal. Supracellular actomyosin networks are commonly reported in developmental processes and tissue repair during wound healing. These cytoskeletal structures provide large scale mechanical forces and can function as segregating barriers[1,2] or as mechanical actuators responsible for tissue remodeling as for example epithelial buckling[3], closure[4,5], and extension[6,7]. While we do have a good understanding of the molecular processes and their regulation controlling the cytoskeleton in the cell, little is known on how cellular properties and mechanics are integrated at the tissue level and finally at the animal level to give shape.

In this study, we focus on tissue extension and take advantage of a recently established model process: the elongation of the *Drosophila* egg chamber. The egg chamber is composed of a monolayer follicular epithelium surrounding a 16-cell germline cyst. During oogenesis, the egg chamber gradually changes its shape from round to elongated by extending along the anterior-posterior (AP) axis[8]. Tissue elongation occurs between stage 6 (S6) and S10B, and it is controlled by two distinct processes: global egg chamber fast rotation from S6 to S8 (refs. [9,10]) and oscillating contractions of basal non-muscle myosin II (Myo-II) between S9 and S10B[11]. We here report that during S9-S10B a supracellular actomyosin network along the dorsal-ventral (DV) axis is established via polarized intercellular filopodia that interdigitate.

Filopodia are dynamic, finger-like plasma membrane protrusions of cells that act as antennae to sense the mechanical and chemical environment, and thus they are often regarded as "sensory organelles"[12,13]. Filopodia are involved in many biological processes, such as growth cone guidance, cell migration, wound closure, and macrophage-induced cell invasion[12–14]. These thin membrane protrusions are 60–200 nm in diameter and contain parallel bundles of 10–30 actin filaments held together by actin-binding proteins[15,16]. The formation of parallel actin bundles and filopodia is initiated by the IRSp53-mediated plasma membrane bending and the recruitment of the small GTPase Cdc42 and its downstream effectors, including ENA/VASP, WASP/N-WASP, and mDia2 (refs. [17–21]). These Cdc42 effectors synergistically nucleate actin polymerization to deliver actin monomers to the filopodia tip, and thus the barbed end of the actin filaments is directed towards the protruding membrane[17–21].

In addition to chemical cue sensing, filopodia can probe the mechanical properties of the physical environment surrounding the cell (e.g., the extracellular matrix)[22–30], and eventually apply traction forces[31,32]. Nevertheless, it is still unknown whether cells use filopodia to mechanically sense each other and if filopodia mechanosensitivity plays a role in epithelial morphogenesis. Recently, filopodia have been reported to be present between follicular epithelial cells at basal domains[9]. Nevertheless, their regulation and function are yet unknown.

By using live-cell imaging together with genetic, optogenetic, and infrared (IR) femtosecond (fs) laser manipulations, here we demonstrate that (1) stress fibers at the basal domain of the *Drosophila* ovarian follicular epithelial cells exert polarized contractile forces parallel to the DV axis both at the intracellular and supracellular scales; (2) intercellular filopodia, which extend towards the dorsal and ventral sides in a polarized manner, can be mechanosensitive and function as cell–cell anchoring sites between stress fiber networks, and (3) both intercellular filopodia and intracellular stress fibers are under the control of the activity of the small GTPase Cdc42. Our data support the notion that intercellular filopodia function as guiding cues organizing F-actin stress fibers parallel to the egg chamber DV axis. Finally, a Cdc42-dependent supracellular F-actin network integrates local Myo-II-dependent cellular contractions to drive a global DV-polarized contraction force and AP-directed tissue elongation.

## Results

**Supracellular fibers emerge from interdigitating filopodia.** During egg chamber elongation at S9-S10, the actin stress fibers at the basal side of follicle cells are polarized and run parallel to the DV direction[33] (Fig. 1a). Actin stress fibers are periodically distributed along the AP axis with an interval of ~9 µm (as revealed by Fourier analysis in Fig. 1b and Supplementary Fig. 1) corresponding to follicle cell AP length. No periodic F-actin distribution is detected along the DV axis (Fig. 1c and Supplementary Fig. 1). This shows that F-actin has a discrete cellular organization along the AP axis but not along the DV axis. Remarkably, a supracellular actin fiber pattern emerges along the DV axis (Fig. 1a, d). To understand how filamentous actin structures spanning multiple cells emerge, we imaged isolated clones with mCD8 membrane marker to discern the organization of F-actin in single cells along the DV axis. Actin bundles radiate from the medial stress fiber network, and they normally extend to the cell dorsal and ventral cortexes thus forming polarized protrusions (Fig. 1e). By performing live imaging of Enabled (ENA), we could confirm that these protrusions are filopodia (Supplementary Fig. 2a, b). Filopodia of opposing cells interdigitate (Fig. 1f, g) and penetrate the actomyosin cortex of the neighboring cell (Fig. 1h). As shown in previous studies[34], E-cadherin based junctions are present at the sub-basal cortex including along the filopodia (Fig. 1i). While actin bundles extend from medial stress fibers to the sub-basal cell–cell contact zone forming filopodia, Myo-II is not enriched at this zone and concentrates instead in the medial region (Fig. 1j).

During stages 7–8, filopodia at the basal side of follicle cells are unidirectionally polarized along the DV axis, following the orientation of egg chamber rotation[9]. After egg chamber rotation arrests at the end of S8, filopodia gradually increase their length and become bidirectionally polarized; they reach their maximal length between middle and late S9, and then shorten in S10B (Supplementary Fig. 2c).

To summarize, at S9 a supracellular F-actin network parallel to the DV axis emerges from bidirectionally polarized filopodia that interdigitate and penetrate the actomyosin cortex of opposing cells (Supplementary Fig. 2d).

**Cdc42 controls filopodia and stress fiber formation.** It is known that filopodia formation is under the control of the *Rho*-family small GTPase Cdc42 (refs. [19,20]). We, therefore, assessed whether Cdc42 is involved in the control of intercellular filopodia in follicle cells. Firstly, we examined the localization of the Cdc42 protein which, like Rac1 and Rho1 proteins, appeared to be enriched at the sub-basal cell–cell contact zone of follicle cells (Fig. 2a and Supplementary Fig. 3a, b). Chemical and genetic inhibition of Cdc42 activity, but not Rac1 or Rho1 activity, reduced filopodia length (Fig. 2b, c and Supplementary Fig. 3c–f, h, i) and also affected the filopodia directionality (Supplementary Fig. 4). The inhibition of Cdc42 activity by its dominant-negative (DN) form or by the loss-of-function (LOF) mutation[35] also decreased the intensity and changed the central/lateral distribution of the basal actomyosin network (Fig. 2b, d–g and Supplementary Fig. 3h, j). In contrast, the inhibition of Rac1 or Rho1 activity did not affect the distribution of the medio-basal actomyosin network (Supplementary Fig. 3e, g). Furthermore, basal Myo-II periodic oscillations were still present with smaller

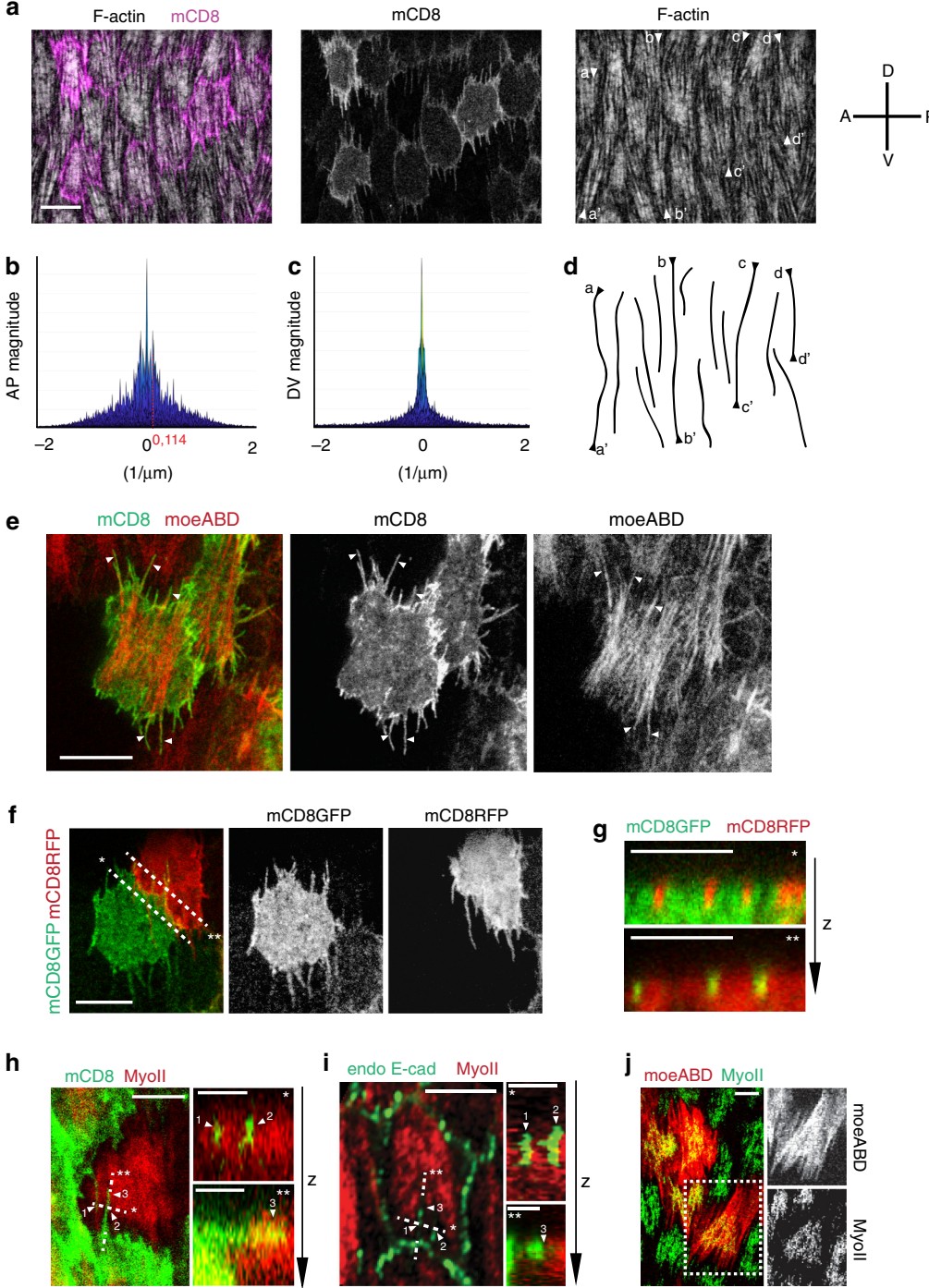

amplitude in cells with reduced Cdc42 activity in contrast to cells expressing Rho1DN (Fig. 2h, i and Supplementary Movies 1–3). These results show that Cdc42 is required for filopodia formation and to organize the basal actomyosin network whereas Rho1 controls the medial actomyosin oscillations.

**Cdc42 effectors control both stress fibers and filopodia**. To further investigate the role of Cdc42 in filopodia and stress fiber formation, we tested three Cdc42 downstream effectors: WASP, Enabled, and Dia[18–20].

RNAi inhibition of each effector resulted in similar phenotypes, as observed in Cdc42 downregulation experiments: (1) filopodia length reduction (Fig. 3a, b), (2) basal Myo-II intensity decrease

(Fig. 3a, c), and (3) central/lateral redistribution of stress fibers (Fig. 3a, d). Next, we characterized the effect of Cdc42 inhibition on these effectors. Cdc42 activity inhibition by its DN form strongly reduced the medio-basal distribution of all effectors, compared to wild-type cells (Fig. 3e–h). Moreover, concurrent expression of Enabled and Cdc42DN recovered the basal F-actin distribution and also returned the filopodia length back to normal (Fig. 3i–k). Similarly, significant but weaker recovery effects were observed when either WASP or Dia was concurrently expressed with Cdc42DN (Supplementary Fig. 5).

Finally, our data corroborate the notion that Cdc42 signaling controls filopodia formation and the organization of medio-basal stress fibers.

**Fig. 1 Supracellular actin stress fibers emerge from interdigitating filopodia. a** Basal view of follicle cells at stage 10A (S10A), with stochastic induction of mCD8GFP expression (magenta). F-actin is marked by phalloidin staining. A-P and D-V indicate the anterior-posterior and ventral-dorsal axes. Arrow heads indicate the extremities of supracellular F-actin fiber patterns. **b**, **c** Fourier analysis of F-actin distribution along the AP (**b**) and DV (**c**) axes. **d** Lines corresponding to the supracellular F-actin fiber patterns indicated in **a**. **e** Basal view of follicle clone cells co-expressing mCD8GFP and moeABD-mCherry. Arrow heads indicate filopodia that radiate from the medial-basal stress fiber network. **f** Basal view of two follicle clone cells in contact, one marked with mCD8GFP and the other with mCD8RFP showing filopodia interdigitating. **g** Cross-sections of cells along the two dashed lines shown in **f**: filopodia marked by mCD8RFP insert into the mCD8GFP-expressing cell (*, top panel); filopodia marked by mCD8GFP insert into the mCD8RFP-expressing cell (**, bottom panel). The z-axis indicates the basal to apical direction. **h** Left panel: close-up view of intercellular filopodia from follicle clone cells with stochastic expression of mCD8GFP (green). MyoII (red) is visualized by using a MyoII-mCherry construct. Right panels: cross-sections along the two dashed lines indicated in the left panel showing filopodia penetrating the actomyosin cortex of the neighboring cell. The z-axis indicates the basal to apical direction. **i** Left panel: close-up view of follicle cells marked by MyoII-mCherry (red) and E-cadherin-GFP (endo E-cad, green). Right panels: cross-sections along the two dashed lines indicated in the left panel showing filopodia penetrating the actomyosin cortex of a neighboring cell. The z-axis indicates the basal to apical direction. **j** Left panel: basal view of follicle cells, labeled with moeABD-mCherry and MyoII-GFP. Right panels: moeABD and MyoII shown separately. Scale bars are 10 μm in **a**, **e**, **f**, 5 μm in **h** left panel, **i** left panel, **g** and **j**, 2 μm in **h** right panels and **i** right panels. The results shown in **a**, **e–j** have been successfully repeated from the at least four independent experiments.

**Optogenetics reveals Cdc42 spatiotemporal-specific control**. Genetic modifications often lack spatial and temporal specificity. To probe the role of Cdc42 with greater spatiotemporal resolution, we constructed a photoactivatable Cdc42 (PA-Cdc42, described in Fig. 4a) to activate or inhibit Cdc42 activity with blue light. Similar to PA-Rac[36,37], blue light illumination can change the conformation of the LOV2 domain, thereby releasing the active site of Cdc42. The effectiveness of PA-Cdc42 was confirmed by GST-pulldown experiments in which a mutant, mimicking the illuminated state, strongly enhances the interaction between PA-Cdc42 and GST-PAK compared to the effect of a mutant mimicking the dark state (Supplementary Fig. 6a). We produced transgenic flies expressing PA-Cdc42 in the constitutively-active (CA) or DN form and a light-insensitive control in which the C450M mutation stabilizes the dark-state conformation of LOV2 (refs. [36,37]). For either the PA-Cdc42CA or PA-Cdc42DN form, exposure to blue light was able to significantly increase or decrease, respectively, the accumulation of PAK1-RBD-GFP at the basal cortical domain of follicle cells compared to the dark conditions (Supplementary Fig. 6b, c). Furthermore, compared with the C450M light-insensitive control, expression of PA-Cdc42DN in dark condition led to little effect on basal Myo-II and F-actin signals (Supplementary Fig. 6d, e). Taken together, these results demonstrate the effectiveness of PA-Cdc42CA and PA-Cdc42DN in vivo.

First, we tested the effect of Cdc42 photo-inhibition on intercellular filopodia in follicle cells at late S9 and S10A. The inhibition of Cdc42 activity by light reduced the length of filopodia within 10 min (Fig. 4b, c). Subsequently, we checked the effect of Cdc42 photo-inhibition on the basal actomyosin network of follicle cells at late S9 and S10A. Light-induced inhibition of Cdc42 activity gradually decreased the medio-basal distribution of F-actin and Myo-II signals within 20–30 min (Fig. 4d, e and Supplementary Movies 4 and 6). The C450M light-insensitive control caused no significant effect (Fig. 4d, e, Supplementary Fig. 7a, b and Supplementary Movies 5 and 7). In addition, the activation of PA-Cdc42DN at the apical domain had no prominent effect on basal stress fibers (Supplementary Fig. 7c–e). This demonstrates that stress fibers are under the control of basal Cdc42 activity.

Furthermore, we confirmed that Cdc42 photoactivation restores stress fiber formation in Cdc42-LOF mutant follicle cells. In these mutant cells, basal stress fibers are severely disrupted and misaligned. However, the activation of PA-Cdc42CA by light significantly recovered the formation and distribution of stress fibers within the medio-basal region in a short time period (Fig. 4f, g). Finally, PA-Cdc42CA activation in Cdc42-LOF mutant clones rescued filopodia length, while C450M

light-insensitive control caused no significant effect (Supplementary Fig. 7f, g).

Finally, these results further support Cdc42 spatiotemporal control over filopodia and stress fibers.

**Cdc42 non-cell autonomous role in stress fiber organization**. Since intercellular filopodia, which extend in a polarized manner along the DV axis and penetrate the neighboring cell cortex, are under the control of Cdc42, we tested whether Cdc42 plays a non-cell autonomous role in stress fiber organization. Three cell categories can be defined in follicular epithelial tissues containing Cdc42DN clones: (1) clone cells, (2) cells neighboring clone cells, and (3) cells not neighboring clone cells. While cells not neighboring Cdc42DN clones show the normal stress fiber distribution parallel to the DV axis, both clones and clone neighbors showed stress fiber mis-organization (Fig. 5a, b and Supplementary Fig. 8e). Cdc42DN clones and clone neighbors also showed the redistribution of the actin bundles (lateral vs. medial, e.g., Fig. 2d and Supplementary Fig. 8c, d). This led us to ask whether this local non-cell autonomous effect was due to reduced Myo-II contractility (hypothesis 1) or to the loss of intercellular filopodia (hypothesis 2), both being mediated by the inhibition of Cdc42 activity. To test the first hypothesis, we specifically reduced actomyosin contractility by interfering with the Rho1 pathway. The inhibition of Rho1 or ROCK activity in clone cells had no major effect on the neighboring and more distant wild-type cells, thus ruling out the first hypothesis (Supplementary Fig. 8 and ref. [11]). To test the second hypothesis, we took advantage of the PA-Cdc42DN optogenetic construct that allows spatiotemporal specific Cdc42 inhibition. We specifically photoactivated the sub-basal cell–cell contact zone of PA-Cdc42DN clone cells from which the filopodia radiate and monitored stress fiber organization in a neighboring wild-type cell (Fig. 5c). During photoactivation, filopodia of photoactivated Cdc42DN clones retracted and the actomyosin medial network of the neighboring wild-type cell was disrupted (Fig. 5d–i). Photoactivation had no effect on the light-insensitive control PA-Cdc42DN C450M cells (Supplementary Fig. 9). Finally, this shows that Cdc42 plays a non-cell autonomous role in stress fiber organization that could be mediated by intracellular filopodia.

**Cortical tension anisotropy is under the control of Cdc42**. Basal actomyosin contractility drives tissue elongation[11]. While it is known that basal Myo-II contractility is necessary to reduce basal cell area in a polarized manner along the DV axis[11], cell basal tension anisotropy has never been tested directly. We implemented IR fs laser ablation to dissect the actomyosin cytoskeleton

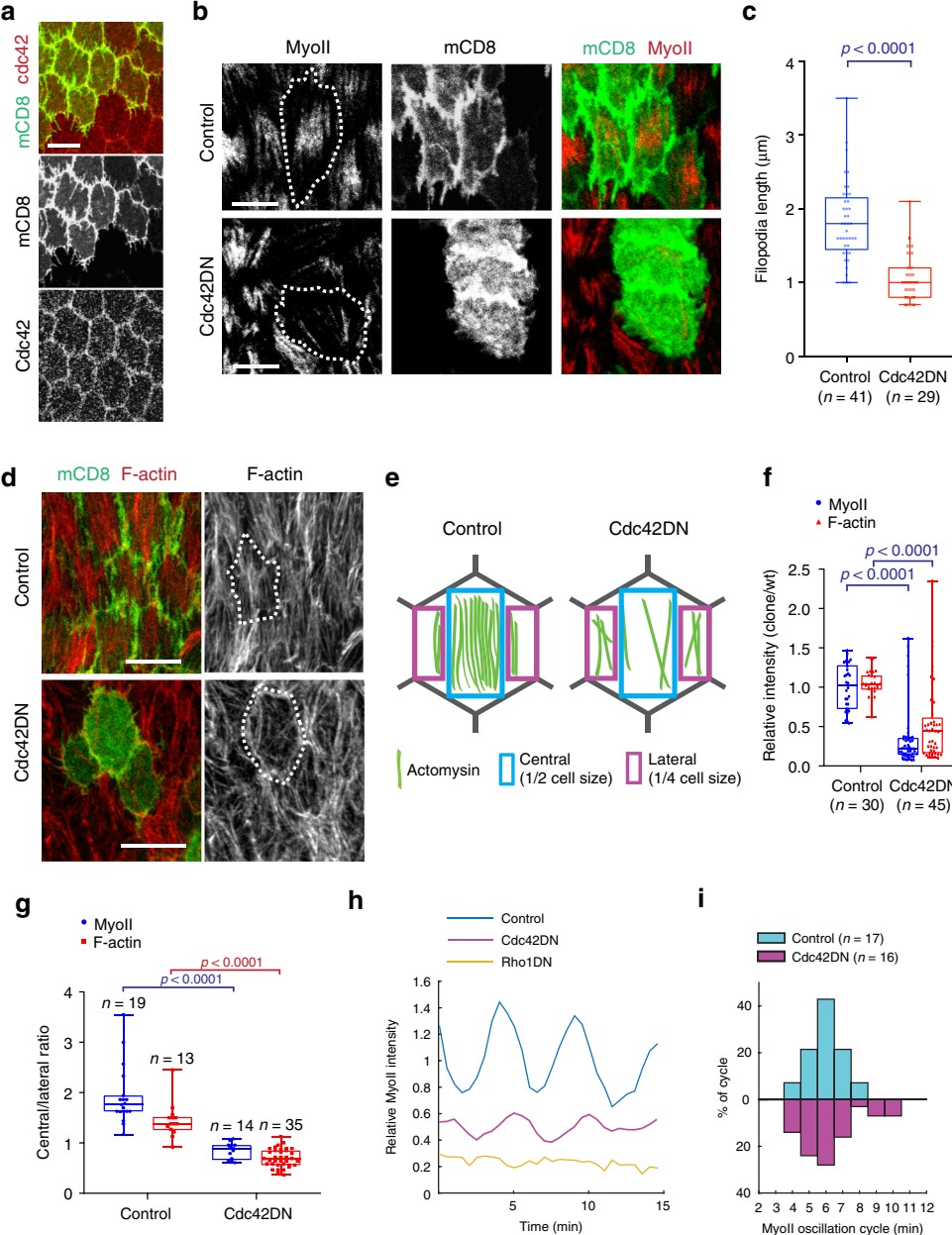

**Fig. 2 Cdc42 activity controls intercellular filopodia and stress fiber formation at the basal domain of follicle cells. a** Basal view of follicle cells at stage 10A (S10A): Cdc42-mCherry (red) and stochastic clone expression of mCD8GFP (green). **b** Basal views of follicle cell clones not expressing (top panels) and expressing (bottom panels) the Cdc42DN transgene marked by mCD8GFP coexpression. MyoII is visualized using a MyoII-mCherry construct (Sqh:: Sqh-mCherry). **c** Average filopodia length per cell for the *n* individual follicle cells not expressing (control, only mCD8GFP-expressing) and expressing Cdc42DN. For each analyzed individual cell, all filopodia not <0.5 µm were measured (3–11 filopodia per cell for control and 2–7 filopodia per cell for Cdc42DN). Nine egg chambers were analyzed for control and five for Cdc42DN. *p* < 0.0001 by two-sided Mann–Whitney test. **d** Basal view of follicle cell clones not expressing (control, only mCD8GFP expressing, top panels) and expressing (bottom panels) the Cdc42DN transgene, marked by mCD8GFP coexpression. F-actin is marked by phalloidin staining. The results shown in **a**, **b**, **d** have been successfully repeated from the at least four independent experiments. **e** Schematic representation of the actomyosin distribution at the basal side of follicle cells in control and Cdc42DN-expressing backgrounds. **f** Relative (clone/wild type) MyoII and F-actin intensities in the *n* individual control (only mCD8GFP expressing) and Cdc42DN cells. *p* < 0.0001 by two-sided Mann–Whitney test. **g** Relative (central/lateral) distribution of MyoII and F-actin intensity in the *n* individual control (only mCD8GFP expressing) and Cdc42DN cells. *p* < 0.0001 by two-sided Mann–Whitney test. In **c**, **f**, **g** boxes extend from the 25th to 75th percentiles, the mid line represents the median and the whiskers indicate the maximum and the minimum values. **h** MyoII-mCherry intensity over time for a representative cell in the control (only mCD8GFP expressing), Cdc42DN and Rho1DN backgrounds. The average MyoII intensity in the control background is set to 1. **i** Percentage of oscillating cycle time periods in the *n* individual control (only mCD8GFP expressing) and Cdc42DN-expressing cells. Scale bars are 10 µm in **a**, **b**, **d**.

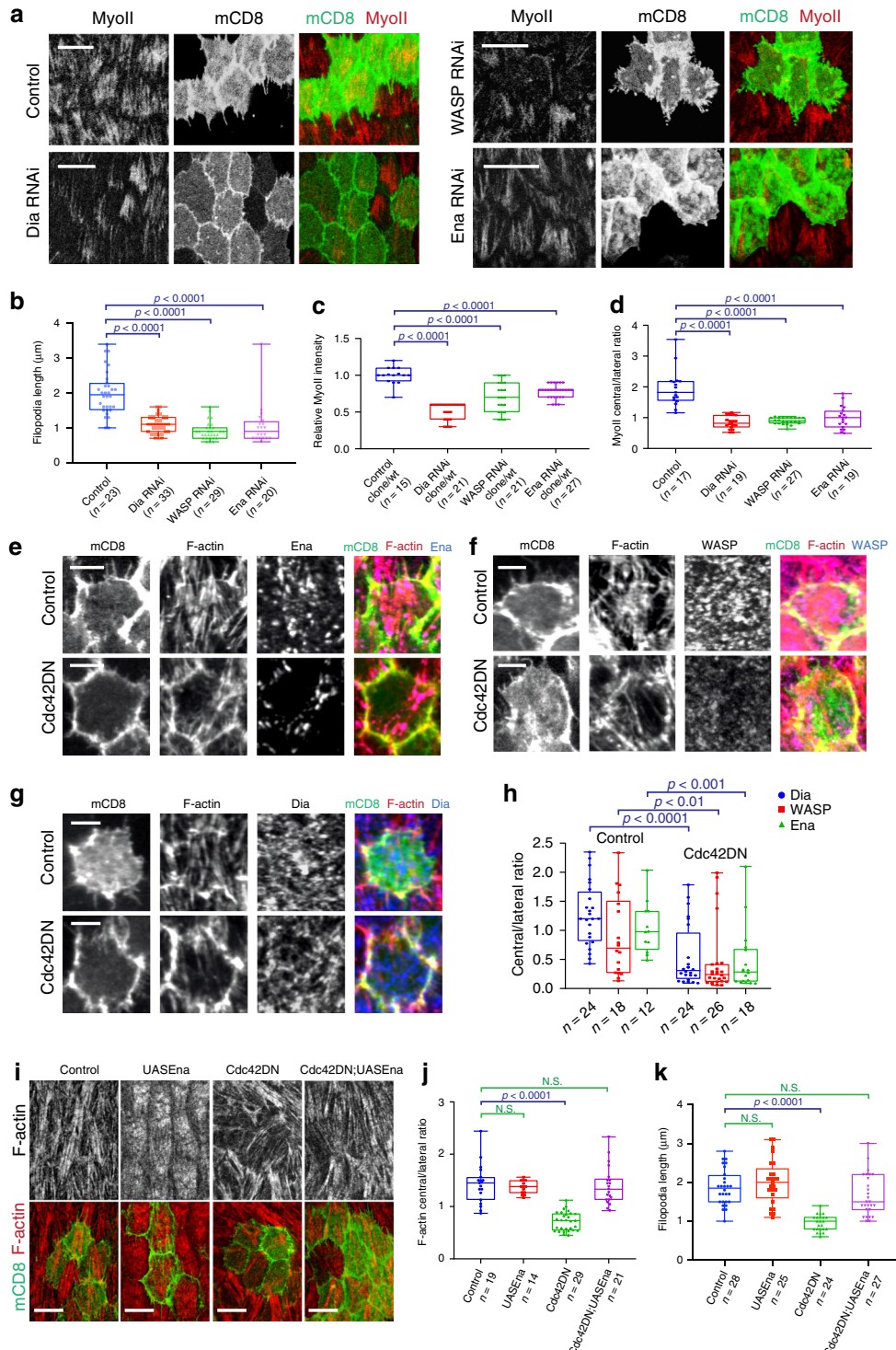

with subcellular precision[38,39] to directly probe tension anisotropy. When ablating the cell medial network along a line that is parallel to the AP direction, a maximum recoil speed of 0.22 μm s$^{-1}$ was measured on average along the DV axis (Fig. 6a, e and Supplementary Movie 8). When ablating along a line that is parallel to the DV direction, a maximum recoil speed of 0.01 μm s$^{-1}$ was measured on average along the AP axis (Fig. 6b, e and Supplementary Movie 9). This result directly shows that tension in the medial network of follicle cells at S9 of *Drosophila* egg chamber development is 20 times higher along the DV axis compared to the AP axis. We next tested the role of Cdc42 in cell tension and the

establishment of tension anisotropy. When performing laser dissection in Cdc42DN clones, the basal-medial network showed little recoil along both the DV and the AP axis and tension anisotropy of a factor 2 (Fig. 6c–e and Supplementary Movie 10). This shows that Cdc42 plays a key role in DV tension and DV/AP tension anisotropy establishment at the cellular scale. We then tested DV tension in wild-type cells that neighbored or were distantly located from Cdc42DN clones respectively. When ablating along a line parallel to the AP direction, little or no recoil was measured for wild-type neighboring cells while a normal maximum recoil speed was maintained in wild type far located

**Fig. 3 Filopodia and stress fibers are under the control of Cdc42 downstream effectors. a** Basal view of follicle cells showing Dia RNAi, WASP RNAi and Enabled RNAi clones marked with mCD8GFP coexpression. MyoII is visualized by using a MyoII-mCherry construct. **b** Average filopodia length per cell for the $n$ individual cells (from at least four egg chambers) in the indicated genetic backgrounds. For all comparisons, $p < 0.0001$ by two-sided Mann–Whitney test. **c** Relative (clone/wild type) MyoII intensity in the $n$ individual cells under the indicated genetic backgrounds. For all comparisons, $p < 0.0001$ by two-sided Mann–Whitney test. **d** Relative (central/lateral) distribution of MyoII intensity in the $n$ individual cells under the indicated genetic backgrounds. For all comparisons, $p < 0.0001$ by two-sided Mann–Whitney test. **e–g** Representative confocal micrographs showing Enabled (**e**), WASP (**f**), and Dia (**g**) at LS9 in a cell not expressing (control, only mCD8GFP expressing) or expressing the Cdc42DN transgene marked by mCD8GFP coexpression. F-actin is marked by phalloidin staining. The results shown in **e–g** have been successfully repeated from the at least four independent experiments. **h** Relative (central/lateral) distribution of downstream effectors in the $n$ individual control (only mCD8GFP expressing) and Cdc42DN-expressing cells. For Dia signal comparison: $p < 0.0001$; for WASP signal comparison: $p = 0.0032$; for Enabled signal comparison: $p = 0.0009$; all by two-sided Mann–Whitney test. **i** Basal view of follicle cell clones expressing the indicated transgenes marked by mCD8GFP coexpression. F-actin is marked by phalloidin staining. **j** Relative (central/lateral) distribution of F-actin intensity in the $n$ individual cells under the indicated genetic backgrounds. For the control vs. UAS-Enabled comparison: $p = 0.6197$; for the control vs. Cdc42DN comparison: $p < 0.0001$; for the control vs. Cdc42DN/UAS-Enabled comparison: $p = 0.6628$; all by two-sided Mann–Whitney test. **k** Average filopodia length per cell for the $n$ individual follicle cells (from at least three egg chambers) in the indicated genetic backgrounds. For the control vs. UAS-Enabled comparison: $p = 0.3191$; for the control vs. Cdc42DN comparison: $p < 0.0001$; for the control vs. Cdc42DN/UAS-Enabled comparison: $p = 0.0841$; all by two-sided Mann–Whitney test. Scale bars are 10 μm in **a**, **i**, and 5 μm in **e–g**. In **b–d**, **h**, **j**, **k** boxes extend from the 25th to 75th percentiles, the mid line represents the median and the whiskers indicate the maximum and the minimum values.

cells (see Fig. 6f–h). This shows that Cdc42 also plays a non-cell autonomous role in cell tension anisotropy. Finally tension was restored by overexpressing Ena in Cdc42DN follicle cells (Supplementary Fig. 10).

**Cdc42 controls supracellular tension and tissue elongation.** We have demonstrated that at S9-S10 follicle stress fibers generate DV-polarized contractile forces at the cellular scale (Fig. 6) while fiber actin structures, resulting from the stress fibers and filopodia intermingling, emerge at the supracellular scale (Fig. 1). We therefore wondered if polarized supracellular forces are acting on the follicular epithelium. To test this, we used IR laser ablation and dissected the basal actomyosin network over a 100 μm line (about ten cells). If forces are only acting at the cellular scale, multiple local recoils of the actomysin network would occur (a recoil for each ablated cell network). If supracellular forces emerge from the interaction of local contractile units, a global recoil at higher speed would be expected. Upon ablation along a line parallel to the AP direction, the basal network recoiled along the DV axis with a maximum speed of 0,76 μm s$^{-1}$ on average. The recoil resulted in one single large opening in the basal network (Fig. 7a, d and Supplementary Movie 11). This demonstrates that a supracellular force is acting at the level of the follicular epithelium and that this is >3 times greater than the contractile force acting at the cellular scale. Ablation along the DV axis resulted in a much slower recoil (0,07 μm s$^{-1}$ maximum speed on average) compared to AP (Fig. 7b, d and Supplementary Movie 12), showing that tension anisotropy is also established at the tissue scale. IR fs laser ablation has been demonstrated to be capable of high-resolved dissection of the actomyosin network with little or no perturbation of the cell membrane[38,40]. Our experiments corroborate this notion: membranes are preserved since they (i) are visible after ablation and (ii) stretch under the action of tissue scale unbalanced forces (Fig. 7a and Supplementary Movie 11). These observations suggest that cell unities are preserved after IR fs tissue scale dissection.

Tissue scale IR fs ablation results in the dissection of the basal network, including both the medial-basal actomyosin stress fibers and the basal junctional F-actin network that is not enriched with Myo-II (Fig. 1j). In order to specifically probe the medial-basal actomyosin stress fibers network (which is the focus of our study), we performed a tissue scale segmented ablation to dissect the actomyosin stress fibers while avoiding the sub-basal cell–cell contact zones (probed already in a previous study[41]). The tissue scale segmented ablation still resulted in one single large opening (see Supplementary Movie 13). In addition, network recoil

maximum speed after segmented dissection was >3-fold greater than after single cell network dissection (Supplementary Fig. 11). These evidences show that egg-chamber surface mechanics, during stage 10, is dominated by polarized supracellular tension generated by the medial-basal actomyosin stress fiber network.

We further tested tissue scale forces in follicular epithelia containing large Cdc42DN clones. Under these conditions, tension was drastically reduced along the DV axis as observed in subcellular dissection experiments (Fig. 7c, d). Therefore, we assessed the effect of Cdc42 inhibition on tissue shape. Egg chambers containing clones of Cdc42 mutant cells were significantly more round-shaped than wild type (Fig. 7e, f). This round egg chamber shape was mainly due to the Cdc42-LOF in follicle cells, since the expression of Cdc42DN specifically in follicle cells resulted in an equally strong phenotype (Fig. 7e, f) despite no apical-basal polarity loss nor any egg chamber rotation defects (Supplementary Fig. 12). These results show that DV-polarized supracellular tension and AP-directed global tissue elongation are under the control of Cdc42.

To further investigate the role of filopodia in establishing a functional supracellular contractile network, we probed filopodia mechanosensitivity and mechanical function. Firstly we performed IR nano-dissection of stress fibers juxtaposed to filopodia: this induced stress fiber tension release and filopodia retraction (Fig. 7g white arrowhead, h). Remarkably, filopodia juxtaposed to preserved stress fibers did not retract (Fig. 7g yellow arrowhead, h). This data shows that filopodia in follicle cells are tension sensitive. We next aimed to test whether filopodia are mechanically coupled to neighboring cell cortexes. To test this, we performed a supracellular laser dissection parallel to the AP direction intersecting a filopodium (Fig. 7i first panel). The hypothesis supposing a strong mechanical coupling between the filopodium and the neighboring cell cortex can be ruled out if, during recoil, the filopodium tip were to move following the displacement of the cell from which it radiates. Remarkably, after dissection, the filopodium tip moved away from the cell that it belonged to: this resulted in filopodium stretching (Fig. 7i and Supplementary Movie 14). After persisting in a stretch configuration for several minutes, the stretched filopodium partially retracted as the tip finally moved back in a way that was similar to a spring that, under load, has been stretched out and then suddenly released from one extremity (Fig. 7i last panel and Supplementary Movie 14). This directly demonstrates that filopodia can be coupled to neighboring cells, thus functioning as mechanical anchors and establishing the bridging links between cell cortexes.

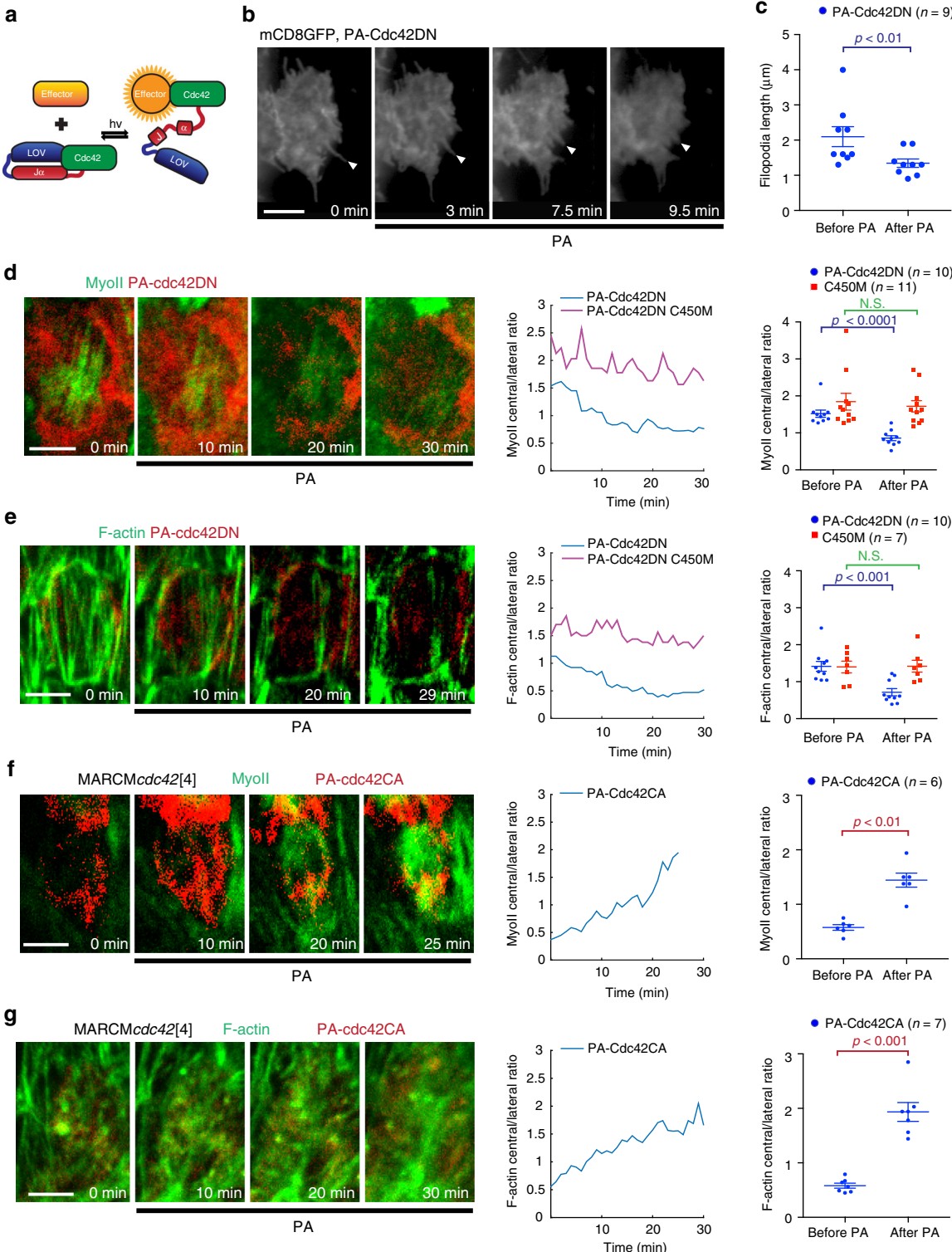

**Discussion.** Previous studies have shown that subcellular actomyosin oscillations play a key role in egg chamber elongation. Here, we extend this by directly probing subcellular mechanics and by showing how cellular forces are integrated at the tissue scale to drive tissue extension. A supracellular and DV-polarized F-actin network emerges from interdigitating filopodia that penetrate actomyosin stress fibers of neighboring cells. Filopodia are formed by F-actin bundles which radiate from the medial-basal network. Both filopodia and actomyosin stress fibers are under the control of Cdc42. The cellular actomyosin stress fibers generate DV-polarized forces. Cellular forces are integrated at the tissue level to generate tissue scale tension anisotropy. Finally, tissue scale forces and epithelia extension are mediated by the supracellular and polarized F-actin network controlled by Cdc42.

In contrast to other epithelial cells, in which Myo-II is enriched at junctions generating forces that are directly transmitted from cell to cell[38,42], in follicle cells Myo-II is not enriched at basal junctions and concentrates instead in the medial region. How can then forces be transmitted from one cell to the other at the basal domain in the follicular epithelium? Here, we show that filopodia,

**Fig. 4 Optogenetics reveal specific spatial and temporal control of Cdc42. a** Schematic diagram showing the mechanism of PA-Cdc42 photoactivation by blue light (*hv*). **b, d, e, f, g**. Time-lapse series of representative mCherry-tagged PA-Cdc42DN-expressing (**b, d, e**) and mCherry-tagged PA-Cdc42CA-expressing follicle cell with *cdc42*[4] mutant genetic background (**f, g**), labeled with mCD8GFP (**b**), MyoII-GFP (**d, f**) and UtrABD-GFP (**e, g**). PA indicates the time of photoactivation. Arrow heads indicate filopodia in **b. c** Average filopodia length per cell (from seven egg chambers) in the *n* individual PA-Cdc42DN clone cells before and 10–15 min after photoactivation. *p* = 0.0099 by two-sided Mann–Whitney test. **d, e, f, g** (central panel). Relative (central/lateral) distribution over time of MyoII (**d, f**) and F-actin (**e, g**) for a representative case in the indicated genetic backgrounds. **d, e, f, g** (right panel). Relative (central/lateral) distribution of MyoII (**d, f**) and F-actin (**e, g**) before and 25–30 min after photoactivation in the *n* individual cells under the indicated genetic backgrounds. For MyoII signal, the PA-Cdc42DN before- vs. after- photoactivation comparison: *p* < 0.0001, the PA-Cdc42DN C450M before- and after- photoactivation comparison: *p* = 0.7107; for F-actin signal, the PA-Cdc42DN before- vs. after- photoactivation comparison: *p* = 0.0005, the PA-Cdc42DN C450M before- and after- photoactivation comparison: *p* = 0.972; for MyoII signal, the PA-Cdc42CA before- vs. after- photoactivation comparison: *p* = 0.0022; for F-actin signal, the PA-Cdc42CA before- vs. after- photoactivation comparison: *p* = 0.0006; all by two-sided Mann–Whitney test. Data are presented as mean values +/− SEM. Scale bars are 5 μm in **b, d, e, f, g**.

bridging basal networks to establish F-actin supracellular patterns, can be mechanosensitive and anchored to neighboring actomyosin cortexes presumably via E-cadherin mediated adherens junctions. Filopodia could thus function as cell–cell coupling units. While previous work has shown that actomyosin stress fiber organization is not affected in integrin downregulated follicular epithelia[34], further work is necessary to rule out the possibility that the extracellular matrix may also function as a scaffold integrating stress fiber contractions during S9.

Our work demonstrates that Cdc42 has a non-cell autonomous role in stress fiber organization. By inhibiting Cdc42, specifically at the sub-basal cell–cell contact zone of PA-Cdc42DN clones, we show that filopodia retract and that stress fibers of neighboring wild-type cells are mis-organized within a time period of tens of minutes. This suggests that filopodia could function also as guiding cues to organize stress fibers. A recent study shows that engulfed cadherin "fingers" are asymmetrically distributed between endothelial cells to maintain actomyosin network polarity during collective cell migration[43]. These findings delineate a functioning mode in which thin protrusions might commonly serve to coordinate the collective behaviors of cell sheets.

During the early stages of *Drosophila* oogenesis, global egg chamber rotation polarizes (1) the extracellular matrix, (2) the intracellular basal F-actin network, and (3) the filopodia (the latter in a unidirectional manner)[9,10,44,45]. When rotation is arrested at early S9, filopodia maintain a DV polarity but become bidirectional penetrating the cortexes of dorsally and ventrally positioned neighboring cells. As egg chamber development proceeds, stress fibers become densely packed and more aligned while filopodia retract beginning at S10B. While during S9 and S10A, bidirectional filopodia might provide a physical cue to guide stress fibers and to stabilize their DV polarity, filopodia might become dispensable in later stages after stress fiber maturation. Filopodia bidirectional polarity might be under the control of the prepolarized extracellular matrix. Further work is necessary to better understand the origin of bidirectionally polarized filopodia.

Finally, we show that actomyosin stress fiber dynamics and organization are separately controlled. While Rho1-ROCK signaling controls basal Myo-II oscillations (as also shown in previous work[11,34,46]), Cdc42 controls stress fiber distribution. While both levels of control are necessary for tissue extension, future work is needed to better understand the role and mechanics of Myo-II periodic oscillations acting at the cell scale and of Myo-II phase waves propagating along the egg chamber.

## Methods

***Drosophila* stocks and genetics.** The following fly stocks were used (information is listed in Supplementary Table 1): *Sqh::RLCmyosinII–GFP, Sqh::RLCmyosinII–mCherry* (from Eric E. Wieschaus)[47], *Sqh::UtrABD–GFP* (from

Thomas Lecuit)[38], E-cadherin-GFP (from Yohanns Bellaiche), UAS-WASP-GFP (from Arno Muller), *UAS-moesinABD-mCherry*[48] (from Brooke M. McCartney), *UAS-Dia^RNAi, UAS-ROCK^RNAi* (from Vienna *Drosophila* RNAi center), *Sqh::Cdc42-mCherry*[49], *Rac1-GFP*[49] (a GFP tag has been inserted at the N-terminal end of the Rac1 coding sequence), *Rho1-mCherry*[49] (a mCherry tag has been inserted at the N-terminal end of the Rho1 coding sequence), *Indy-GFP* (Mi{MIC} GFP knockin), *UAS-Dia, UAS-WASP, UAS-Ena-RFP, UAS-Ena^RNAi, UAS-WASP^RNAi, Sqh::PAK-RBD-GFP, cdc42[4] P{neoFRT}19A/FM6, UAS-mCD8GFP, UAS-mCD8RFP, UAS-Cdc42DN, UAS-Rho1DN, UAS-Rac1DN, UAS-dsRed*, and *CoinFLP-LexA::GAD.GAL4* (from Bloomington *Drosophila* stock center). Clones were generated using the FLP-OUT technique by crossing flies with UAS transgenes to (1) *P{hsp70-flp}; Sqh::RLCmyosinII–mCherry; UAS-mCD8GFP, AyGal4;* (2) *P{hsp70-flp}; +/+; UAS-mCD8GFP, AyGal4;* (3) *P{hsp70-flp}; +/+; UAS-mCD8RFP, AyGal4;* (4) *P{hsp70-flp}; Sqh::RLCmyosinII–GFP; AyGal4;* (5) *P{hsp70-flp}; Sqh::UtrABD–GFP; AyGal4;* (6) *P{hsp70-flp}; +/+; Ay(CD2)Gal4;* (7) *P{hsp70-flp}; UAS-dsRed; Ay(CD2)Gal4.* All stocks and crosses were maintained at room temperature. For signal analysis in mosaic tissues, *hs*FLPase was induced twice, for 1 h at 37 °C, with a 5–h interval, then flies were kept at 18 °C for 2 days and then fattened at 25 °C overnight before dissection. For the analysis of tissue elongation, *P{hsp70-flp}; +/+; Ay(CD2)Gal4* was used for cross and later heat shock treatment, which can induce more than 90% clone cells in the egg chamber. *hs*FLPase of this mosaic system was induced for 1 h at 37 °C once, then flies were kept at 18 °C for 1–2 days and then fattened at 25 °C for overnight before dissection. For follicle rotation, *P{hsp70-flp}; UAS-dsRed; Ay(CD2)Gal4* was used for cross and later heat shock treatment, which can induce more than 90% clone cells in the egg chamber. *hs*FLPase of this mosaic system was induced for 1 h at 37 °C once, then flies were fattened at 25 °C overnight before dissection in order to achieve the moderate expression of Cdc42DN without the disruption of apical/basolateral polarity of follicle cells. For MARCM experiments of the global tissue elongation or the analysis of F-actin/myosin subcellular distribution, flies with the genetic backgrounds (either: *cdc42[4] P{neoFRT}19A/P{hsFLP}1, P{tubP-GAL80}LL1 P{neoFRT}19A; tubGal4 UAS-mCD8GFP* or: *cdc42[4] P{neoFRT}19A/P{hsFLP}1, P{tubP-GAL80}LL1 P{neoFRT}19A; tubGal4 mCD8GFP/Sqh::RLCmyosinII–mCherry)* were used. *P{neoFRT}19A/P{hsFLP}1, P{tubP-GAL80}LL1 P{neoFRT}1A; tubGal4 mCD8GFP* flies were used as a control. *hs*FLPase was induced five times for 1 h at 37 °C over 2 days, flies were kept at 18 °C for 4–5 days and then fattened at 25 °C overnight before dissection. For the optogenetic PA-Cdc42DN experiments, *P{hsp70-flp}; Sqh::RLCmyosinII–GFP; AyGal4 or P{hsp70-flp}; Sqh::UtrABD–GFP; AyGal4 or P{hsp70-flp}; +/+; UAS-mCD8GFP, AyGal4; or P{hsp70-flp}; Sqh::PAK-RBD-GFP; AyGal4* was used for cross with *UAS-PA-Cdc42DN* or *UAS-PA-Cdc42DN C450M* transgenic fly. *hs*FLPase was induced twice for 1 h at 37 °C with a 5–h interval, flies were kept at 18 °C for 1 day and then fattened at 25 °C overnight before dissection. All steps were carried on in dark conditions, including cross, maintenance, and heat shock. For the optogenetic PA-Cdc42CA experiments, *cdc42[4] P{hsFLP}1, P{tubP-GAL80}LL1 P{neoFRT}19 A; UAS-PA-Cdc42CA/actGal4, Sqh::RLCmyosinII–GFP or cdc42[4] P{neoFRT}19A/P{hsFLP}1, P{tubP-GAL80}LL1 P{neoFRT}19A; UAS-PA-Cdc42CA/Sqh::UtrABD–GFP; tubGal4* was used. For the optogenetic experiments of filopodia growth, *cdc42[4] p{neoFRT}19A/P{hsFLP}1, P{tubP-GAL80}LL1 P{neoFRT}19A; tubGAL4, UAS-mCD8GFP* was used for cross with *UAS-PA-Cdc42CA* or *UAS-PA-Cdc42CA C450M* transgenic fly. *hs*FLPase was induced four times for 1 h at 37 °C over 2 days, then flies were kept at 18 °C for 2 days and then fattened at 25 °C overnight before dissection. All steps were carried on in dark conditions, including cross, maintenance, and heat shock, as for PA-Cdc42DN experiments. *Drosophila* ovaries were dissected in weak light conditions, and egg chambers were mounted under red light condition before blue light illumination.

**DNA constructs and transgenic fly generation.** N-terminal-mCherry-tagged PA-Cdc42CA (Q79L/E91H/N92H), PA-Cdc42DN (T17N) and the light-insensitive controlPA-Cdc42CA C450M and PA-Cdc42DN C450M were inserted into the pUASt *Drosophila* expression vector using the Gateway recombination system

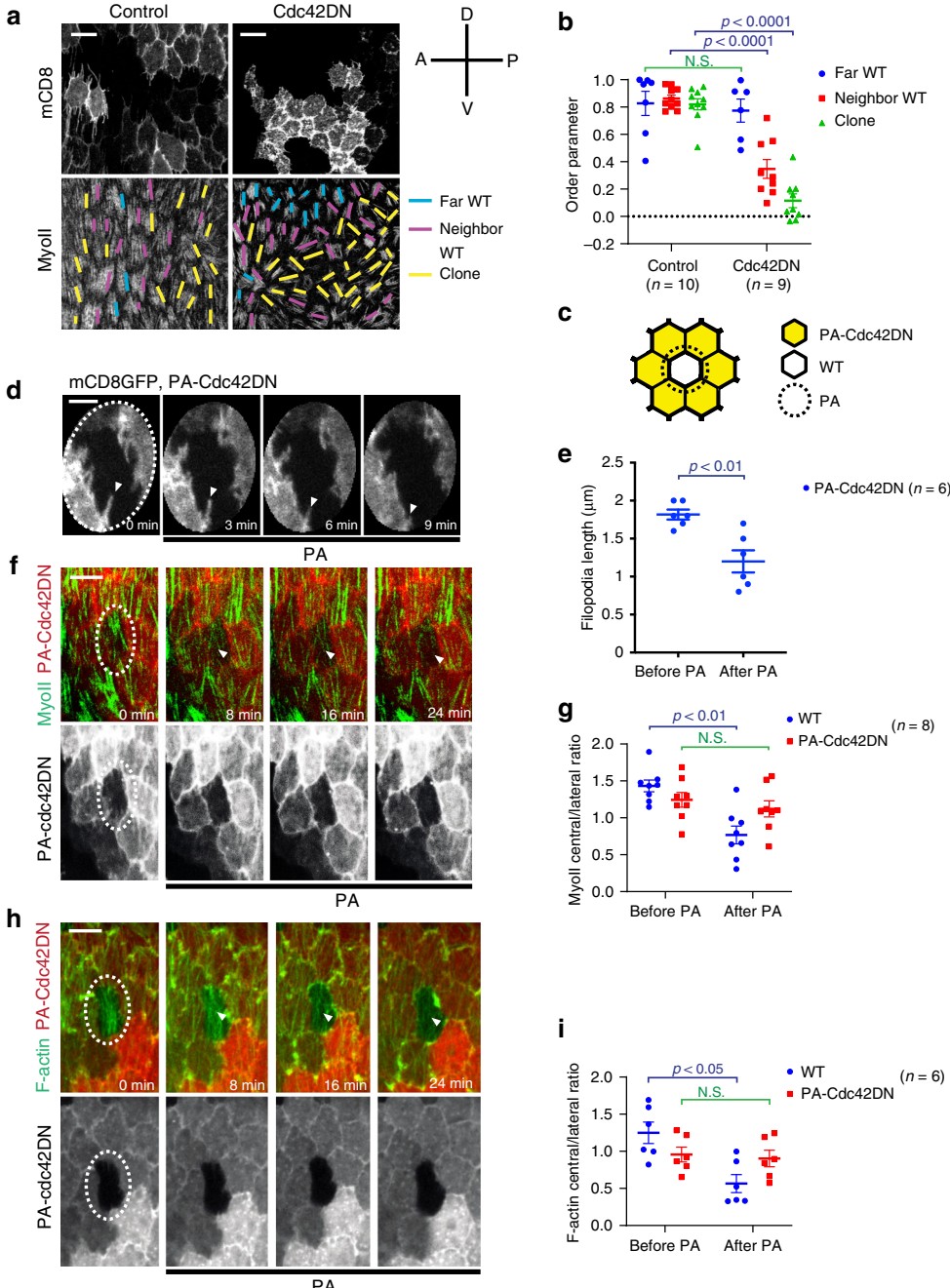

**Fig. 5 Cdc42 has a non-cell autonomous role in stress fiber organization. a** Micrographs showing follicle cell clones not expressing (left panels) and expressing (right panels) the Cdc42DN transgene marked by mCD8GFP coexpression. MyoII is visualized by using a MyoII-mCherry construct. Yellow lines, purple lines and blue lines mark the orientation of the basal stress fibers in the clone cells, in the wild-type cells that neighbor clone cells and in the wild-type cells that are not neighboring the clone cells, respectively. A-P and D-V indicate the anterior-posterior and ventral-dorsal axes. **b** Order parameter for the three types of follicle cells indicated from the $n$ egg chambers not expressing (control, only mCD8GFP expressing) and expressing the Cdc42DN transgene. For the far WT comparison: $p = 0.5338$; for the neighbor WT comparison: $p < 0.0001$; for the clone comparison: $p < 0.0001$; all by two-sided Mann–Whitney test. **c** Schematic diagram showing the region of photoactivation (PA, dashed circle) in experiments where one wild-type cell is surrounded by PA-Cdc42DN-expressing clones. Only the sub-basal cell–cell contact zones of PA-Cdc42DN clones facing the wild-type cells are photoactivated. **d**, **f**, **h** Time-lapse series showing the experiment represented in **c** for PA-Cdc42DN-expressing clonal cells labeled with mCD8GFP (**d**), MyoII-GFP (**f**), and UtrABD-GFP (**h**), respectively. Dashed ellipses indicate the photoactivated region. Arrow heads indicate filopodia in **d** and medial-basal regions in **f**, **h**. **e** Average filopodia length per cell (from four egg chambers) measured before and 10–15 min after photoactivation in the $n$ individual cells of the experiments represented in **c** and **d**. $p = 0.0065$ by two-sided Mann–Whitney test. **g**, **i** Relative (central/lateral) distribution of MyoII (**g**) and F-actin (**i**) in the $n$ individual wild-type and PA-Cdc42DN-expressing cells before and 20–30 min after photoactivation as shown in **c**. For relative distribution of MyoII signal, the WT before- vs. after- photoactivation comparison: $p = 0.0011$, the PA-Cdc42DN before- vs. after- photoactivation comparison: $p = 0.3282$; for relative distribution of F-actin signals, the WT before- vs. after- photoactivation comparison: $p = 0.0108$, the PA-Cdc42DN before- vs. after-photoactivation comparison: $p = 0.6991$; all by two-sided Mann–Whitney test. Data are presented as mean values $+/-$ SEM. Scale bars are 10 μm in **a**, **f**, **h** and 5 μm in **d**.

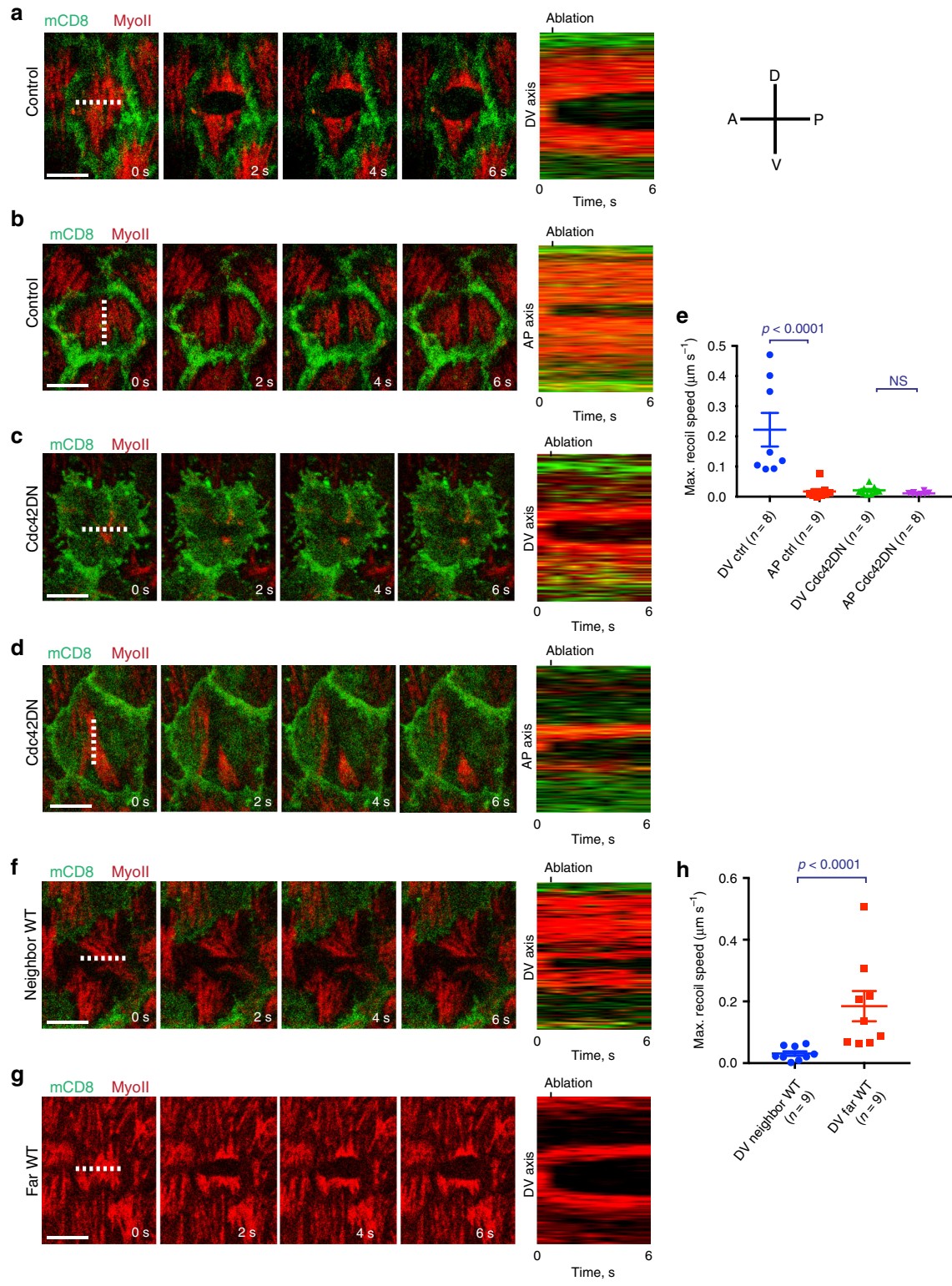

(Invitrogen). The respective primers for the introduction of E91H/N92H, Q79L, and T17N are as follow:

E91H/N92H primers:
Sense: 5′-GGTCTCTCCATCTTCATTTGAGAACGTGAAAGAAAAGTGGG TGCC-3′
Antisense:
5′-GGCACCCACTTTTCTTTCACGTTCTCAAATGAAGATGGAGAGAC C-3′
Q79L primers:
Sense:
5′-GATACTGCAGGGCAAGAGGATTATGACAGATTACGACCGC-3′

Antisense:
5′-GCGGTCGTAATCTGTCATAATCCTCTTGCCCTGCAGTATC-3′
T17N primers:
Sense:
5′-GGGCGATGGTGCTGTTGGTAAAAACTGTCTCCTGATATCCTACA C-3′
Antisense:
5′-GTGTAGGATATCAGGAGACAGTTTTTACCAACAGCACCATCGCC C-3′
The primers for the introduction of C450M in the LOV domain:
PA-Cdc42 C450M Sense:

**Fig. 6 Cortical tension anisotropy is under the control of Cdc42. a–d** Time-lapse series of a representative cell not expressing (**a**, **b**) and expressing (**c**, **d**) the Cdc42DN transgene marked by mCD8GFP coexpression, before (panel on the left) and after ablation (indicated by the dashed line) of the basal actomyosin network along the AP (**a**, **c**) and the DV (**b**, **d**) axes. MyoII is visualized by using a MyoII-mCherry construct. Right panel: kymographs illustrating network recoil along the DV (**a**, **c**) and the AP (**b**, **d**) axes after ablation. The results shown in **a–d** have been successfully repeated from the at least four independent experiments. **e** Maximum recoil speed after ablations in the $n$ individual control (only mCD8GFP expressing) and Cdc42DN clones. For the control DV vs. AP comparison: $p < 0.0001$; for the Cdc42DN DV vs. AP comparison: $p = 0.0788$; both by two-sided Mann–Whitney test. **f**, **g** Time-lapse series of a representative wild-type cell, neighboring (**f**) or distantly located (**g**) from Cdc42DN-expressing clones, before and after ablation (indicated by the dashed line) of the actomyosin network along the AP axis. Right panel: kymographs illustrating network recoil after ablation. The results shown in **f**, **g** have been successfully repeated from the at least four independent experiments. **h** Maximum recoil speed after ablations in the $n$ individual wild-type cells that are neighboring or distantly located from Cdc42DN-expressing clones. $p < 0.0001$ by two-sided Mann–Whitney test. Data are presented as mean values $+/-$ SEM. Scale bars are 5 μm in **a**, **b**, **c**, **d**, **f**, **g**.

5′-CCGTGAAGAAATTTTGGGAAGAAACATGAGGTTTCTACAAGGTCC TGAAACT-3′

PA-Cdc42 C450M Antisense:

5′-AGTTTCAGGACCTTGTAGAAACCTCATGTTTCTTCCCAAAATTTC TTCACGG-3′

The gateway primers to insert all these 3 PA-Cdc42 constructs from mammalian vector to pUASt *Drosophila* expression vector are as following:

Sense:

5′-GGGGACAAGTTTGTACAAAAAAGCAGGCTTCACCGATCC GAAATTTCTGCTCC-3′

Antisense:

5′-GGGGACCACTTTGTACAAGAAAGCTGGGTTTTATTCATAGCAGCA CACACCTGCG-3′

*UAS-PA-Cdc42CA* transgenic flies were generated by the Bestgene using the w1118 fly. Moreover, *UAS-PA-Cdc42DN* and *UAS-PA-Cdc42DN* with LOV C450M mutant version transgenic flies were generated by Centro de Biologia Molecular Severo Ochoa (CSIC/UAM) using the w1118 fly.

**Dissection and mounting of the *Drosophila* egg chamber.** One- to three-day-old females were fattened on yeast with males for 1–2 days before dissection. *Drosophila* egg chambers were dissected and mounted in live-imaging medium (Invitrogen Schneider's insect medium with 20% FBS and with a final PH adjusted to 6.9), using a similar version of the protocol described in ref. [50]. In contrast to the normal mounting conditions, egg chambers were slightly compressed to overcome the endogenous curvature. Under this condition, basal oscillation pattern, intensity and period were similar to those observed under conditions without compression.

**GST pulldown.** The assay of Cdc42 activity in cell lysates was performed as described in ref. [51]. Briefly, the p21-binding domain (PBD) domain from PAK1 was subcloned into pGEX-4T1 vector and expressed as an N-terminal GST fusion protein in BL21(DE3) competent *E. coli* (New England BioLabs). GST-PDB was purified from bacterial lysates using glutathione magnetic beads (Thermo-Scientific). The PA-Cdc42 lit state mutant (I531E/I539E) or dark-state mutant (C450A, L514K, G528A, L531E, N538E) were expressed in HEK293 cells via transient transfection with Fugene6 (Promega). HEK293 cell line was from Genetica, MA, USA, and it was not authenticated. This cell line has been regularly tested negative for Mycoplasma contamination. Lysates from HEK293 cells were incubated with ~10 μg of GST-PDB beads and incubated at 4 °C for 1 h. Following incubation, beads were washed, boiled with sample buffer, and analyzed via western blot.

**Imaging and photomanipulation.** Time-lapse imaging was performed with a Zeiss LSM710 or Leica SP8 confocal microscope with a 40×, numerical aperture 1.3 inverted oil lens, with a 488 nm argon laser and a 561 nm green laser in LSM710 or a 552 nm laser in SP8. The basal focal plane, which is about 1 μm beneath the basal surface inside the cell, was selected during live imaging to maximize the basal Myo-II intensity. For the dynamics of UtrABD–GFP, the similar basal focal plane was selected to maximize the basal F-actin intensity, as for basal Myo-II dynamic imaging. The same microscope setup was used when comparing intensity between different samples. For the dynamics of follicle cell rotational movement, the focal plane, which is centered at follicle cell nuclei, was selected to maximize the nuclear dsRed to better view the localization of individual follicle cells. Imaging data have been collected by Zeiss Zen software (version: Zen 2007 light edition) or Leica Metamorph software (version: Metamorph 7.8.13.0) or Leica SP8 LAS software (version: LAS X).

For photoactivation experiment, live-cell imaging was performed using a Zeiss LSM710 confocal microscope with a 40×, numerical aperture 1.3 inverted oil lens, with a 488 nm argon laser and a 561 nm green laser. To photoactivate, the 458 nm laser was set at 6–8% power for 0.1 ms per pixel in a 5–15 μm circle or rectangle and the photoactivation scan took ~15–20 s. After 30 s, follicle cells were imaged using 488 or 568 nm. This series of steps was repeated for the duration of the time-lapse experiment.

**Drug treatments.** Egg chambers were dissected in live-imaging medium, and then incubated with Cdc42 inhibitor ML141[52] (Sigma) at 400 μM, Rac inhibitor NSC23766[53] (Sigma) at 400 μM, Rho1 inhibitor Rhosin[54] (Merk) at 400 μM for 20 min before being mounted for imaging. Information of chemical inhibitors is listed in Supplementary Table 2.

**Image processing and data analysis.** Images were processed with MATLAB (version: R2018a) and Image J (version: 1.51j8). For all images the background (intensity of area without sample) was subtracted.

Image J was used to calculate the intensity of an individual cell as the average value of all pixels within the cell area. In the time-lapse experiments, images were processed by MATLAB to correct photo-bleaching automatically. To determine the signal ratio between the medio-basal region and the outside region close to basal junction, each follicle cell was separated into these two regions (one is medio-basal cortex, which is around 1/2 at center region, while the other is around 1/4 regions near both anterior and posterior junctional membrane), and then the signals in either region were analyzed using ImageJ.

For the quantification of egg chamber rotation, the rotation speed of follicle cells was measured from the time-lapse images of S6-S7 egg chambers expressing nuclear dsRed driven by FLP-OUT system. The time-lapse positions of individual follicle cell nuclear centers were automatically tracked by MATLAB and the migration speed was automatically calculated also by MATLAB. Migration speed (each dot in Supplementary Fig. 12c) was obtained from the average of each follicle cell nuclear movement in the same egg chamber.

The distribution of oscillation periods was generated by measuring the intervals between each pair of adjacent peaks. We applied autocorrelation to calculate the period of a time series with different time offsets. This method averages out irregularities in the sequence and gives a similar average period. To quantify the percentage distribution of the oscillating time period, the 25 to 30 minute-dynamic intensity of the $n$ individual cells ($n$ is indicated in Fig. 2i) from four independent LS9 egg chambers were tracked, then the oscillating cycle time of each individual cell was calculated by the autocorrelation method.

Tissue elongation was measured by the A-P to D-V length ratio of S10 and S14 egg chambers.

For the quantification of the planar cell polarity of stress fibers in follicle cells, we calculated the order parameter, S, for the mutant clone, neighboring WT and far WT follicle cells (three categories of follicle cells) in egg chambers, by a similar version of the protocol described in ref. [9]. We defined S as: $S = 2(<cos^2(\theta_{ij})> - 1/2)$, where $\theta_{ij}$ is the angle between the directors of cells $i$ and $j$. The average was performed over all unique cell pair combinations in each category of follicle cells.

Box and whiskers plots (GraphPad Prism software) were used to represent the filopodia length and distribution as well as stress fiber orientation, and also the data with sample number >10: boxes extend from the 25th to 75th percentiles, the mid line represents the median and the whiskers indicate the maximum and the minimum values. GraphPad Prism software (version: 8.0.1) has been used for box and whiskers plots.

For filopodia length analysis, the average filopodia length per cell was quantified. For each analyzed individual cell, all filopodia not <0.5 μm were measured.

**Ablation experiments.** Ablations of basal actomyosin stress fibers were performed using a 780 NLO (Zeiss) inverted confocal microscope, equipped with the Spectra-Physics Mai Tai DeepSee IR fs laser (Newport Corp., Irvine, CA, UAS), using 40×/ N.A. 1.2 water-immersion objective (ZeissC-Apo 421767-9970). Ablations were obtained by exposing the fibers to the focused beam with an average power of 200 mW (at 950 nm with 90% transmission) measured on the focal plane of the objective using a thermic slide-photoreceptor. IR fs laser dissections were performed using FRAP function of ZEN software. The region of interest (ROI) was always set to a horizontal line (the egg chamber was thus digitally rotated accordingly for AP or DV ablations). The frame rate is constant regardless of the length of the ROI. Images were acquired in mono-photon mode, using an Argon laser (488 nm) and DPSS 561 nm.

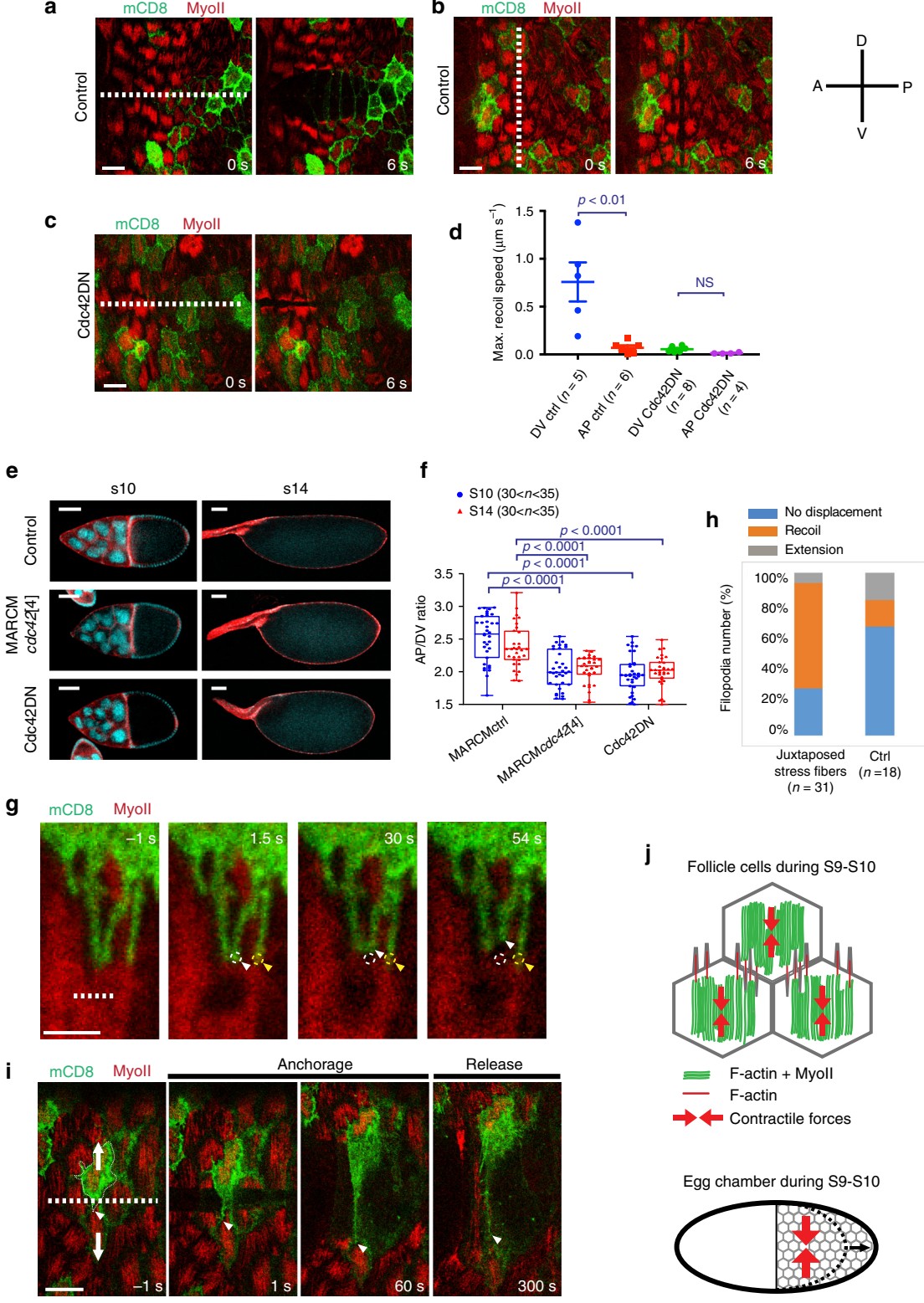

Measurements of the maximum recoil speed were performed with Fiji software, using the particle image velocimetry (PIV) plugin[55]. For each ablation event, the velocity vector field was determined by PIV between the pre-cut frame and an image 2 s post-cut. The maximum recoil speed was estimated from the velocity vector field as for the average velocity component orthogonal to the cut line.

**Immunohistochemistry.** *Drosophila* ovaries were dissected in Schneider's medium and fixed with 4% paraformaldehyde in PBS (phosphate-buffered saline) for 20 min. After fixation, the egg chambers were rinsed with PBST (PBS with 0.3% Triton X-100) three times. The egg chambers were incubated with various primary antibodies, usually overnight at 4 °C. The anti-WASP antibody (P5E1, 1:200 dilution; validation information available at https://dshb.biology.uiowa.edu/P5E1-Wasp), anti-Ena antibody (5G2, 1:200 dilution; validation information available at https://dshb.biology.uiowa.edu/5G2-anti-enabled), anti-Dlg (4F3, 1:200 dilution; validation information available at https://dshb.biology.uiowa.edu/4F3-anti-discs-large), and anti-Arm (N27A1, 1:50 dilution; validation information available at https://dshb.biology.uiowa.edu/N2-7A1-Armadillo) were from the Developmental Studies Hybridoma Bank. The anti-aPKC (sc-216, 1:500 dilution; validation information available at https://www.scbt.com/fr/p/pkc-zeta-antibody-c-20) was from Santa Cruz biotechnology. Anti-Dia antibody (1:5000) was a gift from the

**Fig. 7 Cdc42 controls DV-polarized supracellular tension and AP-directed global tissue elongation. a–c** Time-lapse series of a representative egg chamber not expressing (**a**, **b**) and expressing (**c**) the Cdc42DN transgene marked by mCD8GFP coexpression before and after ablation (indicated by the dashed line) of the basal actomyosin network over a 100 μm line along the AP (**a**, **c**) and the DV (**b**) axes. MyoII is visualized by using a MyoII-mCherry construct. **d** Maximum recoil speed after ablations in the *n* individual control (only mCD8GFP expressing) and Cdc42DN-expressing egg chambers. Data are presented as mean values +/− SEM. For the control DV vs. AP comparison: $p = 0.0043$; for the Cdc42DN DV vs. AP comparison: $p = 0.1157$; both by two-sided Mann–Whitney test. **e** Egg chamber morphology under the indicated genetic backgrounds at S10 and S14. Armadillo staining in red and DAPI staining in blue. **f** Anterior-posterior (AP) to dorsal-ventral (DV) length ratio in the *n* individual egg chambers under the indicated genetic backgrounds at S10 and S14. Boxes extend from the 25th to 75th percentiles, the mid line represents the median and the whiskers indicate the maximum and the minimum values. For all comparisons, $p < 0.0001$ by two-sided Mann–Whitney test. **g** Time-lapse images showing the region of contact of one mCD8GFP expressing cell with a wild-type cell before and after nano-ablating (indicated by the dashed line) stress fibers juxtaposed to filopodia (white arrowhead). Circles indicate the initial positions of the filopodia tips juxtaposed to severed (white) or preserved (yellow) stress fibers. Arrow heads indicate the current position of the filopodia. **h** Percentage of filopodia undergoing either no displacement, recoil or extension during the first minute after ablation. **i** Time-lapse series of a representative egg chamber, before and after ablation (indicated by the dashed line) of the actomyosin network along the AP axis and across a filopodium. The arrowhead indicates the current position of the filopodium tip. Arrows indicate the actomyosin recoil directions. **j** Schematic representation of follicle cells and the subcellular distribution of stress fibers and intercellular filopodia forming a supracellular contractile network (top panel) surrounding the egg chamber and generating tissue scale forces driving AP tissue elongation (bottom panel). Scale bars are 10 μm in **a–c**, **i**, 50 μm in **e**, and 2 μm in **g**.

laboratory of Steve Wasserman at University of California, San Diego (validation has been done in our previous study[34]). Secondary antibodies conjugated with Alex-561 and Alexa-647 (Molecular Probes, A21244, A21235, A21428, and A21422) were used in 1:400 dilutions. Alexa-561-conjugated phalloidin (1:200 dilution; Invitrogen, A34055) was used for F-actin staining. Information of all these materials is listed in Supplementary Table 2. Samples were imaged on a Zeiss LSM710 or Leica SP8 confocal microscope.

**Statistics**. All data are presented as mean ± SEM. Statistical analysis to compare results among groups was carried out by the Mann–Whitney test (GraphPad Prism software). A value of $p < 0.05$ was considered to be statistically significant.

**Reporting summary**. Further information on research design is available in the Nature Research Reporting Summary linked to this article.

## Data availability
The data sets generated during and/or analyzed during the current study are available from the corresponding author on request. The source data underlying Figs. 2c, f, g, 3b–d, h, j, k, 4c–g, 5b, e, g, i, 6e, h, 7d and f, Supplementary Figs. 3d, f, g, i, j, 4a, b, 5b, c, 6c, e, 7b, d, e, g, 8b, d, e, 9c, e, g, 10d, 11, and 12c are provided as a Source Data file.

## Code availability
The codes used for analyses of different images are available from the corresponding author on request.

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

## Acknowledgements

We thank Adam Martin, Thomas Lecuit, Brooke M. McCartney,Yohanns Bellaiche, Arno Muller, Bloomington *Drosophila* stock center and Vienna *Drosophila* RNAi center for flies. We thank Steve Wasserman for the Dia antibody. We thank the CBI Toulouse imaging facility at the Université Paul Sabatier and also the iBV PRISM imaging facility at the Université Côte d'Azur. We thank the *Drosophila* facility at both CBI Toulouse and iBV Nice. We thank Francois Schweisguth and Jiong Chen for discussion of manuscript preparation. We thank Abby Cuttriss for feedback on the paper. This work was supported by the Institut National de la Santé et de la Recherche Médicale [ATIP-Avenir program (2012–2016)], Région Midi-Pyrénées Excellence program (2013–2016) and Scientifiques de la Fondation ARC (grant number PJA 20171206526, PJA20191209714), all to XW; the French government through the UCA^JEDI Investments in the Future project managed by the National Research Agency (ANR-15-IDEX-01), the "Investments for the Future" LABEX SIGNALIFE (ANR-11-LABX-0028-01), the Tramplin ERC program from the National Research Agency (ANR-16-TERC-0018-01), the ATIP-Avenir program from the CNRS and the Human Frontier Science Program (CDA00027/2017-C), all to M.R.; the National Institutes of Health (GM-R35GM122596) to K.M.H.; and the National Institutes of Health (grant GM46425) to D.J.M.

## Author contributions

A.P., K.M.H., X.W., and M.R. designed the project and the experiments. A.P., L.C., X.Q., and C.L. performed image acquisition and transgene analysis. A.P., L.C., X.Q., and J.L. processed and analyzed images. M.R. performed the Fourier analysis. O.J.S. conducted GST-pulldown experiment. A.P. made the constructs for transgenic flies. A.P., K.B., D.J.M., X.W., and M.R. prepared the manuscript. All authors participated in the interpretation of the data and the production of the final manuscript.

## Competing interests

The authors declare no competing interests.
