## [Peer Review File · Nature Communications]

Reviewers' comments:

Reviewer #1 (Remarks to the Author):

Nature Communication NCOMMS-17-08771-T

Cdc42-mediated supracellular cyoskeleton governs basal myosin contractility and tissue elongation by Popkova et al.

The egg chambers of *Drosophila* ovaries have recently developed into a model for studying the cellular mechanisms that cause tissue elongation. Work from other groups provided evidence that the follicular epithelium undergoes collective cell migration from early to mid oogenesis with the help of basal protrusions, which leads to follicle rotation that is thought to drive tissue elongation. Afterwards, oscillations of the basal actin-myosin network of follicle cells have been shown to further promote follicle elongation.

Here the authors uncover a novel mechanism that contributes to tissue elongation at later stages of oogenesis. They find that Cdc42 is crucial for the generation of cellular protrusions that enable follicle cells to interdigitate at the level of the basal stress fiber-like actin-myosin network, preserving this stress fiber network and coordinating its planar-polarized orientation across the follicular epithelium.

The main message of the manuscript that implicates Cdc42 as an important regulator of tissue-wide coordination of a basal actin-myosin contractile network through interdigitation of follicle cells by cellular protrusions is novel and well supported through a wide range of experiments using cutting edge techniques, including an optogenetic tool that allows a quick and spatially controlled inactivation/up-regulation of Cdc42 activity, which they newly generated. The effect of Cdc42 on the contractility of the stress fibers was not measured directly but indirectly addressed by using E-cadherin as a tension sensor. The quantitative analysis is extensive and robust. All images in Figures and Supp. figures are excellent.

This study will be of interest to a wide audience. It will be interesting in the future to determine how similar the follicle cell protrusions are to the recently identified cadherin fingers in collectively migrating endothelial cells, and whether E-cadherin-mediated cell-cell interactions are necessary for intercellular filopodia and stress fiber maintenance and/or formation.

Critique:

1) The authors call the protrusions 'intercellular filopodia'. However, they do not provide any evidence that would show that these bidirectional protrusions fit the definition of filopodia. For example, the Ena staining of Figure 2g does not show localization at the tips of the protrusions as one might expect from filopodia. The authors need to provide convincing evidence for their claim that these are filopodia or alternatively, call them simply 'cellular protrusions'. It is also not clear whether and how these protrusions are related to the filopodia at earlier follicle stages.

2) The Abstract of the manuscript makes little sense as it lacks any context for the described results. It is essential to mention that the protrusions are made by epithelial cells, that the coordination of the actin-myosin contractile network takes place within an epithelium, and it is essential to name *Drosophila* follicles as the studied system (e.g DV polarity makes no sense without this context).

3) The authors did not properly describe the location of the protrusions in relation to the apical-basal organization of the follicle cells at the beginning of the paper. So, the reader is left to speculate what is meant by 'protrusions between cells' until much later in the manuscript. Adding a schematic drawing, showing two follicle cells in a side view and indicating the location of the protrusions and stress fibers would help to orient the reader.

4) There are two apparently contradictory statements in the text: "We noticed the lengths of intercellular filopodia correlated with the amplitude of basal Myo-II oscillation" (lines 60-61) and "... we didn't detect a prominent signal correlation between filopodia length and basal Myo-II intensity from filopodia-producing or invaded-cells" (lines 132-134). Please, clarify.

5) Although there is a negative correlation between myosin II levels in the vicinity of filopodia and the FRET ratio signal indicated by EcadTS (Fig.4A,B), it is apparent that intercellular filopodia do not sustain more tension than the rest of the cell cortex (Fig. 4A) as it would be expected if these structures were major anchoring points for stress fibers. Are these correlations similar in regions without filopodia? How do the authors comment on this?

6) Drawing schematics in Fig. 4I and K would help the reader understanding these experiments more easily. In addition, is there leakiness in PA-Cdc42 DN expression? In Fig.4J, the levels of medial myosin intensity seem reduced in PA-Cdc42 DN expressing cells compared to wild type, even before photoactivation.

7) In Fig.4E, are the n values presented the number of filopodia or the number of cells analysed ? If filopodia, they are very low and need to be increased for statistical tests to be appropriately used.

8) Supplementary figure 5e,f: Visually, the FRET data between the two encircled areas for the EcadTS-C +ML141 image look very different and I am surprised that this shows as non-significant. Please, comment.

9) In the Materials and Methods, the statement starting at line 493 “We found that autocorrelation...” is very long (spans throughout 6 lines) and needs to be rewritten for sake of clarity.

10) In Figure legend 1M, the statement “Error bars...” (lines 581-582) doesn’t belong here, since there are no error bars in the figure.

11) p 7, line 139: basal junction -> basal junctions

12) p 10 lines 211-212: this sentence needs to be revised as it currently ignores that there are two plasma membranes in between.

Reviewer #2 (Remarks to the Author):

In this manuscript, Popkova and colleagues investigated the function of filopodia in follicle cells of *Drosophila* egg chamber. By using live imaging, quantitative image analyses, genetics, including opto-genetics, the authors claimed that 1) Cdc42 regulate the formation and distribution of the “intercellular filopodia” and the stress fibers, and 2) the “intercellular filopodia” further regulate the distribution of the stress fibers in the neighboring cells. The results using PA-Cdc42DN (and CA) are interesting and points authors tried to claim are attractive. However, the data and the interpretations presented in this manuscript did not support their claims. As a result, I felt that many of the claims are overstatement and/or over-interpretation. I have suggestions and comments that could improve further presentation of the results.

Major comments

#1. It is not clear to me which part of the egg chamber the authors are imaging. Please clarify with a schematic. This also leads to my other question: Is there a spatial inhomogeneity in cell shape and myosin intensity among the monolayer of follicular epithelium? Figs 5c-d show that the cells with

higher myosin intensity have more elongated cell shape compared to the cells with lower myosin intensity. Please clarify.

#2. The authors used the results from E-cad FRET force sensor to interpret the membrane tension. The force measured with E-cad FRET force sensor represent the tension exerted within E-cad, which is distinct from the membrane tension. The authors seem to miss-interpret the FRET signal of E-cad force sensor. This confusion needs to be addressed.

#3. In Fig. 4a-b, although the authors highlighted that changes in tension at the intercellular filopodia region (within the ROIs), the images shows that the tension outside of the intercellular filopodia regions also correlated with the MyoII intensity within the cells, suggesting the change in tension is not restricted to the region related to intercellular filopodia. How the authors reconcile this? Please explain. This makes the author's argument "intercellular filopodia might function as biomechanical sensors" unconvincing.

#4. Based on the drug treatments (ionomycin and Y27632) to control the actomyosin contractility, the authors claimed that "stable actomyosin contractility is able to control the growth of intercellular filopodia". It is not clear to me what "stable actomyosin contractility" means in the context of wild-type egg chamber. Please clarify. In wild-type, the actomyosin contraction fluctuate in time, i.e., not stable. The other related question is whether the myosin density fluctuation diminished in these treatments and it looks like "stable" contractility. Please explain.

#5. The authors claimed that "Light-induced inhibition of Cdc42 at the filopodia region resulted ..., but not in the light-treated PA-Cdc42DN cells themselves (Fig. 4i-l)" without showing the images of the light-treated PA-Cdc42DN cells. The authors need to provide the images that include the neighboring non- light-treated cells. Because of this lack of evidence, it is overstate to claim that "intercellular filopodia might maintain the medio-basal localization of stress fibers in their invaded cells".

#6. The downstream effectors of Cdc42, WASP, Ena, and Dia were claimed to be "distributed within the F-actin regions of both stress fibers and filopodia" (Supp. Figs. 3c-e). The staining signal of these effectors were everywhere without clear distributions, which made these images and claims unconvincing.

#7. The authors claimed that "This supracellular network functions to maintain the medio-basal distribution and the global DV polarity of Myo-II contractility (Fig. 5j)". It has been shown that the global DV polarity of contraction is governed by the direction of ECM. Thus I felt that this is an overstatement and creates confusions. The authors need to thoroughly and fairly discuss on the global DV polarity.

#8. In the model (Fig. 5j), the authors claimed that the “Intercellular filopodia” link to stress fibers through F-actin (red in the figure). To claim this link, the authors provided Supp. Fig. 3a-b. The actin rich structure that the authors highlighted in Supp. Fig. 3a-b is shorter than the “Intercellular filopodia” shown in the other figures (for instance Fig. 1b-c and Suppl. Fig 1f-g). It looks like this actin rich structure (in Supp. Fig. 3a-b) was not located among the actomyosin meshwork in the neighboring cell, suggesting this fiber is a part of stress fibers. The authors used the images of membrane marker (mCD8) and presented the “Intercellular filopodia”. However, there are no clear images of actin fiber representing the F-actin shown in Fig. 5j (red in the figure). Thus I felt this model is overstating their findings. The authors need to show more clear actin data and/or clarify this.

Minor comments

#a. “data not shown” should not be used. Please provide the data.

#b. In Fig. 4a-b, the authors claimed that “the tension ... at the intercellular filopodia was correlated with the intensity of local Myo-II at the region where a filopodium sticks”. How the local MyoII intensity was measured? How the ROIs were set? Please explain.

#c. In PA-Cdc42CA/DN experiments (Fig 3 &4), how the “% of medio-basal F-actin (or MyoII) intensity” was measured? Precise cell shape information is required to measure this value. However, from the images presented, I could not see the clear cell shape. The authors need to 1) clarify how the cell shape measurement was done, and 2) show the images showed clear cell shape.

#d. The authors claimed that “the morphogenetic change mediated by Cdc42 inhibition was not related to either apical-basolateral polarity or the global tissue rotation occurring from S5 to S8 (Supplementary Fig. 7)”. It is hard to understand this without providing the complement data sets from control egg chamber. Please provide these data.

#e. It was quite a hustle for me to read through the manuscript mainly because so many critical data were shown in the supplement and I need to go back and forth between main and supplemental figures.

Reviewer #3 (Remarks to the Author):

Popkova et al. use a combination of genetic, pharmacologic, and optogenetic perturbations to examine the role of intercellular filopodia of follicle cells in organizing dorsal-ventral polarization of stress fibers and myosin contraction to drive *Drosophila* egg chamber elongation. They find that Cdc42 activity is required for proper intercellular filopodia formation and that these filopodia help to enforce uniform tissue-level orientation of stress fibers and myosin. While this work presents a number of intriguing observations, there are some issues that need to be addressed.

Major concerns:

1. Sufficient validation of the optogenetic tools for controlling Cdc42 activity is lacking. For the data shown in Fig S4A, the 'lit' mutant of PA-Cdc42 appears to bind PBD with higher affinity than the 'dark', but how do these apparent differences in affinity correspond to the ability of light to control PA-Cdc42 activity in vivo? Additional experiments should be performed to assess the extent to which PA-Cdc42 can modulate the activity of its direct effectors in cells. For example, in ref #25, the authors showed that a similarly-designed light-controllable Rac could modulate PAK localization/activity in a light-dependent manner. Such experiments are critical for determining the extent to which PA-Cdc42 activity is enhanced in the 'lit' vs. 'dark' state. F-actin/myosin organization and other readouts used in this study are too many steps removed from Cdc42 activity to reliably make such assessments.

Along similar lines, does expression of PA-Cdc42DN produce any defects similar to those of Cdc42DN cells independent of light-based activation? Experiments should be performed as in Fig 11 where F-actin and Myo-II levels are compared among 'control', 'Cdc42DN', and 'PA-Cdc42DN' cells kept in the 'off' (dark) state. As PA-Cdc42DN in particular is used to generate some of the most compelling observations in this study (e.g., that disrupting intercellular filopodia produces defects in F-actin/myosin regulation in adjacent cells) it is essential to ensure that these optogenetic tools for control of Cdc42 activity are adequately vetted.

2. The data linking Ena, Dia, and WASP to Cdc42 activity is somewhat thin. While the authors provide strong evidence that disruption of these factors phenocopies loss of Cdc42, little is done to demonstrate that these factors function downstream of Cdc42 to promote F-actin/myosin organization and intercellular filopodia growth. There is one instance where expression of Ena is shown to rescue defects in F-actin organization in Cdc42DN cells (Fig. 2L-M); however, similar experiments were not attempted with either WASP or Dia. Does either WASP or Dia expression rescue these defects in Cdc42DN cells? Alternatively, does the further disruption of Cdc42 in either WASP or Dia knockdown cells result in compounded defects in F-actin/myosin organization or filopodia morphology?

Minor concerns/comments:

1. The experiments depicted in Fig 4I-L should be repeated with the PA-Cdc42DN C450M control.

2. In the figures and movies depicting the optogenetic experiments, it would be helpful to indicate where activating light is being applied.
3. The arrow depicting activating light in Fig 3A appears to be inverted.
4. Are the data represented by the controls in Fig 2B-C identical to the controls in Fig 1F and 1I? If so, this should be made explicit in the text or figure legend.
5. What is the time interval between 'before PA' and 'after PA' in Fig 3J, 3M, etc.?
6. Methods describing the experiments represented in Fig S4A appear to be absent from the manuscript.

Reviewer #1 (Remarks to the Author):

Critique:

1) The authors call the protrusions 'intercellular filopodia'. However, they do not provide any evidence that would show that these bidirectional protrusions fit the definition of filopodia. For example, the Ena staining of Figure 2g does not show localization at the tips of the protrusions as one might expect from filopodia. The authors need to provide convincing evidence for their claim that these are filopodia or alternatively, call them simply 'cellular protrusions'. It is also not clear whether and how these protrusions are related to the filopodia at earlier follicle stages.

Answer: From our study, during ES9 to S10, cdc42 and its downstream signaling factors (including Ena) control both filopodia and stress fibers. Meanwhile intercellular filopodia are mainly inserted into the stress fiber networks in neighboring cells. Thus, it would not be easy to detect Ena signal from Figure 2g. We noticed that a lot of Ena signals are highly enriched at stress fiber regions, where intercellular filopodia are inserted; meanwhile, the Ena signals at stress fibers are much stronger than other regions at basal domain. So we think that the high Ena signals distributed at the filopodia-invaded cells strongly blocked our view of Ena signal at the tip of filopodia. However, this is not the situation for filopodia occurring at early stages such as S7-S8 (during S7-8, stress fiber intensity is weaker than stress fiber intensity in late stages), and also Ena signals in stress fibers of neighboring cells during S7-8 are undetectable (please see "Cetera M et al. Nat Commun, 2014" for Ena signals at S7-8 egg chamber basal domain. And we guess that in S7-S8, Cdc42 signaling/Ena are not involved in control of stress fiber network).

To better view Ena signal at the tip of filopodia protrusions, we cannot use anti-Ena antibody shown in Figure 2g, as we explained above. Thus, to avoid the noise effect on the clear view of Ena signals at filopodia protrusions, we used flipout system to randomly express both Ena-RFP and mCD8GFP for Ena signal and protruding membrane signal (although a direct F-actin reporter is better, but unfortunately, the one we have for this flipout-driven clonal expression of F-actin reporter is MoeABD-RFP, which has the same fluorescence as Ena-RFP). In this experiment, we can see that although Ena-RFP has very strong signal at stress fibers and basal junctions, Ena-RFP has very clear (but weak) signal at the tip region of protrusion (not completely tip of mCD8GFP, possibly due to the F-actin structure is shorter than mCD8GFP, as shown in main figure 1f). From live cell imaging, we can clearly see that this tip Ena-RFP relative position within protrusion is dynamically correlated with protrusion length change. In this condition, we tried Phalloidin staining, but unfortunately the signal of F-actin by phalloidin staining in protrusions are too weak to be detected, as the F-actin signals at stress fibers where protrusions are insert are super strong. Taken together with Ena RNAi phenotype and Ena rescue phenotype, thus we are quite confident that these protrusions are filopodia.

We tried to address the correlation between filopodia at S9-S10 and filopodia at earlier stages. Unfortunately, our experiment of live cell imaging didn't work, since the developmental process from S8 to early-middle S9 is too long, so that follicle cells are not healthy in late phases. But from Ena-RFP results, we can clearly see that in early stage such as S7-S8, Ena-RFP signals are similar to those in late stages, which is at the tip region of protrusions. The only difference is that protrusions are much shorter. Please see the comparison of filopodia and Ena-RFP in follicle cells from different stages of egg chambers. Here, we didn't put this comparison data since it is obviously reasonable. But we

mentioned in our main text, the short filopodia in early stages will be developed into much longer filopodia in late stages.

For late stages, it is not easy to detect Ena-RFP signals always at tip of protrusions. This is mainly due to the not-the same-focus issue (when protrusion is very long, the protrusion focus is some different from the short filopodia).

2) The Abstract of the manuscript makes little sense as it lacks any context for the described results. It is essential to mention that the protrusions are made by epithelial cells, that the coordination of the actin-myosin contractile network takes place within an epithelium, and it is essential to name *Drosophila* follicles as the studied system (e.g DV polarity makes no sense without this context).

Answer: Thank you very much for the reviewer's comment to clearly describe our results. As reviewer suggested, we revised our abstract and included the information of epithelial cells/epithelium and *Drosophila* follicles as the studied system.

3) The authors did not properly describe the location of the protrusions in relation to the apical-basal organization of the follicle cells at the beginning of the paper. So, the reader is left to speculate what is meant by 'protrusions between cells' until much later in the manuscript. Adding a schematic drawing, showing two follicle cells in a side view and indicating the location of the protrusions and stress fibers would help to orient the reader.

Answer: Thank you very much for the reviewer's comment. We agreed with this point and added a schematic drawing in order to show two follicle cells in a side view and also indicate the location of the protrusions and stress fibers. This schematic result has been introduced immediately in the beginning figure. We updated this in the current main figure 1b.

4) There are two apparently contradictory statements in the text: "We noticed the lengths of intercellular filopodia correlated with the amplitude of basal Myo-II oscillation" (lines 60-61) and "... we didn't detect a prominent signal correlation between filopodia length and basal Myo-II intensity from filopodia-producing or invaded-cells" (lines 132-134). Please, clarify.

Answer: We are so sorry for this confusing writing at lines 60-61 and lines 132-134. In lines 60-61, we quantified the average length of filopodia in epithelial cells at different stages, and we noticed that the average filopodia length is highly correlated with the average amplitude of global basal Myo-II oscillations at different stages (here we quantified the average length from many filopodia within

many follicle cells, but not the dynamic length of individual filopodia). However, in lines 132-134, we asked whether local individual filopodia length might be correlated with basal myosin oscillating changes, either from filopodia-producing or filopodia-invaded cells, and thus quantified the length of individual filopodia related to total or local basal Myo-II intensity within follicle cells mainly from LS9 egg chamber (the quantification of local Myo-II intensity is similar to the local Myo-II intensity in our current figure 5a,b). From this signal analysis, we didn't find any significant positive or negative correlation (we updated this quantification of individual filopodia length in the current supplementary figure 5). We did the revision at these 2 sections to clarify our results.

5) Although there is a negative correlation between myosin II levels in the vicinity of filopodia and the FRET ratio signal indicated by EcadTS (Fig.4A,B), it is apparent that intercellular filopodia do not sustain more tension than the rest of the cell cortex (Fig. 4A) as it would be expected if these structures were major anchoring points for stress fibers. Are these correlations similar in regions without filopodia? How do the authors comment on this?

Answer: Thank you very much for the reviewer's comments. Our previous writing might result in the confusion points. Although intercellular filopodia are inserted into the stress fiber network in neighboring cells and meanwhile they can provide a physical cue/guidance to maintain the medio-basal and DV-polarized distribution of stress fibers in neighboring cells, we didn't mean that intercellular filopodia would function as the only one structure to anchor stress fibers. As shown in the following cartoon, we can see that many stress fibers are anchored directly to the basal junctions at DV axis direction. Between basal junctions and stress fibers, we can easily detect prominent F-actin signals (only F-actin signal at regions out of stress fibers). Thus, it seems that many stress fibers are anchored directly at basal junctions by these only F-actin regions. This might explain why there are strong E-cad TS FRET signal at DV basal junctions. Interestingly, the vicinity of filopodia also showed strong E-cadTS tension by E-cad FRET ratio, which is highly negative correlated with local Myo-II intensity from stress fibers where filopodia are inserted. Although filopodia are not the only anchor points of stress fibers, from our data, it seems that this additional physical cue is efficient enough to highly maintain the distribution and polarity of stress fibers. This might be due to the much more E-cadherin adhesion tension supported by filopodia protrusion membrane regions, in addition to basal junctions, which can facilitate the distribution and polarity pattern, so that stress fibers would not be disorganized after egg chamber rotation stops.

Stress fibers, marked by green color and arrow, are many anchored by the DV basal junctions (thus this explains why E-cadherin adhesion tension at DV basal junctions is also very high and somehow

correlated with the global basal myosin intensity from live cell imaging). Outside of stress fibers, there are many actin-filaments, marked by purple color and arrow; and these F-actin structure links stress fibers to E-cadherin adhesion at basal junctions. E-cadherin adhesion is present in both bi-directional intercellular filopodia and basal junctions. From this cartoon, we highlighted that E-cadherin adhesion at intercellular filopodia is heavily surrounded by the stress fiber network in the filopodia-invaded cells, indicating that E-cadherin adhesion in filopodia might be tightly connected and linked with stress fibers in these invaded cells (thus it explains the tension/local myosin correlation shown in our main figure 5a, b). From this cartoon, we should also notice that, in addition to DV basal junctions that might anchor stress fibers, intercellular filopodia will additionally provide much more tension/biomechanic cues (Combining intercellular filopodia with basal junctions, the total tension/biomechanics by all E-cadherin adhesions should be significantly higher than the tension/biomechanics provided by only basal junctions). This additional physical tension/biomechanic cue might be strong enough to stabilize the DV and medial-basal stress fibers.

6) Drawing schematics in Fig. 4I and K would help the reader understanding these experiments more easily.

Answer: Thank you very much for the reviewer's suggestion. We added a schematic to show how we did the experiment for our original figure 4i and k. And we updated this cartoon in the current main figure 5i.

In addition, is there leakiness in PA-Cdc42 DN expression? In Fig.4J, the levels of medial myosin intensity seem reduced in PA-Cdc42 DN expressing cells compared to wild type, even before photoactivation.

Answer: Thank you for the reviewer's careful reading. PA-Cdc42DN-expressing cells had some mild leaking effect in our previous experiments with myosin-GFP background, compared with UtrABD-GFP background. We thought that we might have done this experiment in some light-leaking or expression-leaking conditions. So we re-did this experiment, by expressing PA-Cdc42DN in mild condition and also preventing some visible light, before photoactivation. In our new experiments, we successfully reduced the leaking effect to minimal level, and we can see that in some mCherry-PA-Cdc42DN strong expression-clonal cells, there are still strong levels of medial myosin intensity, and photoactivation treatment would not change their distribution patterns. We updated these new results in the current main figure 5j, k.

7) In Fig.4E, are the n values presented the number of filopodia or the number of cells analysed? If filopodia, they are very low and need to be increased for statistical tests to be appropriately used.

Answer: Sorry for this confusing information. The N values in Figure 4E represented the number of total cells analysed. And we revised the figure legends to clarify this confusion.

8) Supplementary figure 5e,f: Visually, the FRET data between the two encircled areas for the EcadTS-C +ML141 image look very different and I am surprised that this shows as non-significant. Please, comment.

Answer: In Supplementary Figure 5e,f experiments, we also noticed the opposite FRET pattern samples in these 2 encircled areas (myosin strong vs. myosin weak before drug treatment); and with

some unknown reason, it is very difficult to get non-significant FRET pattern difference samples under ML141. Please see the two opposite FRET patterns in the following figure:

FRET is low in left region in case 1; and FRET is high in left region in case 2; two types of circle marks the myosin strong vs. weak region that should be detected in the control egg chambers.

If we quantified all these data together, there is no significant difference between these 2 encircled areas, as shown in the current supplementary figure 6f.

9) In the Materials and Methods, the statement starting at line 493 “We found that autocorrelation...” is very long (spans throughout 6 lines) and needs to be rewritten for sake of clarity.

Answer: Thank you for the reviewer’s suggestion. To avoid the confusion, we removed these 6 lines. Because the 4 lines above are clear enough (“The distribution of oscillation periods.... gives a similar average period) and the similar methods have also been used in our previous papers (He L et al, Nat Cell Biol), we cited this reference. Thus, finally we removed these 6 lines to avoid the confusion.

10) In Figure legend 1M, the statement “Error bars...” (lines 581-582) doesn’t belong here, since there are no error bars in the figure.

Answer: Thank you for the reviewer’s careful reading, we revised the information of error bars and put them in the corresponding figure sections.

11) p 7, line 139: basal junction -> basal junctions

Answer: We corrected these words in line 139, as reviewer suggested.

12) p 10 lines 211-212: this sentence needs to be revised as it currently ignores that there are two plasma membranes in between.

Answer: Thank you for the reviewer’s comment. As suggested, we revised the sentence to mention that there are two plasma membranes in between.

Reviewer #2 (Remarks to the Author):

Major comments

#1. It is not clear to me which part of the egg chamber the authors are imaging. Please clarify with a schematic. This also leads to my other question: Is there a spatial inhomogeneity in cell shape and myosin intensity among the monolayer of follicular epithelium? Figs 5c-d show that the cells with higher myosin intensity have more elongated cell shape compared to the cells with lower myosin intensity. Please clarify.

Answer: As reviewer suggested, we added a schematic to show the region of the egg chamber imaged. This schematic is inserted in the updated Figure 6c.

Yes, as reviewer's feeling, there is a spatial inhomogeneity in cell shape and myosin intensity among the monolayer of follicular epithelium, as shown in our original Figure 5c-d, in LS9 and S10A, cells with higher myosin intensity have more elongated cell shape compared with the cells with lower myosin intensity. But these cells have significant higher myosin intensity and contractility ability (if analyzing cell area dynamics, these cells are very active, data not shown here), compared with the ones with normal shape and lower myosin intensity. Currently, it is still unclear how this elongated shape is controlled. Because all these are unclear yet, we didn't update this clarification information in our main text.

#2. The authors used the results from E-cad FRET force sensor to interpret the membrane tension. The force measured with E-cad FRET force sensor represent the tension exerted within E-cad, which is distinct from the membrane tension. The authors seem to miss-interpret the FRET signal of E-cad force sensor. This confusion needs to be addressed.

Answer: Thank you very much for the reviewer's careful reading. We are sorry to miss-interpret the membrane tension by E-cad FRET force sensor. To clarify the tension force, we revised our main text and introduced why we need to use E-cad FRET force sensor to monitor and measure the effect from the myosin contractility sensed by intercellular filopodia. Since E-cad adhesion is tightly linked with actomyosin network at basal domain of follicle cells (as we reported previously, oscillatory myosin intensity is consistent with and 0.5min earlier than basal membrane area reduction change, mainly along DV axis direction), E-cad adhesion tension is highly consistent with the local/global actomyosin contractility. This is the reason why we used E-cad FRET force sensor to measure and quantify the tension of E-cad adhesion in both basal junctions and intercellular filopodia. And it really works to measure and quantify mechanic tension driven by myosin contraction ability. In addition, we also thought that membrane tension might not be the precise way to monitor and quantify the myosin contractility, considering that membrane tension can also be affected by many other factors. Thus, finally we revised our writing.

#3. In Fig. 4a-b, although the authors highlighted that changes in tension at the intercellular filopodia region (within the ROIs), the images shows that the tension outside of the intercellular filopodia regions also correlated with the MyoII intensity within the cells, suggesting the change in tension is not restricted to the region related to intercellular filopodia. How the authors reconcile this? Please explain. This makes the author's argument "intercellular filopodia might function as biomechanical sensors" unconvincing.

Answer: Thank you very much for the reviewer's comments. Our previous writing might result in the confusion points. Although intercellular filopodia are inserted into stress fiber network in neighboring cells and meanwhile they can provide a physical cue/guidance to maintain the medio-basal and DV-polarized distribution of stress fibers in neighboring cells, we didn't mean that intercellular filopodia would function as the only one structure to anchor stress fibers. As shown in the following cartoon, we can see that many stress fibers are anchored directly to the basal junctions at DV axis direction. Between basal junctions and stress fibers, we can easily detect prominent F-actin signals (only F-actin signal at regions out of stress fibers). Thus, it seems that many stress fibers are anchored directly at basal junctions by these only F-actin regions; and this might explain that the tension outside of the intercellular filopodia regions, mainly at DV basal junctions, also correlated with the MyoII intensity within the cells. Interestingly, the vicinity of filopodia also showed strong E-cadTS tension by E-cad FRET ratio, which is highly correlated with local Myo-II intensity from stress fibers where filopodia are inserted. Although filopodia are not the only anchor points of stress fibers, from our data, it seems that this additional physic cue is efficient enough to highly maintain the distribution and polarity of stress fibers. This might be due to the much more E-cadherin adhesion tension supported by filopodia membrane regions, in addition to basal junctions, which can facilitate the distribution and polarity pattern, so that stress fibers would not disorganized after egg chamber rotation stops.

Stress fibers, marked by green color and arrow, are many anchored by the DV basal junctions (thus this explains why E-cadherin adhesion tension at DV basal junctions is also very high and somehow correlated with the global basal myosin intensity from live cell imaging). Outside of stress fibers, there are many actin-filaments, marked by purple color and arrow; and these F-actin structure links stress fibers to E-cadherin adhesion at basal junctions. E-cadherin adhesion is present in both bi-directional intercellular filopodia and basal junctions. From this cartoon, we highlighted that E-cadherin adhesion at intercellular filpodia is heavily surrounded by the stress fiber networks in the filopodia-invaded cells, indicating that E-cadherin adhesion might be tightly connected and linked with stress fibers in these invaded cells (thus it explains the tension/local myosin correlation shown in our main figure 5a,b). From this cartoon, we should also notice that, in addition to DV basal junctions that might anchor stress fibers, intercellular filopodia will additionally provide much more tension/biomechanic cues (Combining intercellular filopodia with basal junctions, the total tension/biomechanics by all E-cadherin adhesions should be significantly higher than the tension/biomechanics provided by only basal junctions). This additional physical tension/biomechanic cue might be strong enough to stabilize the DV and medial-basal stress fibers.

Finally to avoid the confusing and unconvincing sentence “intercellular filopodia might function as biomechanical sensors”, we revised it to “This indicates that E-cadherin adhesion at intercellular filopodia might be biomechanically sensitive to local actomyosin contractility, implicating the potential role of intercellular filopodia in sensing their invaded environment, reminiscent of the functions of filopodia in sensing the stiffness of extracellular matrix”.

#4. Based on the drug treatments (ionomycin and Y27632) to control the actomyosin contractility, the authors claimed that “stable actomyosin contractility is able to control the growth of intercellular filopodia”. It is not clear to me what “stable actomyosin contractility” means in the context of wild-type egg chamber. Please clarify. In wild-type, the actomyosin contraction fluctuate in time, i.e., not stable. The other related question is whether the myosin density fluctuation diminished in these treatments and it looks like “stable” contractility. Please explain.

Answer: We are so sorry for this confusing writings. “Stable actomyosin contractility” should be actomyosin contractility without myosin intensity fluctuation. If myosin intensity is oscillatory, the force intensity is varied. In both drug treatments, the myosin fluctuation was strongly diminished. In the following imaging, we put the myosin intensity dynamics in Wt and Ionomycin treatment conditions so it is clear to see that myosin oscillation is diminished and myosin intensity is quite constant. Thus, we revised our writing to clarify this confusion. Please see the updated main text line “Considering the stochastic characteristic of pulsatile Myo-II contractility, here we used either Ionomycin or a chemical inhibitor of ROCK, Y27632, to enhance or inhibit basal actomyosin contractility, each of which blocks oscillatory behaviours, and thus constant basal Myo-II intensities were detected in both chemical treatments” in the current main text.

Ionomycin treatment strongly enhanced basal myosin intensity and meanwhile it blocked oscillatory behaviours. Although representative figures are not shown here, Y27632 treatment strongly inhibited basal myosin intensity and also blocked oscillatory behaviours.

#5. The authors claimed that “Light-induced inhibition of Cdc42 at the filopodia region resulted ..., but not in the light-treated PA-Cdc42DN cells themselves (Fig. 4i-l)” without showing the images of the light-treated PA-Cdc42DN cells. The authors need to provide the images that include the neighboring non- light-treated cells. Because of this lack of evidence, it is overstate to claim that “intercellular filopodia might maintain the medio-basal localization of stress fibers in their invaded cells”.

Answer: As reviewer suggested, we provided the images that include the neighboring non-light-treated cells. We updated this in the current main figure 5j, since our original images for this

experiment are difficult to see the neighboring PA-Cdc42 clonal cells. However, we can see from images, some cells might be not easy to see the F-actin or myosin signals. This is due to the not good-focus issue (considering that some curved plane occurs for some cells, which is very often for a long time series of imaging) during live tissue imaging, which made difficult to achieve the good view of signals in all cells at different time points. Thus, we mainly focused on the cells where we can see the F-actin or myosin signals for the quantification. From our quantification, it is easy to see that wt cells have the effect of medio-basal distribution, but not the clonal cells, in this experiment.

#6. The downstream effectors of Cdc42, WASP, Ena, and Dia were claimed to be “distributed within the F-actin regions of both stress fibers and filopodia” (Supp. Figs. 3c-e). The staining signal of these effectors were everywhere without clear distributions, which made these images and claims unconvincing.

Answer: Thank you very much for the reviewer’s careful reading. For WASP and Dia signals, they are distributed everywhere. It might be due to their control of not only stress fibers but also other F-actin structure located near basal junctions. For Ena, we can get many significant signals enriched at the both ends of stress fibers and also the clear (but weak) signals distributed at the tip region of filopodia protrusion (please see the updated main figure 1f, g), while there is still some signals located near basal junctions. The wide distribution of these effectors at basal domain of epithelial cells indicates that these effectors might affect some other structures or play other functions. However, our RNAi knockdown experiments didn’t show prominent effect on basal junctions while all RNAi significantly reduced filopodia and stress fiber signals/patterns, indicating that they mainly controls both stress fibers and filopodia. To better conclude our results, we revised our writing in the updated main text, in order to avoid our original unclear statement.

#7. The authors claimed that “This supracellular network functions to maintain the medio-basal distribution and the global DV polarity of Myo-II contractility (Fig. 5j).”. It has been shown that the global DV polarity of contraction is governed by the direction of ECM. Thus I felt that this is an overstatement and creates confusions. The authors need to thoroughly and fairly discuss on the global DV polarity.

Answer: For the global control of DV polarity of myosin contractility, it has been shown that the global DV polarity of contraction is governed by the direction of ECM. This is mainly due to the clockwise or anticlockwise rotation of follicle cells so that collagen and other matrix proteins are secreted and aligned along the DV axis. Through the interaction with integrin adhesion complex of follicle cells, stress fibers are mainly located at the DV axis direction. If we checked integrin RNAi phenotype, what we observed that stress fibers are strongly reduced but still mainly distributed along the DV axis and mainly at medial basal cortex of follicle cells from S9 to S10B (please see the following figures), indicating that in later stages, filopodia structure might function as a memory of mechanic cue to maintain the effect of ECM direction. To clarify this relationship between ECM direction and filopodia, we discussed the formation of the DV polarized stress fibers, the formation of filopodia, and how filopodia function as a type of memory to remember the effect of the egg chamber rotation and ECM polarity, so that after rotation is over, stress fibers are still well-organized by Cdc42 activity-mediated filopodia protrusions. However, it is unclear about the link and correlation between ECM-integrin adhesion and Cdc42 activity. Thus, we put this unclear information (needs future study) into the updated discussion section.

Mosaic clones with beta-integrin RNAi at S10A, which showed stress fibers are strongly reduced but still DV-polarized and medial basal distributed. This result is similar to the phenotype of DN-Rho1 in follicle cells at the same stages. And we knew before that integrin adhesion positively regulates the Rho1 basal activity. It indicates that at late stages, integrin effects might become weaker, while Cdc42-mediated stress fibers and intercellular filipodia might function as a memory for DV polarity, which follows the original DV polarity induced by rotations, ECM direction and integrin adhesion.

#8. In the model (Fig. 5j), the authors claimed that the “Intercellular filopodia” link to stress fibers through F-actin (red in the figure). To claim this link, the authors provided Supp. Fig. 3a-b. The actin rich structure that the authors highlighted in Supp. Fig. 3a-b is shorter than the “Intercellular filopodia” shown in the other figures (for instance Fig. 1b-c and Suppl. Fig 1f-g). It looks like this actin rich structure (in Supp. Fig. 3a-b) was not located among the actomyosin meshwork in the neighboring cell, suggesting this fiber is a part of stress fibers. The authors used the images of membrane marker (mCD8) and presented the “Intercellular filopodia”. However, there are no clear images of actin fiber representing the F-actin shown in Fig. 5j (red in the figure). Thus I felt this model is overstating their findings. The authors need to show more clear actin data and/or clarify this.

Answer: Thank you for the reviewer’s careful reading. If we used moeABDRFP in the flipout experiments, we can clearly see that F-actin length within filopodia protrusion is very long and it was deeply inserted into the neighboring cells (please see the current main figure 1e, f). However, if we used UtrABDGFP SqhRFP to view the F-actin structure in filopodia, it was not good enough to view the F-actin structure in the filopodia protrusion tip, so that it looks like this actin rich structure (in our original Supp. Fig. 3a-b) was not located into the actomyosin meshwork in the neighboring cell, but rather looks like a part of stress fibers. However, we didn’t consider this F-actin structure in protrusions as a part of stress fibers, since it is distributed outside of myosin signal region. The reason why we can’t detect the nice UtrABDGFP signals at filopodia protrusions which are inserted into stress fiber network in neighboring cells, might be due to that UtrABDGFP signals at stress fibers in neighboring cells are much stronger than the signal of UtrABDGFP in filopodia protrusions. To better view the F-actin signal in intercellular filopodia, we used the flipout system in which only F-actin in the mosaic clone cells can be viewed, together with ubiquitous myosinGFP signals in all cells, to monitor the stress fiber network. We updated these images in the current supplementary figure 2b, b’. From these images, we can clearly see that F-actin marked by moeABDRFP in filopodia protrusions is very long and is also deeply inserted into the actomyosin network in neighboring cells. Thus, based on these updated new data, we think that our schematic cartoon is correct.

Minor comments

#a. “data not shown” should not be used. Please provide the data.

Answer: This data has been updated in the current supplementary figure 5.

#b. In Fig. 4a-b, the authors claimed that “the tension ... at the intercellular filopodia was correlated with the intensity of local Myo-II at the region where a filopodium sticks”. How the local MyoII intensity was measured? How the ROIs were set? Please explain.

Answer: Normally we used Matlab to set up ROI in the stress fiber network near filopodia. We measure the intensity of local Myo-II at the regions of each 0.85 micron from the filopodia membrane (left region 0.85um to left membrane of filopodia, and right region 0.85um to right membrane of filopodia, and also 0.85um to the tip of filopodia; we didn't measure the basal junctions from filopodia base). To guarantee the precise quantification, we used Matlab to define this ROI so that all selected regions have the similar area to avoid the side effect of selection area difference. We updated this information in the methods section.

#c. In PA-Cdc42CA/DN experiments (Fig 3 &4), how the “% of medio-basal F-actin (or MyoII) intensity” was measured? Precise cell shape information is required to measure this value. However, from the images presented, I could not see the clear cell shape. The authors need to 1) clarify how the cell shape measurement was done, and 2) show the images showed clear cell shape.

Answer: Sorry for this unclear information. But for quantification, we quantified the % of signal intensity distribution based on mCherry-PACdc42 signal which can show the membrane. In the following figures, we showed the example of cell shape at the beginning time point of all experiments. But mCherry-PACdc42 signals are often hidden by GFP signals, it is not easy to detect. But including both channels takes too much space for main figures. Thus, we didn't include this. If necessary, we can update the mCherry-PACdc42 channel in one supplementary figure.

The figures show both mCherry-PACdc42 and MyosinGFP or UtrABDGFP (F-actin reporter). Attention: some mCherry-PACdc42DN cells in Fig.5j and 5l have less Myosin or F-actin intensity at basal medial cortical regions, while blue light illumination as shown Fig. 5i didn't reduce medial myosin or F-actin, but oppositely at some late PA-treatment time periods, the weak Myosin or F-actin medial intensity got strongly enhanced at these mCherry-PACdc42DN clonal cells surrounding a central WT cells.

However, for the central WT cells, at the late PA-treatment time periods, myosin/F-actin signals are strongly redistributed near basal junction, thus leaving their medial signals to be empty. For C450M control in the current Supplementary Fig. 7e-h, the central WT cells have no prominent changes in both Myosin and F-actin medial basal distribution.

#d. The authors claimed that “the morphogenetic change mediated by Cdc42 inhibition was not related to either apical-basolateral polarity or the global tissue rotation occurring from S5 to S8 (Supplementary Fig. 7)”. It is hard to understand this without providing the complement data sets from control egg chamber. Please provide these data.

Answer: Thank you very much for the reviewer’s comments. We need to explain how we did the apical-basolateral polarity experiments and rotation experiments. In the apical-basolateral polarity experiments, we used the flipout expression system to randomly drive the expression of UAS-Cdc42DN in some GFP-coexpressing cells, while all other non-GFP expressing cells are the control wild type cells. From the comparison between GFP-positive (Cdc42-inhibition) and GFP-negative (wild type) cells within the same egg chamber, it is easy to check the effects of Cdc42DN on the apical-basolateral polarity. However, for the rotation of egg chambers, we cannot use the same flipout system, considering the low clonal number in a whole tissue, which might not significantly affect the rotation of whole tissue. Thus, for the experiments of egg chamber rotation, we used another flipout system which can drive most of clonal cells within a whole tissue (we can detect a lot of NLS-dsRed in most of epithelial cells within egg chamber). Thus, in our original supplementary figure 7, we really included both controls. To clarify the confusion, we labelled the clonal cells with arrows in our original supplementary figure 7A. We updated this in the current supplementary figure 9.

#e. It was quite a hustle for me to read through the manuscript mainly because so many critical data were shown in the supplement and I need to go back and forth between main and supplemental figures.

Answer: Thank you very much for the reviewer’s comments. We moved one supplementary figure into main figure, considering its importance. But we mainly keep the control experiment results and other supplementary results in supplementary figures.

Reviewer #3 (Remarks to the Author):

Major concerns:

1. Sufficient validation of the optogenetic tools for controlling Cdc42 activity is lacking. For the data shown in Fig S4A, the 'lit' mutant of PA-Cdc42 appears to bind PBD with higher affinity than the 'dark', but how do these apparent differences in affinity correspond to the ability of light to control PA-Cdc42 activity in vivo? Additional experiments should be performed to assess the extent to which PA-Cdc42 can modulate the activity of its direct effectors in cells. For example, in ref #25, the authors showed that a similarly-designed light-controllable Rac could modulate PAK localization/activity in a light-dependent manner. Such experiments are critical for determining the extent to which PA-Cdc42 activity is enhanced in the 'lit' vs. 'dark' state. F-actin/myosin organization and other readouts used in this study are too many steps removed from Cdc42 activity to reliably make such assessments.

Answer: As reviewer suggested, we did the experiments of PAK-RBD-GFP to assess the extent to which PA-Cdc42 can modulate the activity of its direct effectors in cells. This data has been updated in the supplementary figure 4b, c. The experimental methods have been updated in methods sections. Since PAK-RBD-GFP signal is very weak, compared with our Myosin-GFP or UtrABD-GFP reporter, we cannot use the same strategy to check the lit or dark effect of PA-Cdc42 on PAK-RBD-GFP (a little bit 458um local illumination can strongly bleach the PAK-RBD-GFP signals at PA-treatment region in 1-2mins, so it is impossible to view the lit vs. dark effect by live cell imaging). Thus, we dissected the PA-Cdc42DN or PA-Cdc42CA clonal egg chambers under dark status; after dissection, with around 10-15min of moderate visible-light exposure or with dark condition, we fixed the samples under dark. With this method, we can detect the prominent effect of PA-Cdc42DN or PA-Cdc42CA in lit vs. dark condition.

Along similar lines, does expression of PA-Cdc42DN produce any defects similar to those of Cdc42DN cells independent of light-based activation? Experiments should be performed as in Fig 1I where F-actin and Myo-II levels are compared among 'control', 'Cdc42DN', and 'PA-Cdc42DN' cells kept in the 'off' (dark) state. As PA-Cdc42DN in particular is used to generate some of the most compelling observations in this study (e.g., that disrupting intercellular filopodia produces defects in F-actin/myosin regulation in adjacent cells) it is essential to ensure that these optogenetic tools for control of Cdc42 activity are adequately vetted.

Answer: As reviewer suggested, we did the control experiment to show the leaking effect of PA-Cdc42DN expression, under the dark state. F-actin and Myo-II levels have been compared between 'PA-Cdc42DN'- or 'PA-Cdc42DN C450M'-expressing cells kept in the 'off' (dark) state, in the flipout driven clonal egg chambers; while we didn't reshown the images for both control and Cdc42DN, but showed the quantification of both too. All these results have been updated in the current Supplementary Fig. 4d, e. We can see that PA-Cdc42DN cells under dark have minimal leaking effect, in the expression conditions we used for our original main figure 3 and 4. Thus, with these new data, we can conclude that our optogenetic tools for control of Cdc42 activity are adequately vetted.

2. The data linking Ena, Dia, and WASP to Cdc42 activity is somewhat thin. While the authors provide strong evidence that disruption of these factors phenocopies loss of Cdc42, little is done to demonstrate that these factors function downstream of Cdc42 to promote F-actin/myosin organization and intercellular filopodia growth. There is one instance where expression of Ena is

shown to rescue defects in F-actin organization in Cdc42DN cells (Fig. 2L-M); however, similar experiments were not attempted with either WASP or Dia. Does either WASP or Dia expression rescue these defects in Cdc42DN cells? Alternatively, does the further disruption of Cdc42 in either WASP or Dia knockdown cells result in compounded defects in F-actin/myosin organization or filopodia morphology?

Answer: In addition to check the rescue effect of Ena on Cdc42 loss, we also did the rescue experiment of WASP and Dia on actin organization and filopodia. These data have been updated in the current supplementary figure 3. In this new supplementary figure, we can see that WASP or Dia overexpression can also significantly rescue the phenotype of filopodia and stress fiber F-actin distribution back to the normal, although the rescue is some weaker compared with the rescue mediated by Ena. We also updated the result of filopodia rescue by Ena. This has been updated in the current main figure 3n.

In the opposite case, we also tested whether Ena RNAi, WASP RNAi or Dia RNAi can result in compounded effects in F-actin organization, shown in the following figures.

Although it is not easy to see the difference from the figures (considering that Cdc42DN effect is already quite strong), we can still detect that co-expression of these 3 RNAi with Cdc42DN can produce stronger effects, compared with either RNAi alone or Cdc42DN alone. Because of the limited space and meanwhile a lot of current data in this manuscript version, we didn't put this result in our main text. If reviewer feels it is important, we can update it in one supplementary figure.

Minor concerns/comments:

1. The experiments depicted in Fig 4I-L should be repeated with the PA-Cdc42DN C450M control.

Answer: As reviewer suggested, we did the control experiment for our original figure 4i-l, and the results of PA-Cdc42C450M have been included in supplementary figure 7. From this control result, we can see that light illumination of PA-Cdc42C450M-expressing cells at membrane region would not affect both intercellular filopodia and stress fiber distribution in the central wild type cells.

2. In the figures and movies depicting the optogenetic experiments, it would be helpful to indicate where activating light is being applied.

Answer: We added the information about where activating light is being applied, in either figure legends or movie legends, or in figures.

3. The arrow depicting activating light in Fig 3A appears to be inverted.

Answer: Thank you very much for the reviewer's careful reading, we corrected this mistake.

4. Are the data represented by the controls in Fig 2B-C identical to the controls in Fig 1F and 1I? If so, this should be made explicit in the text or figure legend.

Answer: Thank you very much for the reviewer's careful reading. We made a mistake here, and now we updated the correct data for controls in our original Figure 2B-C. Please see the update results in the current main figure 3b-c.

5. What is the time interval between 'before PA' and 'after PA' in Fig 3J, 3M, etc.?

Answer: We put the information in the corresponding figure legends about the time interval between "before PA" and "after PA" in Fig 3J, 3M, etc. Please see this information in the updated figure legends.

6. Methods describing the experiments represented in Fig S4A appear to be absent from the manuscript.

Answer: Thanks for the reviewer's reminding comment, we really forgot to include the experiment protocol in our methods section. We updated this in the methods section.

Reviewers' comments:

Reviewer #1 (Remarks to the Author):

The authors have addressed my concerns and the manuscript overall has improved. The authors' findings are very interesting and the study has been carefully executed and documented. Overall, I find the manuscript excellent. I have only some minor points.

Minor points:

- The authors describe the filopodia as emerging from the basal 'domain'. However, as the filopodia form in continuation with the stress fibers, it seems more likely that they emerge from the basal-most aspect of the lateral membrane. If this correct, I would ask the authors to make according changes to the manuscript.

- Fig 3 h-j: The effect on Dia, WASP and Ena appears to be not restricted to the Cdc42 expressing cell but is also seen in the surrounding cells. Is the effect non-cell autonomous? Please, comment

- I strongly recommend to add the Sup Fig 4 panels f and g to main figure 4

- The authors have now included a nice drawing, Fig 1b that illustrates the location of the intercellular filopodia. I suggest using thinner green lines to separate the stress fibers.

- lines 98-101: please, revise the sentence as a "strong positive correlation between membrane and F-actin" does not specifically define filopodia.

- Grammar is a bit shaky in several parts of the manuscript, particularly in the introduction and M&M.

Figure 2a-c are not referenced in the manuscript text

Line 150: please, revise the title (not clear what it means)

- the term 'confirms/confirmed' is used in several places where "show or indicate" would be more appropriate (e.g. line 168)

- Line 199: I did not understand this sentence

- Line 249 and 256: Supp Fig 8: indicate specific panels

- line 279: replace 'tissue shape' with 'follicle shape'

- italics in M&M is used erratically. All genotypes should be indicated in italics

- line 357: what was the CoinFLP-lexA::GAD.GAL4 used for?

- line 359: change 'crossing transgenic flies with' to 'crossing flies with UAS transgenes to'

- line 458-459: "beneath the basal surface": is it not 'above'?

- lines 511-513: I did not understand

- line 700 of text and line 5 of sup fig leg: replace 'random' with 'stochastic'

- line 7 of Sup fig leg: reference for Indy-GFP is missing

- line 126 of Sup fig leg: 'dotted line' ? - figure shows continuous lines

- Sup fig 1i: what is the orientation of the cells regarding the D/V axis?

- Sup fig 2: Arrows, arrowheads and boxes are barely visible: please, improve!

Comments on reviewer 2 report:

Reviewer 1: Point #2

I agree with Reviewer 2 that the E-cad tension sensor data do not necessarily reflect E-cad homophilic interactions (see Borghi N et al (PNAS 2012), and that the term 'E-cad adhesion tensions' should be avoided. The authors' response to the reviewer's comments could have been better. Despite this, I don't see a problem with the description of the experiment in the Result section where the authors state they used the sensor "to quantify the mechanic tension of follicle cells in response to actomyosin contractility". And, I think that the authors are sufficiently cautious with their interpretation of their data with this sensor in the Result section. On the other hand, the statements about E-cad in the discussion seem too strong as it has not been shown that filopodia from neighboring cells adhere to each other through E-cad. I urge the authors' to be more cautious with their interpretation.

Reviewer 1: Point #3

I raised the same question about the homogeneous FRET signal around the whole cell in my first review. It looks as if E-cad is under tension along the whole membrane of the cell although the stress fibers are polarized. Nevertheless, the FRET signal clearly correlates with the oscillations in the contractility of the stress fiber network, and the FRET signal is present in the region of the filopodia. These data do not point to a special function of the filopodia in mechanosensing but it is consistent with the notion that mechanical forces act on DE-cad at these protrusions. It remains unanswered how exactly the filopodia regulate the polarity of stress fibers, whether simply through a much larger surface area (which increases the overall capacity of contact sites for actin bundles and for adhesive contacts) or through another mechanism. It will be interesting to figure out in the future how this works. An experiment that could directly address membrane tension in the interdigitating region of neighboring cells would be the measurement of recoil velocity after laser cuts (and comparing it to the behavior of the non-interdigitating membranes). Such experiments, however, would be challenging and have not been attempted in ovaries yet, as far as I know.

Reviewer 1: Point #4

I agree with the request of the 2nd reviewer.

Reviewer 1: Point #5

I felt that the experiment has been clearly explained in the figure legend. Fig. 5j shows a wild-type cell outlined by a stippled line that is surrounded by five PA-Cdc42DN cells and one wild-type cell. The stippled line also indicates the applied ring of blue light illumination. The quality of Fig. 5j is good. In contrast, Fig. 5l (F-actin staining) is not interpretable. I strongly recommend adding images that only show the green channel for both, 5j and 5l. This would help a lot to better reveal the orientation and amount of stress fibers in PA-Cdc42DN and wild-type cells. In addition, outlining the cell boundaries of all cells in each image by dotted lines (different color for different genotypes) or

highlighting the mutant and wild-type cells by asterisks (different colors) would be helpful. Furthermore, discrepancies between graphs of the original and revised manuscript, as pointed out by the 2nd reviewer, have to be resolved! This is indeed an important experiment. The English needs to be improved here and throughout the manuscript.

Reviewer 1: Point #6

In wild-type cells, Dia and Wasp puncta appear to be somewhat enriched in the region where the stress fibers are located. Even if there is no enrichment, there are clearly many puncta in the stress fiber area, which is consistent with a function there. To show the normal distribution of Dia and Wasp is valuable. Especially, as in Cdc42DN cells, the pattern seems to change and the central accumulation is no longer visible, consistent with a disturbance of the stress fibers.

My concern, as mentioned in my second review is that the neighboring wild-type cells of the Cdc42DN clone seem to have the same Dia and Wasp pattern as the mutant cell. Furthermore, the focal planes in the images of Fig. 3e-j are not quite comparable as seen by the mCD8-GFP staining, which appears in some images as a ring (lateral membrane) and in others as a flat surface (basal membrane).

Reviewer 1: Point #7

Taking together all experimental data, I concur with the Authors' interpretation and model. I agree with the authors not to integrate the integrin data into the manuscript. They seem preliminary and to analyze the relationship between Cdc42 and integrin will require considerably more work, which in my opinion goes beyond the scope of this manuscript.

Reviewer #2 (Remarks to the Author):

First of all, I appreciated the authors efforts to clear my comments. I reviewed the comments and the revised manuscript carefully. After a rigorous evaluation, I found that some of the data were not presented/analyzed in a proper way, some of the data were not convincing, and some of the claims were overstated. Overall, the manuscript suffers from pitfalls that preclude publication in a high standard journal like Nature Communications. Here are my detailed comments.

Major comments

#2. The authors used the results from E-cad FRET force sensor to interpret the membrane tension. The force measured with E-cad FRET force sensor represent the tension exerted within E-cad, which is distinct from the membrane tension. The authors seem to miss-interpret the FRET signal of E-cad force sensor. This confusion needs to be addressed.

Answer: Thank you very much for the reviewer's careful reading. We are sorry to miss-interpret the membrane tension by E-cad FRET force sensor. To clarify the tension force, we revised our main text and introduced why we need to use E-cad FRET force sensor to monitor and measure the effect from the myosin contractility sensed by intercellular filopodia. Since E-cad adhesion is tightly linked with actomyosin network at basal domain of follicle cells (as we reported previously, oscillatory myosin intensity is consistent with and 0.5min earlier than basal membrane area reduction change, mainly along DV axis direction), E-cad adhesion tension is highly consistent with the local/global actomyosin contractility. This is the reason why we used E-cad FRET force sensor to measure and quantify the tension of E-cad adhesion in both basal junctions and intercellular filopodia. And it really works to measure and quantify mechanic tension driven by myosin contraction ability. In addition, we also thought that membrane tension might not be the precise way to monitor and quantify the myosin contractility, considering that membrane tension can also be affected by many other factors. Thus, finally we revised our writing.

[Responds to Answers #2]

First of all, the term "E-cadherin adhesion tensions" is confusing. To me, this implies the tension between E-cad through homophilic interactions. I do not think the E-cad tension sensor used in this manuscript is able to measure the tension between E-cad through homophilic interactions. Using "E-cadherin adhesion tensions" throughout the manuscript is not proper and is confusing.

The authors stated that "Since E-cad adhesion is tightly linked with actomyosin network at basal domain of follicle cells (as we reported previously, oscillatory myosin intensity is consistent with and 0.5min earlier than basal membrane area reduction change, mainly along DV axis direction)". It is not clear to me which data shows "E-cad adhesion is tightly linked with actomyosin network at basal domain of follicle cells". The time lag between the rate of decrease in basal area and change in myosin intensity, which the authors referring here (in the parentheses), does not logically support the statement "E-cad adhesion is tightly linked with actomyosin network at basal domain of follicle cells".

The authors further stated that "And it really works to measure and quantify mechanic tension driven by myosin contraction ability.". Without showing convincing evidences and/or arguments, it is impossible to believe a statement like "it really works to...".

Last, the time lag between the rate of decrease in basal area and change in myosin intensity that the authors reported previously in He et al., (NCB, 2010) was ~0.9min, not 0.5min as the authors stated in this response. It looks like the authors did not pay proper attention to their own works.

#3. In Fig. 4a-b, although the authors highlighted that changes in tension at the intercellular filopodia region (within the ROIs), the images shows that the tension outside of the intercellular filopodia regions also correlated with the MyoII intensity within the cells, suggesting the change in tension is not restricted to the region related to intercellular filopodia. How the authors reconcile this? Please explain. This makes the author's argument "intercellular filopodia might function as biomechanical sensors" unconvincing.

Answer: Thank you very much for the reviewer's comments. Our previous writing might result in the confusion points. Although intercellular filopodia are inserted into stress fiber network in neighboring cells and meanwhile they can provide a physical cue/guidance to maintain the medio-basal and DV-polarized distribution of stress fibers in neighboring cells, we didn't mean that intercellular filopodia would function as the only one structure to anchor stress fibers. As shown in the following cartoon, we can see that many stress fibers are anchored directly to the basal junctions at DV axis direction. Between basal junctions and stress fibers, we can easily detect prominent F-actin signals (only F-actin signal at regions out of stress fibers). Thus, it seems that many stress fibers are anchored directly at basal junctions by these only F-actin regions; and this might explain that the tension outside of the intercellular filopodia regions, mainly at DV basal junctions, also correlated with the MyoII intensity within the cells. Interestingly, the vicinity of filopodia also showed strong E-cadTS tension by E-cad FRET ratio, which is highly correlated with local Myo-II intensity from stress fibers where filopodia are inserted. Although filopodia are not the only anchor points of stress fibers, from our data, it seems that this additional physic cue is efficient enough to highly maintain the distribution and polarity of stress fibers. This might be due to the much more E-cadherin adhesion tension supported by filopodia membrane regions, in addition to basal junctions, which can facilitate the distribution and polarity pattern, so that stress fibers would not disorganized after egg chamber rotation stops.

[Responds to Answers #3]

May be my previous comments were not clear for the authors. My major issue was that in the current Fig. 5a-b (originally Fig. 4a-b), the AP boundaries (vertical line in the cartoon the authors provided), which are parallel to the basal stress fibers and presumably do not interact with the basal stress fibers as strong as the DV boundaries, also change the FRET signal. In other words, there is no difference in the tension changes between AP boundaries and DV boundaries.

In He et al., (NCB, 2010), the authors reported that "The basal area change was highly polarized, correlating with a change in the length of cells along the D-V axis, whereas we observed little or no change in cell length along the anterior-posterior (A-P) axis (Fig. 2a)". This strongly implied (and the authors claimed) that the contraction has direction and it is along D-V axis.

How the authors explain the discrepancy between the two papers? Why the results of E-cad tension sensor showed non-polarized tension, even the two different boundaries (AP vs DV) behave differently (i.e., one is changing its length, the other is not)?

Thus, I feel the author's argument "the mechanosensory role of ^{f_{SEP}} filopodia between cells" (in the original version, the authors claimed "intercellular filopodia might function as biomechanical sensors") unconvincing.

#4. Based on the drug treatments (ionomycin and Y27632) to control the actomyosin contractility, the authors claimed that “stable actomyosin contractility is able to control the growth of intercellular filopodia”. It is not clear to me what “stable actomyosin contractility” means in the context of wild-type egg chamber. Please clarify. In wild-type, the actomyosin contraction fluctuate in time, i.e., not stable. The other related question is whether the myosin density fluctuation diminished in these treatments and it looks like “stable” contractility. Please explain.

Answer: We are so sorry for this confusing writings. “Stable actomyosin contractility” should be actomyosin contractility without myosin intensity fluctuation. If myosin intensity is oscillatory, the force intensity is varied. In both drug treatments, the myosin fluctuation was strongly diminished. In the following imaging, we put the myosin intensity dynamics in Wt and Ionomycin treatment conditions so it is clear to see that myosin oscillation is diminished and myosin intensity is quite constant. Thus, we revised our writing to clarify this confusion. Please see the updated main text line “Considering the stochastic characteristic of pulsatile Myo-II contractility, here we used either Ionomycin or a chemical inhibitor of ROCK, Y27632, to enhance or inhibit basal actomyosin contractility, each of which blocks oscillatory behaviours, and thus constant basal Myo-II intensities were detected in both chemical treatments” in the current main text.

Ionomycin treatment strongly enhanced basal myosin intensity and meanwhile it blocked oscillatory behaviours. Although representative figures are not shown here, Y27632 treatment strongly inhibited basal myosin intensity and also blocked oscillatory behaviours.

[Responds to Answers #4]

The data showing the relative myosin intensity change over time under Ionomycin treatment should be shown at least in the supplement.

Why the authors could not show the similar results under Y27632 treatment? Please provide this and added to the supplement. I strongly think this is a very important result.

#5. The authors claimed that “Light-induced inhibition of Cdc42 at the filopodia region resulted ..., but not in the light-treated PA-Cdc42DN cells themselves (Fig. 4i-l)” without showing the images of the light-treated PA-Cdc42DN cells. The authors need to provide the images that include the neighboring non- light-treated cells. Because of this lack of evidence, it is overstate to claim that “intercellular filopodia might maintain the medio-basal localization of stress fibers in their invaded cells”.

Answer: As reviewer suggested, we provided the images that include the neighboring non-light-treated cells. We updated this in the current main figure 5j, since our original images for this experiment are difficult to see the neighboring PA-Cdc42 clonal cells. However, we can see from images, some cells might be not easy to see the F-actin or myosin signals. This is due to the not good-focus issue (considering that some curved plane occurs for some cells, which is very often for a long time series of imaging) during live tissue imaging, which made difficult to achieve the good view of signals in all cells at different time points. Thus, we mainly focused on the cells where we can see the F-actin or myosin signals for the quantification. From our quantification, it is easy to see that wt cells have the effect of medio-basal distribution, but not the clonal cells, in this experiment.

[Responds to Answers #5]

The data shown in Fig. 5j and 5k are not clear. First, which cell is the cell of interest with PA-Cdc42DN? I guess the 3 red cells on the right hand side of the image are the ones that extend the filopodia to the WT cell highlighted with white dotted ellipse. Are those cell located at the DV side of the WT cells (my understanding is the filopodia is extended in DV axis)? In this figure, which cell is the "WT cell" the authors analyzed and presented in Fig. 5k. Furthermore, the reader can not understand how to read Fig. 5i from the provided figure legend. It is very clear that the author did not spend efforts to let the audience understand their results.

Even the author argued that "From our quantification, it is easy to see that wt cells have the effect of medio-basal distribution, but not the clonal cells, in this experiment.", the scientific claim is not convincing if we need to see the same trend in the images.

Moreover, I realized that quantifications of the PA-Cdc42 experiments are different from 1st manuscript (old Fig.4j) to revised manuscript (current Fig. 5k). Originally, the % of MyoII intensity before PA in PA-Cdc42DN was lower than WT (old Fig. 4j). Now, the values between PA-Cdc42DN and WT before PA are very close each other (current Fig. 5k). Why the trend of the data changed? As far as I can see, there is no description in the rebuttal saying the authors changed the data.

Although I understand this is a very difficult experiment, this is one of the most critical data for this manuscript. If one of the most critical data for this manuscript was not presented well, there is no other choice but to feel that the manuscript is not convincing.

#6. The downstream effectors of Cdc42, WASP, Ena, and Dia were claimed to be "distributed within the F-actin regions of both stress fibers and filopodia" (Supp. Figs. 3c-e). The staining signal of these effectors were everywhere without clear distributions, which made these images and claims unconvincing.

Answer: Thank you very much for the reviewer's careful reading. For WASP and Dia signals, they are distributed everywhere. It might be due to their control of not only stress fibers but also other F-actin structure located near basal junctions. For Ena, we can get many significant signals enriched at the both ends of stress fibers and also the clear (but weak) signals distributed at the tip region of

filopodia protrusion (please see the updated main figure 1f, g), while there is still some signals located near basal junctions. The wide distribution of these effectors at basal domain of epithelial cells indicates that these effectors might affect some other structures or play other functions. However, our RNAi knockdown experiments didn't show prominent effect on basal junctions while all RNAi significantly reduced filopodia and stress fiber signals/patterns, indicating that they mainly controls both stress fibers and filopodia. To better conclude our results, we revised our writing in the updated main text, in order to avoid our original unclear statement.

[Responds to Answers #6]

The Ena data is clear. There is no clear fiber-like structure in Dia and WASP data from control sample. It is very difficult to draw conclusion from these images.

#7. The authors claimed that "This supracellular network functions to maintain the medio-basal distribution and the global DV polarity of Myo-II contractility (Fig. 5j)." It has been shown that the global DV polarity of contraction is governed by the direction of ECM. Thus I felt that this is an overstatement and creates confusions. The authors need to thoroughly and fairly discuss on the global DV polarity.

Answer: For the global control of DV polarity of myosin contractility, it has been shown that the global DV polarity of contraction is governed by the direction of ECM. This is mainly due to the clockwise or anticlockwise rotation of follicle cells so that collagen and other matrix proteins are secreted and aligned along the DV axis. Through the interaction with integrin adhesion complex of follicle cells, stress fibers are mainly located at the DV axis direction. If we checked integrin RNAi phenotype, what we observed that stress fibers are strongly reduced but still mainly distributed along the DV axis and mainly at medial basal cortex of follicle cells from S9 to S10B (please see the following figures), indicating that in later stages, filopodia structure might function as a memory of mechanic cue to maintain the effect of ECM direction. To clarify this relationship between ECM direction and filopodia, we discussed the formation of the DV polarized stress fibers, the formation of filopodia, and how filopodia function as a type of memory to remember the effect of the egg chamber rotation and ECM polarity, so that after rotation is over, stress fibers are still well-organized by Cdc42 activity-mediated filopodia protrusions. However, it is unclear about the link and correlation between ECM-integrin adhesion and Cdc42 activity. Thus, we put this unclear information (needs future study) into the updated discussion section.

[Responds to Answers #7]

The authors defended their statement "This supracellular network functions to maintain the medio-basal distribution and the global DV polarity of Myo-II contractility (Fig. 5j, now Fig. 6k)." by providing the results of integrin RNAi data, which will not be shown in the manuscript". If the author

do not showed this data and discuss in the manuscript, I still feel that this statement is overstatement.

Minor comments

#c. In PA-Cdc42CA/DN experiments (Fig 3 &4), how the “% of medio-basal F-actin (or MyoII) intensity” was measured? Precise cell shape information is requires to measure this value. However, from the images presented, I could not see the clear cell shape. The authors need to 1) clarify how the cell shape measurement was done, and 2) show the images showed clear cell shape.

Answer: Sorry for this unclear information. But for quantification, we quantified the % of signal intensity distribution based on mCherry-PACdc42 signal which can show the membrane. In the following figures, we showed the example of cell shape at the beginning time point of all experiments. But mCherry-PACdc42 signals are often hidden by GFP signals, it is not easy to detect. But including both channels takes too much space for main figures. Thus, we didn't include this. If necessary, we can update the mCherry-PACdc42 channel in one supplementary figure.

[Responds to Answers #c]

How the authors know the shape of the WT cells without any clear fluorescent marker for the cell boundary? The information of cell boundary of WT cell should be a critical information to calculate the “% of medio-basal MyoII intensity” and generate Fig. 5k.

Reviewer #3 (Remarks to the Author):

The authors have provided comprehensive follow-up experiments to sufficiently address the concerns raised in the initial submission. In particular, validation of the optogenetic Cdc42 tools is now much more thorough. I have only a few very minor comments/suggestions:

1. For the PAK-RBD-GFP imaging experiments now included in Supplemental Fig 4B-C, the authors should include details of how the “relative PAK intensity” was calculated, either in the figure legend or in the “Methods”.

2. For the images shown in Supplemental Fig 4D, the coloring of the labels “PA-Cdc42DN” and “MyoII”/“F-actin” appear to be swapped vs. the color scheme used in the images themselves. Also, the individual panels should be labeled along their left edges as in Supplemental Figure 4B, for example.

3. For the Dia/WASP/Ena RNAi experiments conducted in *cdc42DN* cells (data shown to me but not included in the revised manuscript), I would recommend including at least the quantitative data as one additional panel in Supplemental Fig 3.

Dear editor,

We have now profoundly revised the paper in both text and data content. We thus kindly ask all reviewers to read the new revisited version of our paper from the very beginning. In addition you will find here below answers to reviewer's comments on our previous version of the paper.

Reviewers' comments:

Reviewer #1 (Remarks to the Author):

The authors have addressed my concerns and the manuscript overall has improved. The authors' findings are very interesting and the study has been carefully executed and documented. Overall, I find the manuscript excellent. I have only some minor points.

Minor points:

- The authors describe the filopodia as emerging from the basal 'domain'. However, as the filopodia form in continuation with the stress fibers, it seems more likely that they emerge from the basal-most aspect of the lateral membrane. If this correct, I would ask the authors to make according changes to the manuscript.

Response:

The respective changes have been made in our revised manuscript. We have revisited figure 1 to clarify the structure of filopodia and its link with mediobasal stress fibers and thus the formation of a supracellular network.

- Fig 3 h-j: The effect on Dia, WASP and Ena appears to be not restricted to the Cdc42 expressing cell but is also seen in the surrounding cells. Is the effect non-cell autonomous? Please, comment

Response:

We do agree with the suggestion of Reviwer#1. The non-cell autonomous effect of Dia, WASP and ENA distribution is highly possible, which could be in turn dependent on the level of DN-Cdc42 inhibitory effect.

- I strongly recommend to add the Sup Fig 4 panels f and g to main figure 4

Response:

We thank the reviewer for this suggestion. We now present the data showing the PA-Cdc42DN (photo-inhibitory Cdc42) effect on filopodia in the new main figure 4 (b and c). We left the light-insensitive control results (PA-Cdc42DN C450M on filopodia) in the supplementary session: see Supplementary Figure 5 f, g.

- The authors have now included a nice drawing, Fig 1b that illustrates the location of the intercellular filopodia. I suggest using thinner green lines to separate the stress fibers.

Response:

As Reviewer#1 suggested, we revised the schematic illustration to better represent the intercellular filopodia and stress fibers. This has now been moved from our previous main figure 1a and b to the current supplementary figure 2c and 2d.

- lines 98-101: please, revise the sentence as a "strong positive correlation between membrane and F-actin" does not specifically define filopodia.

Response:

As Reviewer#1 suggested, we removed this unclear sentence. Since the recruitment of ENA at the tip of protrusions is the most important cue identifying protrusion as filopodia, we revised our sentence and wrote: "By performing live imaging of Enabled (ENA), we could confirm that these protrusions are filopodia (Supplementary Fig. 2a, b)". This revision change is at lines 112-113 in our current main text.

- Grammar is a bit shaky in several parts of the manuscript, particularly in the introduction and M&M.

Response:

We now have improved this with particular attention to the introduction and M&M.

Figure 2a-c are not referenced in the manuscript text

Response:

We do reference Figure 2a-c in our previous text (at the line 107 in our previous manuscript text). These figures correspond to Figure 2b in our current version, together with PA-Cdc42DN results. Their respective description is in lines 135-139 in our current manuscript text.

Line 150: please, revise the title (not clear what it means)

Response:

We did the corresponding changes in the new title: "Optogenetics reveals Cdc42 spatial and temporal specific control" at line 163 in our current text.

- the term 'confirms/confirmed' is used in several places where "show or indicate" would be more appropriate (e.g. line 168)

Response:

We now made these changes.

- Line 199: I did not understand this sentence

Response:

In our current version we removed this unclear section. We now use infra-red laser dissection to directly test actomyosin tension. Given the profound revision of our manuscript, we kindly ask Reviewer#1 to take time to read our manuscript again in its integrity from the very beginning.

- Line 249 and 256: Supp Fig 8: indicate specific panels

Response:

This original supplementary figure 8 is currently supplementary figure 6. For clarity, we now use arrow heads to indicate the neighboring WT cells to clonal cells.

- line 279: replace 'tissue shape' with 'follicle shape'

Response:

Given the profound revision, we did send the paper as a first new submission. We thus took very seriously and addressed all major comments of the reviewers but did unfortunately overlook this minor comment. Since the paper was finally considered by the NC as a re-submission, we would be very willing to include this change in the next round.

- italics in M&M is used erratically. All genotypes should be indicated in italics

Response:

Given the profound revision, we did send the paper as a first new submission. We thus took very seriously and addressed all major comments of the reviewers but did unfortunately overlook this minor comment. Since the paper was finally considered by the NC as a re-submission, we would be very willing to include this change in the next round.

- line 357: what was the CoinFLP-lexA::GAD.GAL4 used for?

Response:

CoinFLP-LexA::GAD.GAL4 is a specific genetic system to express LexA-driven mCD8-FP or Gal4-driven mCD8-FP in different epithelial cells within the same tissue. By using this system, we get the results of mCD8GFP-expressing cells in contact with mCD8RFP-expressing cells, as shown in the new main figure 1f and 1g.

- line 359: change 'crossing transgenic flies with' to 'crossing flies with UAS transgenes to'

Response:

Given the profound revision, we did send the paper as a first new submission. We thus took very seriously and addressed all major comments of the reviewers but did unfortunately overlook this minor comment. Since the paper was finally considered by the NC as a re-submission, we would be very willing to include this change in the next round.

- line 458-459: "beneath the basal surface": is it not 'above'?

Response:

We are sorry for this confusion. This is the physical relation of objective position, we guess. Normally our imaging started from basal surface and went deep into the tissue. From this point of view, it is "beneath the basal surface" (beneath inside the cell), but not "above". We can update this change in our next turn.

- lines 511-513: I did not understand

Response:

Since the FRET measurement was controversial, we finally removed this part. We now use infra-red laser dissection to test cell and tissue mechanics.

- line 700 of text and line 5 of sup fig leg: replace 'random' with 'stochastic'

Response:

As Reviwer#1 suggested, we replaced “random” with “stochastic”.

- line 7 of Sup fig leg: reference for Indy-GFP is missing

Response:

As Reviwer#1 suggested, we put Indy-GFP information in the respective supplementary figure legends, at line 25 in our current supplementary information.

- line 126 of Sup fig leg: 'dotted line' ? - figure shows continuous lines

Response:

As Reviwer#1 suggested, we changed this confusing parts and used a dashed ellipse to show the region of photo-treatment. Please see the updated supplementary figure 7b, d, f.

- Sup fig 1i: what is the orientation of the cells regarding the D/V axis?

Response:

We thank reviewer#1 for the careful check. Given the profound revision, we did send the paper as a first new submission. We thus took very seriously and addressed all major comments of the reviewers but did unfortunately overlook this minor comment. Since the paper was finally considered by the NC as a re-submission, we would be very willing to include this change in the next round. We attached the updated supplementary figure 3 (3h) to adjust the DV axis as we show in all figures, for reviewer#1 to check this update. We can update this change in our next turn.

- Sup fig 2: Arrows, arrowheads

Response:

Due to the extensive revision of our manuscript, supplementary figure 2 has been replaced with the current main figure 1. All adequate changes have been done.

Comments on reviewer 2 report:

Reviewer 1: Point #2

I agree with Reviewer 2 that the E-cad tension sensor data do not necessarily reflect E-cad homophilic interactions (see Borghi N et al (PNAS 2012), and that the term 'E-cad adhesion tensions' should be avoided. The authors' response to the reviewer's comments could have been better. Despite this, I don't see a problem with the description of the experiment in the Result section where the authors state they used

the sensor "to quantify the mechanic tension of follicle cells in response to actomyosin contractility". And, I think that the authors are sufficiently cautious with their interpretation of their data with this sensor in the Result section. On the other hand, the statements about E-cad in the discussion seem too strong as it has not been shown that filopodia from neighboring cells adhere to each other through E-cad. I urge the authors' to be more cautious with their interpretation.

Response:

The paper has been profoundly reviewed and rewritten. In the new version of the paper, we removed the controversial E-cad tension sensor data and used a more direct and reliable tool to measure tension. We now implement infra-red laser dissection both at the cellular and tissue scales. Our new experimentations with laser manipulation provide clear and decisive results (new main Fig.6 and Fig.7). We also profoundly reviewed the discussion and avoided overstatements.

Reviewer 1: Point #3

I raised the same question about the homogeneous FRET signal around the whole cell in my first review. It looks as if E-cad is under tension along the whole membrane of the cell although the stress fibers are polarized. Nevertheless, the FRET signal clearly correlates with the oscillations in the contractility of the stress fiber network, and the FRET signal is present in the region of the filopodia. These data do not point to a special function of the filopodia in mechanosensing but it is consistent with the notion that mechanical forces act on DE-cad at these protrusions. It remains unanswered how exactly the filopodia regulate the polarity of stress fibers, whether simply through a much larger surface area (which increases the overall capacity of contact sites for actin bundles and for adhesive contacts) or through another mechanism. It will be interesting to figure out in the future how this works. An experiment that could directly address membrane tension in the interdigitating region of neighboring cells would be the measurement of recoil velocity after laser cuts (and comparing it to the behavior of the non-interdigitating membranes). Such experiments, however, would be challenging and have not been attempted in ovaries yet, as far as I know.

Response:

The FRET sensor experiments, resulting too controversial, have now been completely removed from our manuscript. We took very seriously Reviewers#1 suggestion and decided to implement laser manipulation. In the new version of the paper, we have used infra-red laser dissection. This is the first time that laser dissection of follicle cell stress fibers is reported. Laser dissection has been carried out along the egg chamber AP and DV axis at both cellular and tissue scales in wild type and Cdc42DN conditions. Infra-red laser dissection has been used also to shine new light on filopodia mechanical function. Our new results demonstrate subcellular and supracellular tension anisotropy in the *Drosophila* ovarian follicle.

Reviewer 1: Point #4

I agree with the request of the 2nd reviewer.

Response:

In our current version we removed this unclear section. We now use infra-red laser dissection to directly test actomyosin tension and tension anisotropy.

Reviewer 1: Point #5

I felt that the experiment has been clearly explained in the figure legend. Fig. 5j shows a wild-type cell outlined by a stippled line that is surrounded by five PA-Cdc42DN cells and one wild-type cell. The stippled line also indicates the applied ring of blue light illumination. The quality of Fig. 5j is good. In contrast, Fig. 5l (F-actin staining) is not interpretable. I strongly recommend adding images that only show the green channel for both, 5j and 5l. This would help a lot to better reveal the orientation and amount of stress fibers in PA-Cdc42DN and wild-type cells. In addition, outlining the cell boundaries of all cells in each image by dotted lines (different color for different genotypes) or highlighting the mutant and wild-type cells by asterisks (different colors) would be helpful. Furthermore, discrepancies between graphs of the original and revised manuscript, as pointed out by the 2nd reviewer, have to be resolved! This is indeed an important experiment. The English needs to be improved here and throughout the manuscript.

Response:

We thank the reviewers for this comment. We now improved our experimental design by selecting one WT cell surrounded by PA-Cdc42DN clones (PA-Cdc42DN C450M for control) as shown in our current main figure 5c-i and supplementary figure 7b-g. We included separate channel to better show cell boundaries and reiterated our quantifications. We also put much effort to improve the text as requested.

Reviewer 1: Point #6

In wild-type cells, Dia and Wasp puncta appear to be somewhat enriched in the region where the stress fibers are located. Even if there is no enrichment, there are clearly many puncta in the stress fiber area, which is consistent with a function there. To show the normal distribution of Dia and Wasp is valuable. Especially, as in Cdc42DN cells, the pattern seems to change and the central accumulation is no longer visible, consistent with a disturbance of the stress fibers.

My concern, as mentioned in my second review is that the neighboring wild-type cells of the Cdc42DN clone seem to have the same Dia and Wasp pattern as the mutant cell. Furthermore, the focal planes in the images of Fig. 3e-j are not quite comparable as seen by the mCD8-GFP staining, which appears in some images as a ring (lateral membrane) and in others as a flat surface (basal membrane).

Response:

We agree with the reviewers' comments. We now repeated the experiments by co-staining Dia, WASP and ENA with F-actin. The presence of medial basal F-actin can guarantee the correct focus that we used in our imaging and thus avoid confusion. These new images are in main figure 3e-g. Quantifications have been reiterated.

Reviewer 1: Point #7

Taking together all experimental data, I concur with the Authors' interpretation and model. I agree with the authors not to integrate the integrin data into the manuscript. They seem preliminary and to analyze the relationship between Cdc42 and integrin will require considerably more work, which in my opinion goes beyond the scope of this manuscript.

Response:

We do agree with Reviewer#1.

Reviewer #2 (Remarks to the Author):

First of all, I appreciated the authors efforts to clear my comments. I reviewed the comments and the revised manuscript carefully. After a rigorous evaluation, I found that some of the data were not presented/analyzed in a proper way, some of the data were not convincing, and some of the claims were overstated. Overall, the manuscript suffers from pitfalls that preclude publication in a high standard journal like Nature Communications. Here are my detailed comments.

Major comments

#2. The authors used the results from E-cad FRET force sensor to interpret the membrane tension. The force measured with E-cad FRET force sensor represent the tension exerted within E-cad, which is distinct from the membrane tension. The authors seem to miss-interpret the FRET signal of E-cad force sensor. This confusion needs to be addressed.

Answer: Thank you very much for the reviewer's careful reading. We are sorry to have miss-stated the interpretation of the results provided by the E-cad FRET force sensor. We revised the text to clarify that the E-cad tension sensor reports on the actomyosin-dependent tension on E-cad, rather than membrane tension. Since E-cad adhesion is tightly linked with actomyosin network at basal domain of follicle cells (as we reported previously, oscillatory myosin intensity is consistent with and 0.5min earlier than basal membrane area reduction change, mainly along DV axis direction), E-cad adhesion tension is highly consistent with the local/global actomyosin contractility. This is the reason why we used E-cad FRET force sensor to measure and quantify the tension on E-cad adhesion in both basal junctions and intercellular filopodia. And it really works to measure and quantify mechanical tension driven by myosin contractility. Since our goal was to measure the effect of myosin contractility on cell-cell interactions, we believe that the E-cad tension sensor is more informative than measuring membrane tension which can be affected by many other factors. We have clarified these points in the text.

[Responds to Answers #2]

First of all, the term "E-cadherin adhesion tensions" is confusing. To me, this implies the tension between E-cad through homophilic interactions. I do not think the E-cad tension sensor used in this manuscript is able to measure the tension between E-cad through homophilic interactions. Using "E-cadherin adhesion tensions" throughout the manuscript is not proper and is confusing.

The authors stated that "Since E-cad adhesion is tightly linked with actomyosin network at basal domain of follicle cells (as we reported previously, oscillatory myosin intensity is consistent with and 0.5min earlier than basal membrane area reduction change, mainly along DV axis direction)". It is not clear to me which data

shows “E-cad adhesion is tightly linked with actomyosin network at basal domain of follicle cells”. The time lag between the rate of decrease in basal area and change in myosin intensity, which the authors referring here (in the parentheses), does not logically support the statement “E-cad adhesion is tightly linked with actomyosin network at basal domain of follicle cells”.

The authors further stated that “And it really works to measure and quantify mechanic tension driven by myosin contraction ability.”. Without showing convincing evidences and/or arguments, it is impossible to believe a statement like “it really works to...”.

Last, the time lag between the rate of decrease in basal area and change in myosin intensity that the authors reported previously in He et al., (NCB, 2010) was ~0.9min, not 0.5min as the authors stated in this response. It looks like the authors did not pay proper attention to their own works.

Response:

The paper has been deeply reviewed and rewritten. Reviewer#2 comments are pertinent and we agree that E-cadTS experiments are controversial. We thus finally removed the E-cadTS part completely and definitely from the manuscript.

In our revised manuscript, we present new experiments using a more direct and reliable tool to measure tension. We implement infra-red laser dissection both at the cellular and tissue scales. Our new experiments with laser manipulation, for the first time applied on follicle cell stress fibers, provide clear and decisive results (new Fig.6 and Fig.7). Laser dissection has been carried out along the egg chamber AP and DV axis at both cellular and tissue scales in wild type and Cdc42DN conditions. Additional infra-red laser dissections have been carried out at the level of filopodia to shine new light on their mechanical function. Our new results demonstrate subcellular and supracellular tension anisotropy in the *Drosophila* ovarian follicle. Given the profound revision of our manuscript, we kindly ask Reviewer#2 to take time to read our manuscript again in its integrity from the very beginning.

#3. In Fig. 4a-b, although the authors highlighted that changes in tension at the intercellular filopodia region (within the ROIs), the images shows that the tension outside of the intercellular filopodia regions also correlated with the MyoII intensity within the cells, suggesting the change in tension is not restricted to the region related to intercellular filopodia. How the authors reconcile this? Please explain. This makes the author’s argument “intercellular filopodia might function as biomechanical sensors” unconvincing.

Answer: Thank you very much for the reviewer’s comments. Our previous writing might result in the confusion points. Although intercellular filopodia are inserted into stress fiber network in neighboring cells and meanwhile they can provide a physical cue/guidance to maintain the medio-basal and DV-polarized distribution of stress fibers in neighboring cells, we didn’t mean that intercellular filopodia would function as the only one structure to anchor stress fibers. As shown in the following cartoon, we can see that many stress fibers are anchored directly to the basal junctions at DV

axis direction. Between basal junctions and stress fibers, we can easily detect prominent F-actin signals (only F-actin signal at regions out of stress fibers). Thus, it seems that many stress fibers are anchored directly at basal junctions by these only F-actin regions; and this might explain that the tension outside of the intercellular filopodia regions, mainly at DV basal junctions, also correlated with the MyoII intensity within the cells. Interestingly, the vicinity of filopodia also showed strong E-cadTS tension by E-cad FRET ratio, which is highly correlated with local Myo-II intensity from stress fibers where filopodia are inserted. Although filopodia are not the only anchor points of stress fibers, from our data, it seems that this additional physic cue is efficient enough to highly maintain the distribution and polarity of stress fibers. This might be due to the much more E-cadherin adhesion tension supported by filopodia membrane regions, in addition to basal junctions, which can facilitate the distribution and polarity pattern, so that stress fibers would not disorganized after egg chamber rotation stops.

[Responds to Answers #3]

May be my previous comments were not clear for the authors. My major issue was that in the current Fig. 5a-b (originally Fig. 4a-b), the AP boundaries (vertical line in the cartoon the authors provided), which are parallel to the basal stress fibers and presumably do not interact with the basal stress fibers as strong as the DV boundaries, also change the FRET signal. In other words, there is no difference in the tension changes between AP boundaries and DV boundaries.

In He et al., (NCB, 2010), the authors reported that “The basal area change was highly polarized, correlating with a change in the length of cells along the D–V axis, whereas we observed little or no change in cell length along the anterior–posterior (A–P) axis (Fig. 2a)”. This strongly implied (and the authors claimed) that the contraction has direction and it is along D-V axis.

How the authors explain the discrepancy between the two papers? Why the results of E-cad tension sensor showed non-polarized tension, even the two different boundaries (AP vs DV) behave differently (i.e., one is changing its length, the other is not)?

Thus, I feel the author’s argument “the mechanosensory role of filopodia between cells” (in the original version, the authors claimed “intercellular filopodia might function as biomechanical sensors”) unconvincing.

Response:

Please see our response above, and also the corresponding new results showing subcellular and supracellular tension anisotropy along the DV axis of follicle. These new parts are included in our revised main figure 6 and 7. All FRET results and their related experiments and results have been completely removed from the paper.

#4. Based on the drug treatments (ionomycin and Y27632) to control the actomyosin contractility, the authors claimed that “stable actomyosin contractility is able to control the growth of intercellular filopodia”. It is not clear to me what “stable actomyosin contractility” means in the context of wild-type egg chamber. Please

clarify. In wild-type, the actomyosin contraction fluctuate in time, i.e., not stable. The other related question is whether the myosin density fluctuation diminished in these treatments and it looks like “stable” contractility. Please explain.

Answer: We are so sorry for this confusing writings. “Stable actomyosin contractility” should be actomyosin contractility without myosin intensity fluctuation. If myosin intensity is oscillatory, the force intensity is varied. In both drug treatments, the myosin fluctuation was strongly diminished. In the following imaging, we put the myosin intensity dynamics in Wt and Ionomycin treatment conditions so it is clear to see that myosin oscillation is diminished and myosin intensity is quite constant. Thus, we revised our writing to clarify this confusion. Please see the updated main text line “Considering the stochastic characteristic of pulsatile Myo-II contractility, here we used either Ionomycin or a chemical inhibitor of ROCK, Y27632, to enhance or inhibit basal actomyosin contractility, each of which blocks oscillatory behaviours, and thus constant basal Myo-II intensities were detected in both chemical treatments” in the current main text.

Ionomycin treatment strongly enhanced basal myosin intensity and meanwhile it blocked oscillatory behaviours. Although representative figures are not shown here, Y27632 treatment strongly inhibited basal myosin intensity and also blocked oscillatory behaviours.

[Responds to Answers #4]

The data showing the relative myosin intensity change over time under Ionomycin treatment should be shown at least in the supplement.

Why the authors could not show the similar results under Y27632 treatment? Please provide this and added to the supplement. I strongly think this is a very important result.

Response:

In our current version we removed this unclear and controversial section. We now use infra-red laser dissection to directly test actomyosin tension. Given the profound revision of our manuscript we kindly ask Reviwer#2 to take time to read our manuscript again in its integrity from the very beginning.

#5. The authors claimed that “Light-induced inhibition of Cdc42 at the filopodia region resulted ..., but not in the light-treated PA-Cdc42DN cells themselves (Fig. 4i-l)” without showing the images of the light-treated PA-Cdc42DN cells. The authors need to provide the images that include the neighboring non- light-treated cells. Because of this lack of evidence, it is overstate to claim that “intercellular filopodia might maintain the medio-basal localization of stress fibers in their invaded cells”.

Answer: As reviewer suggested, we provided the images that include the neighboring non-light-treated cells. We updated this in the current main figure 5j, since our original images for this experiment are difficult to see the neighboring PA-Cdc42

clonal cells. However, we can see from images, some cells might be not easy to see the F-actin or myosin signals. This is due to the not good-focus issue (considering that some curved plane occurs for some cells, which is very often for a long time series of imaging) during live tissue imaging, which made difficult to achieve the good view of signals in all cells at different time points. Thus, we mainly focused on the cells where we can see the F-actin or myosin signals for the quantification. From our quantification, it is easy to see that wt cells have the effect of medio-basal distribution, but not the clonal cells, in this experiment.

[Responds to Answers #5]

The data shown in Fig. 5j and 5k are not clear. First, which cell is the cell of interest with PA-Cdc42DN? I guess the 3 red cells on the right hand side of the image are the ones that extend the filopodia to the WT cell highlighted with white dotted ellipse. Are those cell located at the DV side of the WT cells (my understanding is the filopodia is extended in DV axis)? In this figure, which cell is the “WT cell” the authors analyzed and presented in Fig. 5k. Furthermore, the reader can not understand how to read Fig. 5i from the provided figure legend. It is very clear that the author did not spend efforts to let the audience understand their results.

Even the author argued that “From our quantification, it is easy to see that wt cells have the effect of medio-basal distribution, but not the clonal cells, in this experiment.”, the scientific claim is not convincing if we need to see the same trend in the images.

Moreover, I realized that quantifications of the PA-Cdc42 experiments are different from 1st manuscript (old Fig.4j) to revised manuscript (current Fig. 5k). Originally, the % of MyoII intensity before PA in PA-Cdc42DN was lower than WT (old Fig. 4j). Now, the values between PA-Cdc42DN and WT before PA are very close each other (current Fig. 5k). Why the trend of the data changed? As far as I can see, there is no description in the rebuttal saying the authors changed the data.

Although I understand this is a very difficult experiment, this is one of the most critical data for this manuscript. If one of the most critical data for this manuscript was not presented well, there is no other choice but to feel that the manuscript is not convincing.

Response:

We thank Reviewer#2 for this comment. We now improved our experimental design by selecting one WT cell surrounded by PA-Cdc42DN clones (PA-Cdc42DN C450M for control) as shown in our current main figure 5c-i and supplementary figure 7b-g. We included separate channels to better show cell boundaries and reiterated our quantifications.

#6. The downstream effectors of Cdc42, WASP, Ena, and Dia were claimed to be “distributed within the F-actin regions of both stress fibers and filopodia” (Supp. Figs. 3c-e). The staining signal of these effectors were everywhere without clear distributions, which made these images and claims unconvincing.

Answer: Thank you very much for the reviewer's careful reading. For WASP and Dia signals, they are distributed everywhere. It might be due to their control of not only stress fibers but also other F-actin structure located near basal junctions. For Ena, we can get many significant signals enriched at the both ends of stress fibers and also the clear (but weak) signals distributed at the tip region of filopodia protrusion (please see the updated main figure 1f, g), while there is still some signals located near basal junctions. The wide distribution of these effectors at basal domain of epithelial cells indicates that these effectors might affect some other structures or play other functions. However, our RNAi knockdown experiments didn't show prominent effect on basal junctions while all RNAi significantly reduced filopodia and stress fiber signals/patterns, indicating that they mainly controls both stress fibers and filopodia. To better conclude our results, we revised our writing in the updated main text, in order to avoid our original unclear statement.

[Responds to Answers #6]

The Ena data is clear. There is no clear fiber-like structure in Dia and WASP data from control sample. It is very difficult to draw conclusion from these images.

Response:

We repeated the experiments by co-staining Dia, WASP and ENA with F-actin. The presence of medial basal F-actin can guarantee the correct focus that we used in our imaging and thus avoid the confusion points and conclusions. These new images are in main figure 3e-g.

#7. The authors claimed that "This supracellular network functions to maintain the medio-basal distribution and the global DV polarity of Myo-II contractility (Fig. 5j)". It has been shown that the global DV polarity of contraction is governed by the direction of ECM. Thus I felt that this is an overstatement and creates confusions. The authors need to thoroughly and fairly discuss on the global DV polarity.

Answer: For the global control of DV polarity of myosin contractility, it has been shown that the global DV polarity of contraction is governed by the direction of ECM. This is mainly due to the clockwise or anticlockwise rotation of follicle cells so that collagen and other matrix proteins are secreted and aligned along the DV axis. Through the interaction with integrin adhesion complex of follicle cells, stress fibers are mainly located at the DV axis direction. If we checked integrin RNAi phenotype, what we observed that stress fibers are strongly reduced but still mainly distributed along the DV axis and mainly at medial basal cortex of follicle cells from S9 to S10B (please see the following figures), indicating that in later stages, filopodia structure might function as a memory of mechanic cue to maintain the effect of ECM direction. To clarify this relationship between ECM direction and filopodia, we discussed the formation of the DV polarized stress fibers, the formation

of filopodia, and how filopodia function as a type of memory to remember the effect of the egg chamber rotation and ECM polarity, so that after rotation is over, stress fibers are still well-organized by Cdc42 activity-mediated filopodia protrusions. However, it is unclear about the link and correlation between ECM-integrin adhesion and Cdc42 activity. Thus, we put this unclear information (needs future study) into the updated discussion section.

[Responds to Answers #7]

The authors defended their statement “This supracellular network functions to maintain the medio-basal distribution and the global DV polarity of Myo-II contractility (Fig. 5j, now Fig. 6k).” by providing the results of integrin RNAi data, which will not be shown in the manuscript”. If the author do not showed this data and discuss in the manuscript, I still feel that this statement is overstatement.

Response:

We agreed with Reviwer#2 comments. The results of integrin seem preliminary and the study of the relationship between Cdc42 and integrin will require considerably more work, which goes beyond the scope of this manuscript. To avoid confusion, we profoundly revised the discussion.

Minor comments

#c. In PA-Cdc42CA/DN experiments (Fig 3 &4), how the “% of medio-basal F-actin (or MyoII) intensity” was measured? Precise cell shape information is requires to measure this value. However, from the images presented, I could not see the clear cell shape. The authors need to 1) clarify how the cell shape measurement was done, and 2) show the images showed clear cell shape.

Answer: Sorry for this unclear information. But for quantification, we quantified the % of signal intensity distribution based on mCherry-PACdc42 signal which can show the membrane. In the following figures, we showed the example of cell shape at the beginning time point of all experiments. But mCherry-PACdc42 signals are often hidden by GFP signals, it is not easy to detect. But including both channels takes too much space for main figures. Thus, we didn't include this. If necessary, we can update the mCherry-PACdc42 channel in one supplementary figure.

[Responds to Answers #c]

How the authors know the shape of the WT cells without any clear fluorescent marker for the cell boundary? The information of cell boundary of WT cell should be a critical information to calculate the “% of medio-basal MyoII intensity” and generate Fig. 5k.

Response:

As mentioned above, we repeated the experiment. These are presented in our revised main figure 5f, 5h and supplementary figure 7d, 7f.

Reviewer #3 (Remarks to the Author):

The authors have provided comprehensive follow-up experiments to sufficiently address the concerns raised in the initial submission. In particular, validation of the optogenetic Cdc42 tools is now much more thorough. I have only a few very minor comments/suggestions:

1. For the PAK-RBD-GFP imaging experiments now included in Supplemental Fig 4B-C, the authors should include details of how the “relative PAK intensity” was calculated, either in the figure legend or in the “Methods”.

Response:

Given the profound revision, we did send the paper as a first new submission. We thus took very seriously and addressed all major comments of the reviewers but did unfortunately overlook this minor comment. Since the paper was finally considered by the NC as a re-submission, we would be very willing to include this change in the next round. We attached the updated supplementary figure 5 (5c) to clarify the details about the “relative PAK intensity”, for reviewer#3 to check this update. We can update this change in our next turn.

2. For the images shown in Supplemental Fig 4D, the coloring of the labels “PA-Cdc42DN” and “MyoII”/“F-actin” appear to be swapped vs. the color scheme used in the images themselves. Also, the individual panels should be labeled along their left edges as in Supplemental Figure 4B, for example.

Response:

Given the profound revision, we did send the paper as a first new submission. We thus took very seriously and addressed all major comments of the reviewers but did unfortunately overlook this minor comment. Since the paper was finally considered by the NC as a re-submission, we would be very willing to include this change in the next round. We attached the updated supplementary figure 5 (5d) to correct this mistake and label individual panels clearly, for reviewer#3 to check this update. We can update this change in our next turn.

3. For the Dia/WASP/Ena RNAi experiments conducted in cdc42DN cells (data shown to me but not included in the revised manuscript), I would recommend including at least the quantitative data as one additional panel in Supplemental Fig 3.

Response:

Although co-expression of RNAi (Dia/WASP/ENA) and Cdc42DN can enhance the phenotypes, the effect is mild and thus we feel it is better not to include these results in our manuscript.

Reviewers' comments:

Reviewer #1 (Remarks to the Author):

Nature Communication NCOMMS-17-08771B-Z

A Cdc42-mediated supracellular network drives polarized forces and tissue extension by Popkova et al.

The authors have now removed the controversial E-cad tension sensor data and replaced them with direct analysis of cell and tissue tension, based on recoil velocities after infrared laser cuts. This is an important change. The authors have also included a better analysis of the filopodia-based membrane interdigitations between cells.

I am delighted about the laser cut experiments shown in Fig 6 and 7. I find the last two experiments in Fig 6 (f,g) particularly interesting as they show that Cdc42DN cells disturb tension in their neighbors, but only in their immediate neighbors, which is consistent with the short-range non-autonomous effect on stress fiber order shown in Fig 5a. Overall the tissue seems to be quite robustly buffered against local disturbance.

Nevertheless, I have remaining concerns:

A. Main critique points

1) Effects on filopodia:

Disruption of Cdc42 appears not only to affect the length of filopodia but also their orientation and location. I find it surprising that the authors only mention the shorter length of filopodia although multiple images show that in Cdc42DN cells, filopodia emerge from all around the cell perimeter and point in random directions (see Figure 2d, 3e, 6c, S6c).

Importantly, either ROCK RNAi or Rho1DN (dependent on which image is which, see point 7) also seems to affect the direction and location of filopodia although filopodia length might not be affected. Similar to Cdc42DN, filopodia emerge from all around the cell perimeter and point in

different directions. The 'order parameter' also seems to be affected (Fig S6a,b). The difference in the stress fiber order parameter might not have reached significance due to the very small sample size.

The conclusion of the authors (R144/145): "The results show that Cdc42 is required for filopodia formation and to organize the basal actomyosin network whereas Rho1 controls the medial actomyosin oscillations" is NOT consistent with the obvious effects of Rho1/ROCK on the cellular location and directionality of filopodia and the orientation of stress fibers. The importance of Cdc42 for sub-basal filopodia and actin stress fiber is not in question, but Rho1/ROCK appears to play a role, too.

2) Filopodia length: Images clearly show that there are substantial variations in the size of the filopodia. The graphs (see all figures and Sup figures), however, don't show this. It is important that the authors indicate SD (instead of SEM) in the graphs!! I also like to ask that all bar graphs that show filopodia length are converted to Dot blots or Dot blot-Box blot combinations. Also, I think that the Student test might be more appropriate than the Mann-Whitney test for comparing filopodia length as a normal distribution is expected.

3) Egg shape: The images of Fig 7e do not convince me. The *cdc42[4] st14* egg looks squashed and the *CDC42DN* egg looks generally smaller. Also, the follicles (on the left) are not of the same stage and look slightly squashed as well. Pictures of eggs have to be taken without a coverslip.

4) The cell non-autonomous effect of *cdc42DN* cell clones is striking and seems not only to affect the orientation of the stress fibers as illustrated in Figure 5A but also the distribution of the actin bundles (lateral vs medial, e.g. Fig 2d, S6c). If the authors agree with this, I would encourage them to mention out.

5) Fig 2A suggests that Cdc42 is rather homogeneously distributed in the cell cortex. Its activity polarized? If a Cdc42 activity sensor (similar to the Rho1 activity sensors) is available, I would like to encourage the authors to test it.

6) I believe that the ROCK and Rho1 genotype labels were mixed up either for Figure S6a or the two bottom panels of S6c. The phenotype in the upper image of panel a (labeled ROCK) looks like the bottom image in c (labeled Rho). And the phenotype in the lower image of panel a (labeled Rho1) looks like the 3rd row image in c (labeled ROCK).

B. Minor points:

It is hard to understand why the authors would ignore the 'minor comments' of the previous reviews and say in their response that they would address them in the next revision.

I am not certain whether the term 'compenetrate' is suitable (frequently used in the manuscript), as this term means 'to penetrate throughout' or 'penetrate every aspect'.

R56-57: Follicles are round from S1-S5. Follicle elongation starts at S6.

R142: I suggest adding that the oscillations have a smaller amplitude

Methods:

- I highly recommend adding tables to Suppl materials, listing (1) all constructs and other fly lines, including their specific designation/number and origin and reference, and (2) drugs and antibodies. This is now rather common practice and very useful.

- Please, check whether heatshock times are correctly indicated. Heatshock times for Flpout clones seem excessive to me.

Improve grammar/sentence structure:

R100 .. elongation at S9-S10

R108/109 'membrane clones' ????? (revise)

R203 whether

R254 'while fiber actin patterns' ??? (revise)

R341 beginning at S10B

R363 CyO

R373 analysis in mosaic tissues

R460 4°C

R713 stochastic induction of mCD8GFP expression

R758 delete one 'in'

R782 indicate specific cdc42 allele

Figures:

Figure 3: add scalebar to panel i

Figure 4:

- Images in panels d,e,f,g appear fuzzy/pixelated. The same images looked much better in the previous submission. Has the resolution decreased??

- in f,g, change to: cdc42^[4] MARCM clone (indicate '4' in superscript)

Reviewer #4 (Remarks to the Author):

The authors investigate supracellular actomyosin networks in follicle cells connected through Cdc42-dependent filopodia. Overall, I found the manuscript interesting and well executed. My main criticism is with overstatement of the novelty of some findings and the corresponding lack of credit to previous authors. I also have a few questions related to the comments by Reviewer 2 and how they have been addressed, particularly with regards to the laser ablation experiments:

- First, I think that the authors should cite a previous study in Nature Communications by Valencia-Exposito and colleagues that had already conducted laser ablation of basal actin networks in follicle cells, and suggested the presence of anisotropic forces. In general, I think the authors should make an effort to give credit to previous studies in the field. For example, the polarized distribution of actin in follicle cells was shown at least ten years ago (Viktorinova et al., Development, 2009) and there may be earlier reports showing the same.

- In 6f-g, the initial opening in the myosin network shown in the kymographs does not match the much smaller differences from the 2 sec time point in the en face views. Are the kymographs scaled differently in f and g?

- Laser cutting experiments used to argue for supracellularity of the actomyosin network: is the delay between the images before and after ablation constant (regardless of the length of the ablated line)? This should be clarified. If the time in between those two images is different depending on the length of the line that they ablate (longer lines should take longer to ablate if they are scanning the laser), comparing recoil velocities immediately after ablation (line 266) is wrong, as the estimation of the immediate recoil velocities will be affected by that delay (although the authors would likely be underestimating the difference). It would be good to clarify exactly how velocities are being calculated (i.e. what is in the denominator when they divide distance/time).

- Line 258: I think the idea that one would expect multiple gaps to open when conducting a line ablation, even if the networks operated independently, does not make a lot of sense. Long line ablations are likely damaging the membranes in between cells, junctional structures, etc., so regardless of the connectivity of the actomyosin network, I'd expect a single opening. Perhaps if the authors attempted to ablate segments along a line while "skipping" the points where the line crosses interfaces between cells ...

Other comments:

1. Does overexpression of Ena in Cdc42N follicle cells restore the tension anisotropy in the myosin network?

2. Throughout the manuscript, the authors claim differences in filopodial lengths of less than 1 μm (e.g. Figures 2c or 3b). How many pixels is that? Do they have the resolution to make those claims? Why not quantify number of protrusions instead, is that metric not affected by disruption of Cdc42 signalling?

3. Line 142: while it is true that there are still basal oscillations in Cdc42DN cells, they are greatly attenuated, and this should be acknowledged in the text.

4. Figure 4f, g: The authors should measure filopodial length when they rescue cdc42 mutant cells with PA-Cdc42CA.

5. Please, review the manuscript carefully, there are lots of typos ...

Reviewers' comments:

Reviewer #1 (Remarks to the Author):

Nature Communication NCOMMS-17-08771B-Z

A Cdc42-mediated supracellular network drives polarized forces and tissue extension by Popkova et al.

The authors have now removed the controversial E-cad tension sensor data and replaced them with direct analysis of cell and tissue tension, based on recoil velocities after infrared laser cuts. This is an important change. The authors have also included a better analysis of the filopodia-based membrane interdigitations between cells.

I am delighted about the laser cut experiments shown in Fig 6 and 7. I find the last two experiments in Fig 6 (f,g) particularly interesting as they show that Cdc42DN cells disturb tension in their neighbors, but only in their immediate neighbors, which is consistent with the short-range non-autonomous effect on stress fiber order shown in Fig 5a. Overall the tissue seems to be quite robustly buffered against local disturbance.

Nevertheless, I have remaining concerns:

A. Main critique points

1) Effects on filopodia:

Disruption of Cdc42 appears not only to affect the length of filopodia but also their orientation and location. I find it surprising that the authors only mention the shorter length of filopodia although multiple images show that in Cdc42DN cells, filopodia emerge from all around the cell perimeter and point in random directions (see Figure 2d, 3e, 6c, S6c).

Response:

We agree with Reviewer 1 and thus updated the quantification of filopodia distribution including filopodia orientation at D and V locations (Supplementary Fig. 4a).

Importantly, either ROCK RNAi or Rho1DN (dependent on which image is which, see point 7) also seems to affect the direction and location of filopodia although filopodia length might not be affected. Similar to Cdc42DN, filopodia emerge from all around the cell perimeter and point in different directions. The 'order parameter' also seems to be affected (Fig S6a,b). The difference in the stress fiber order parameter might not have reached significance due to the very small sample size.

The conclusion of the authors (R144/145): "The results show that Cdc42 is required for filopodia formation and to organize the basal actomyosin network whereas Rho1 controls the medial actomyosin

oscillations" is NOT consistent with the obvious effects of Rho1/ROCK on the cellular location and directionality of filopodia and the orientation of stress fibers. The importance of Cdc42 for sub-basal filopodia and actin stress fiber is not in question, but Rho1/ROCK appears to play a role, too.

Response:

We now quantified and compared filopodia directionality (angle) for ROCK RNAi, Rho1DN, Cdc42DN and wt (Supplementary Fig. 4b). Our analysis show that, while in ROCK RNAi and Rho1 DN there is a wider distribution of filopodia angles compared to wild type, most of the angle values measured still fall along a direction parallel to the DV axis (Supplementary Fig. 4b, boxes). In contrast, Cdc42DN expression results in a stronger phenotype with filopodia directionality being much more stochastic. Std for control (14.34), cdc42DN (44.98), RhoDN (24.29) and ROCK RNAi (22.03) were included into Supplementary Fig. 4 legend.

In addition, we found that we used the incorrect way to quantify the order parameter by comparing the angle of each cell with dorsal axis. Now we redid the quantification of order parameter by comparing each pair of cells within the same category and updated them (in Fig. 5b and Supplementary Fig.8b). We also increased the samples size for the stress fiber order parameter analysis (Supplementary Fig. 8b). All these newly corrected results corroborate our conclusions that inhibition of Rho1 or ROCK in follicle cells does not affect the stress fiber organization in the neighboring cells, compared with the dramatic effect from the inhibition of Cdc42. Furthermore, we measured the angles of stress fibers and plotted the values in the same way as for filopodia directionality (Supplementary Fig. 8e). This new analysis corroborates the conclusion that Cdc42 but not Rho1 can control the polarized distribution of stress fibers.

2) Filopodia length: Images clearly show that there are substantial variations in the size of the filopodia. The graphs (see all figures and Sup figures), however, don't show this. It is important that the authors indicate SD (instead of SEM) in the graphs!! I also like to ask that all bar graphs that show filopodia length are converted to Dot blots or Dot blot-Box blot combinations. Also, I think that the Student test might be more appropriate than the Mann-Whitney test for comparing filopodia length as a normal distribution is expected.

Response:

We understand the Reviewer concern here. Nevertheless, the reason why filopodia variability is not highlighted is because for all graphs, representing the comparison of filopodia length in different genotypes/treatments, a single data point represents the average filopodia length per cell. We now clarify this point in the figure legend. For the same reason, we believe that average and SEM are appropriate for comparison in the different genetic backgrounds/treatments. We finally opted for the Mann-Whitney test since this is appropriate also for datasets which do not have a normal distribution.

3) Egg shape: The images of Fig 7e do not convince me. The cdc42[4] st14 egg looks squashed and the CDC42DN egg looks generally smaller. Also, the follicles (on the left) are not of the same stage and look slightly squashed as well. Pictures of eggs have to be taken without a coverslip.

Response:

We now repeated the experiment as Reviewer 1 suggested and updated the images and quantification (Fig 7e, f).

4) The cell non-autonomous effect of *cdc42DN* cell clones is striking and seems not only to affect the orientation of the stress fibers as illustrated in Figure 5A but also the distribution of the actin bundles (lateral vs medial, e.g. Fig 2d, S6c). If the authors agree with this, I would encourage them to mention out.

Response:

We agreed with Reviewer 1 and mentioned this information in the main text (page 10).

5) Fig 2A suggests that Cdc42 is rather homogeneously distributed in the cell cortex. Is its activity polarized? If a Cdc42 activity sensor (similar to the Rho1 activity sensors) is available, I would like to encourage the authors to test it.

Response:

As suggested by the Reviewer 1, we performed further experiments and analysis to try to answer this question. We monitored the Cdc42 activity using the Cdc42 FRET probe. This probe allowed detecting the global Cdc42 activity at the junctional cortex but not subtle subcellular differences (see Figure 1). We also tested the WASP-RBD-GFP probe without obtaining valid information (see Figure 2). Finally, since both experiments were not conclusive and needed further investigation, we decided not to include them in the manuscript.

Figure 1. Cdc42 FRET activity and D/V ratio in follicle cells.

a. Representative FRET ratio and MyoII images in the basal domain of control or Cdc42DN-expressing follicle cells. b, c. Average values of Cdc42 FRET activity (b) and FRET activity D/V ratio (c) in the control and Cdc42DN-expressing follicle cells.

Figure 2. Cdc42 activity detected by WASP-RBD-GFP probe.

Representative images of WASP-RBD-GFP and MyoII-RFP at the basal domain of the control follicle cells.

6) I believe that the ROCK and Rho1 genotype labels were mixed up either for Figure S6a or the two bottom panels of S6c. The phenotype in the upper image of panel a (labeled ROCK) looks like the bottom image in c (labeled Rho). And the phenotype in the lower image of panel a (labeled Rho1) looks like the 3rd row image in c (labeled ROCK).

Response: We doubled checked this and confirm that the labels are correct.

B. Minor points:

It is hard to understand why the authors would ignore the 'minor comments' of the previous reviews and say in their response that they would address them in the next revision.

Response:

We corrected the unaddressed minor comments from the previous reviews.

I am not certain whether the term 'compenetrate' is suitable (frequently used in the manuscript), as this term means 'to penetrate throughout' or 'penetrate every aspect'.

Response:

We now use the term 'penetrate' instead of 'compenetrate'.

R56-57: Follicles are round from S1-S5. Follicle elongation starts at S6.

Response:

We corrected this information in main text (page 3).

R142: I suggest adding that the oscillations have a smaller amplitude

Response:

We added this information in main text (page 7).

Methods:

- I highly recommend adding tables to Suppl materials, listing (1) all constructs and other fly lines, including their specific designation/number and origin and reference, and (2) drugs and antibodies. This is now rather common practice and very useful.

Response:

We agreed with Reviewer 1 and added 2 supplementary tables.

- Please, check whether heatshock times are correctly indicated. Heatshock times for Flpout clones seem excessive to me.

Response:

Heatshock times for Flpout clones are based on our previous publications (He et al, NCB 2010; Qin et al., NC 2017). And heatshock times are correctly indicated.

Improve grammar/sentence structure:

R100 .. elongation at S9-S10

R108/109 'membrane clones' ????? (revise)

R203 whether

R254 'while fiber actin patterns' ??? (revise)

R341 beginning at S10B

R363 CyO

R373 analysis in mosaic tissues

R460 4°C

R713 stochastic induction of mCD8GFP expression

R758 delete one 'in'

R782 indicate specific cdc42 allele

Figures:

Figure 3: add scalebar to panel i

Figure 4:

- Images in panels d,e,f,g appear fuzzy/pixelated. The same images looked much better in the previous submission. Has the resolution decreased??

- in f,g, change to: cdc42^[4] MARCM clone (indicate '4' in superscript)

Response:

All these have been changed / improved.

Reviewer #4 (Remarks to the Author):

The authors investigate supracellular actomyosin networks in follicle cells connected through Cdc42-dependent filopodia. Overall, I found the manuscript interesting and well executed. My main criticism is with overstatement of the novelty of some findings and the corresponding lack of credit to previous authors. I also have a few questions related to the comments by Reviewer 2 and how they have been addressed, particularly with regards to the laser ablation experiments:

- First, I think that the authors should cite a previous study in Nature Communications by Valencia-Exposito and colleagues that had already conducted laser ablation of basal actin networks in follicle cells, and suggested the presence of anisotropic forces. In general, I think the authors should make an effort to give credit to previous studies in the field. For example, the polarized distribution of actin in follicle cells was shown at least ten years ago (Viktorinova et al., Development, 2009) and there may be earlier reports showing the same.

Response:

We agree with Reviewer#4 to give credit to previous work. We now cite the paper from Valencia-Exposito et al. at page 14. We now cite Viktorinova et al. at page 5.

- In 6f-g, the initial opening in the myosin network shown in the kymographs does not match the much smaller differences from the 2 sec time point in the en face views. Are the kymographs scaled differently in f and g?

Response:

We thank Reviewer#4 to spot that out: we now corrected the kymograph.

- Laser cutting experiments used to argue for supracellularity of the actomyosin network: is the delay between the images before and after ablation constant (regardless of the length of the ablated line)? This should be clarified. If the time in between those two images is different depending on the length of the line that they ablate (longer lines should take longer to ablate if they are scanning the laser), comparing recoil velocities immediately after ablation (line 266) is wrong, as the estimation of the immediate recoil velocities will be affected by that delay (although the authors would likely be underestimating the difference).

Response:

We confirm that the time between the image before and after ablation is constant regardless of the length of the ablated line. With our set-up, the time to make a horizontal line scan is always the same since the galvanometric mirror pivots of the same angle amount. It is the laser shutter that is opened/closed at the right moment during the line scan: this does not impact on timing. We now refer to this in the Methods.

It would be good to clarify exactly how velocities are being calculated (i.e. what is in the denominator when the divide distance/time).

Response:

The denominator is 2 seconds. Velocities were calculated using PIV as mentioned in the methods.

- Line 258: I think the idea that one would expect multiple gaps to open when conducting a line ablation, even if the networks operated independently, does not make a lot of sense. Long line ablations are likely damaging the membranes in between cells, junctional structures, etc., so regardless of the connectivity of the actomyosin network, I'd expect a single opening. Perhaps if the authors attempted to ablate segments along a line while "skipping" the points where the line crosses interfaces between cells ...

Response:

We understand Reviewers #4 concern.

Nevertheless, infra-red (IR) femtosecond (fs) laser ablation has already been demonstrated to be a technique capable of high resolved dissection of the actomyosin network with little or no perturbation of the cell membrane (Rauzi et al. NCB 2008, Rauzi and Lenne Methods Mol. Boil. 2015) differently from other dissection techniques using UV/green picosecond/nanosecond lasers. Our tissue scale dissection experiments corroborate this by showing that membrane are preserved since they (i) are visible after ablation and (ii) stretch over time under the action of tissue scale unbalanced forces (Fig. 7 a, Supplementary movie 11).

On the other hand, we do agree that the F-actin network, free of MyoII and located at the apical-lateral cortex (Fig. 1 j), is also dissected by performing a continuous line ablation. Since in this study we focus more specifically on the actomyosin stress-fiber network, we performed the experiment suggested by Reviewer # 4. We performed tissue scale IR fs dissections over a segmented line to avoid exposing cortical-lateral cell regions. The tissue scale segmented ablation shows overall the same phenotype resulting in one single large opening (see Supplementary movie 13). This evidence shows that egg-chamber surface mechanics during stage 10 is dominated by tension generated by the medial-basal actomyosin stress-fiber network.

We now add the following paragraphs at pages 13-14 to point this out:

"IR fs laser ablation has been demonstrated to be capable of high resolved dissection of the actomyosin network with little or no perturbation of the cell membrane {Rauzi et al. NCB 2008, Rauzi and Lenne Methods Mol. Boil. 2015}. Our experiments corroborate this notion: membrane are preserved since they (i) are visible after ablation and (ii) stretch under the action of tissue scale unbalanced forces (Fig. 7 a, Supplementary Video 11). These evidences ensure that the cell unities are preserved after IR fs tissue scale dissection."

"Tissue scale IR fs ablation results in the dissection of the basal network including both the medial-basal actomyosin stress fibers and the F-actin junctional network, free of MyoII, located at the apical-lateral cortex (Fig. 1 j). In order to better understand the mechanical role of the medial-basal actomyosin stress fibers network (which is the focus of our study) we performed a tissue scale segmented ablation to dissect the actomyosin stress fibers while avoiding the cell junctional regions (probed already in a

previous study {Valenica-Exposito et al. 2016}). The tissue scale segmented ablation still resulted in one single large opening (see Supplementary movie 13). This evidence shows that egg-chamber surface mechanics, during stage 10, is dominated by tension generated by the medial-basal actomyosin stress-fiber network.”.

Other comments:

1. Does overexpression of Ena in Cdc42N follicle cells restore the tension anisotropy in the myosin network?

Response:

To answer Reviewers #4 question, we now performed laser dissection experiments using Ena Cdc42DN egg chambers. Under these conditions the the D-V tension was restored. This is presented in (Supplementary Fig. 10). We also added in the main text in page 12 the following sentence: “Finally tension was restored by overexpressing Ena in Cdc42DN follicle cells (Supplementary Fig. 10).”.

2. Throughout the manuscript, the authors claim differences in filopodial lengths of less that 1 μm (e.g. Figures 2c or 3b). How many pixels is that? Do they have the resolution to make those claims?

Response:

Yes, we do: 1 μm corresponds to 8 pixels.

Why not quantify number of protrusions instead, is that metric not affected by disruption of Cdc42 signalling?

Response:

We agree with Reviewer #4 that counting filopodia is another way of assessing the data, nevertheless, we preferred to measure filopodia length to avoid being biased by other protrusions as for instance lamellipodia.

3. Line 142: while it is true that there are still basal oscillations in Cdc42DN cells, they are greatly attenuated, and this should be acknowledged in the text.

Response:

We now acknowledge this in the text as mentioned by Reviewer #4.

4. Figure 4f, g: The authors should measure filopodial length when they rescue cdc42 mutant cells with PA-Cdc42CA.

Response:

We now performed filopodia rescue in cdc42 mutant clones using the PA-Cdc42CA. The result is presented in (Supplementary Fig. 7f, g).

5. Please, review the manuscript carefully, there are lots of typos ...

Response:

We thank the Reviewer#4 for pointing this out: we reviewed the manuscript to correct additional typos.

Reviewers' comments:

Reviewer #1 (Remarks to the Author):

Nature Communication NCOMMS-17-08771B-Z

A Cdc42-mediated supracellular network drives polarized forces and tissue extension by Popkova et al.

The important points of the reviewers have been addressed through additional experiments, new or improved data, and improved data evaluation. In this reviewer's opinion, the described research data and the manuscript are now excellent and highly interesting to a broad readership.

A few points left:

1) Statistics for filopodia length: The authors have partially clarified their data collection and I agree that using SEM is correct in this case. Missing, however, is a description of the number of filopodia that were measured per cell and how the filopodia were chosen if not all were measured. Also, cells from how many clones and follicles were evaluated for filopodia length? Please, provide details in M&M and/or figure legends.

The authors made an effort to show their newer graphs in dot/box blot format. I highly recommend replacing most of the other bar diagrams with dot/box blot diagrams as well. Especially those showing filopodia length.

2) Terminology regarding epithelial cells:

Row 285-287: "... F-actin junctional network, free of MyoII, located at the apical-lateral cortex". What is the evidence that the apical-lateral cortex of follicle cells is MyoII "free". Obviously, this is an overstatement, and either a reference or data should to be provided showing that MyoII is not detectable at the apical junctional complex.

Discussion: row 339-342: Sentence is very confusing. First, follicle cells are epithelial cells. Second, it is not clear what is meant by junctions at the basal side. Third, it is not clear why authors claim that there is no actomyosin ("actomyosin free"). The filopodia contain F-actin by definition and MyoII is likely associated with the base of the filopodia even though those amounts might be small when compared to the basal stressfibers. Please, correct and clarify this sentence.

The authors frequently use terms, such as: "junctional cortex" or "cell junctional region", which are confusing as they refer to the basal/sub-basal region of epithelial cells. Please, revise. For example, one can speak of sub-basal cell-cell contact zone and sub-basal cortex.

Replace "follicle tissue" with 'follicular epithelium' (row 342, 347). 'Follicle' refers to follicle cells plus germline cells.

4) A description of the control genotypes is missing. Please, add all genotypes in the figure legends, or alternatively, add a table that lists complete genotypes of controls and experimental flies for all experiments.

5) Figure 2 legend says: "MyoII was visualized using a MyoII-mCherry construct". Clarify: sph or zip?

6) row 282/283: Sentence not logic. Please, revise. e.g.: 'These observations suggest that cell unities are preserved ...'

7) alleles are indicated in superscript (not [x]) according to genetic nomenclature.

8) SFIG 7: change "PACdc42CA" to PA-Cdc42CA in panel f

9) M&M: Authors state that codes and data sets "are available ... on reasonable request". "reasonable" sounds vague and subjective. I suggest stating simply: 'available on request'

Reviewer #4 (Remarks to the Author):

The authors have addressed most of my concerns. However, the new experiment with the segmented ablations (Supp. Movie 13) should be quantified, and the recoil velocities compared to the ablation of individual medial networks. I still do not see how the single opening demonstrates network connectedness. Perhaps the authors should find a tissue in which networks are not

interconnected to compare and contrast with the follicular epithelium? Otherwise, I would just concentrate on the differences in recoil velocity when ablating one or multiple networks.

Reviewers' comments:

Reviewer #1 (Remarks to the Author):

A few points left:

1) Statistics for filopodia length: The authors have partially clarified their data collection and I agree that using SEM is correct in this case. Missing, however, is a description of the number of filopodia that were measured per cell and how the filopodia were chosen if not all were measured. Also, cells from how many clones and follicles were evaluated for filopodia length? Please, provide details in M&M and/or figure legends.

The authors made an effort to show their newer graphs in dot/box blot format. I highly recommend replacing most of the other bar diagrams with dot/box blot diagrams as well. Especially those showing filopodia length.

Answer: As reviewer suggested, we now indicated the criteria for filopodia measured in M&M, the number of filopodia per cell in the legend for Figure2, and also the number of egg chambers analysed in all figure legends. We used the dot/box plot format to show the quantification of filopodia length in all related figures.

2) Terminology regarding epithelial cells:

Row 285-287: "... F-actin junctional network, free of MyoII, located at the apical-lateral cortex". What is the evidence that the apical-lateral cortex of follicle cells is MyoII "free". Obviously, this is an overstatement, and either a reference or data should to be provided showing that MyoII is not detectable at the apical junctional complex.

Answer: We thank the reviewer for pointing this out. In this sentence we inadvertently wrote apical instead of basal. We now corrected this: "While actin bundles extend from medial stress fibers to the sub-basal cell-cell contact zone forming filopodia, MyoII is not enriched at this zone and concentrates instead in the medial region (Fig. 1j)".

Discussion: row 339-342: Sentence is very confusing. First, follicle cells are epithelial cells. Second, it is not clear what is meant by junctions at the basal side. Third, it is not clear why authors claim that there is no actomyosin ("actomyosin free"). The filopodia contain F-actin by definition and MyoII is likely associated with the base of the filopodia even though those amounts might be small when compared to the basal stressfibers. Please, correct and clarify this sentence.

Answer: As reviewer suggested, we corrected this confusing sentence. We now write: " In contrast to other epithelial cells, in which MyoII is enriched at junctions generating forces that are directly transmitted from cell to cell ^{38, 42}, in follicle cells MyoII is not enriched at basal junctions and concentrates instead in the medial region. How can then forces be transmitted from one cell to the other at the basal domain in the follicular epithelium? "

The authors frequently use terms, such as: "junctional cortex" or "cell junctional region", which are confusing as they refer to the basal/sub-basal region of epithelial cells. Please, revise. For example, one can speak of sub-basal cell-cell contact zone and sub-basal cortex.

Answer: As reviewer suggested, we now corrected these confusing expressions.

Replace "follicle tissue" with 'follicular epithelium' (row 342, 347). 'Follicle' refers to follicle cells plus germline cells.

Answer: We do agree with the reviewer and we thank him/her for pointing this out. We now corrected this as the reviewer suggested.

4) A description of the control genotypes is missing. Please, add all genotypes in the figure legends, or alternatively, add a table that lists complete genotypes of controls and experimental flies for all experiments.

Answer: As reviewer suggested, we now added all information about the control genotypes in figure legends.

5) Figure 2 legend says: "MyoII was visualized using a MyoII-mCherry construct". Clarify: sph or zip?

Answer: As reviewer suggested, we now clarified this in the text.

6) row 282/283: Sentence not logic. Please, revise. e.g.: 'These observations suggest that cell unities are preserved ...'

Answer: As reviewer suggested, we modified this sentence.

7) alleles are indicated in superscript (not [x]) according to genetic nomenclature.

Answer: We now corrected this as reviewer suggested.

8) SFIG 7: change "PACdc42CA" to PA-Cdc42CA in panel f

Answer: As reviewer suggested, we corrected this.

9) M&M: Authors state that codes and data sets "are available ... on reasonable request". "reasonable" sounds vague and subjective. I suggest stating simply: 'available on request'

Answer: We now changed this as reviewer suggested.

Reviewer #4 (Remarks to the Author):

The authors have addressed most of my concerns. However, the new experiment with the segmented ablations (Supp. Movie 13) should be quantified, and the recoil velocities compared to the ablation of individual medial networks. I still do not see how the single opening demonstrates network connectedness. Perhaps the authors should find a tissue in which networks are not interconnected to compare and contrast with the follicular epithelium? Otherwise, I would just concentrate on the differences in recoil velocity when ablating one or multiple networks.

Answer: As suggested by the reviewer, we now measured the recoil speed of cell networks in segmented ablations and compared it with the recoil speed of networks in simple ablations (shown in Supplementary Fig. 11 and movie 13). As expected, cell networks in segmented ablations recoil with a speed that is more than threefold greater. In the main text we thus added the sentence: "In addition, network recoil maximum speed after segmented dissection was more than 3-fold greater than recoil after single cell network dissection (Supplementary Fig. 11)."

REVIEWERS' COMMENTS:

Reviewer #4 (Remarks to the Author):

The authors have addressed my concerns.

Just a note that the new nomenclature introduced in response to reviewer 1 is somewhat troublesome. Adherens junctions localize at the interface between the apical and the basolateral plasma membranes, so referring to the junctional cortex as sub-basal is going to confuse many readers. Do the authors mean sub-apical when they use sub-basal? Or are they referring to cell contacts just apical to the basal surface (and in that case, wouldn't it be best to use "suprabasal")?

REVIEWERS' COMMENTS:

Reviewer #4 (Remarks to the Author):

The authors have addressed my concerns.

Just a note that the new nomenclature introduced in response to reviewer 1 is somewhat troublesome. Adherens junctions localize at the interface between the apical and the basolateral plasma membranes, so referring to the junctional cortex as sub-basal is going to confuse many readers. Do the authors mean sub-apical when they use sub-basal? Or are they referring to cell contacts just apical to the basal surface (and in that case, wouldn't it be best to use "suprabasal")?

Answer: Different from the normal distribution of adherens junction at the interface between the apical and basolateral plasma membranes, in S9-S10 egg chamber follicular epithelial cells, adherens junction is also distributed at the sub-basal cell-cell contact region. This phenomenon has been reported in our previous papers (He et al., Nat Cell Biol, 2010; Qin et al., Nat Commun, 2017). Actually, sub-basal means the small regions just close to basal surface, shown below and also as indicated in our previous paper (Qin et al., Nat Commun, 2017). Oppositely, we think that supra-basal might be more confusing, and this definition might contain the sub-basal contact region, the medio-basal stress fibers region, and also the basal surface, shown in the below cartoon. Therefore, we think that it will be better to maintain the nomenclature introduced in response to reviewer 1.